# Adapting to Evolving Graphs: A Scalable Framework for Dynamic Coarsening

**Abhishek Gupta** [1]   **Manoj Kumar** [2]   **Sarthak Kumar Singh** [3]   **Ujjwal Yadav** [3]   **Yifan Sun** [4]   **Sandeep Kumar** [1 5 3]

## Abstract

Graph coarsening is a fundamental dimensionality reduction technique for scaling large graphs while preserving structural and feature information. However, most existing coarsening methods are designed for static graphs and do not extend well to dynamic settings where nodes, edges, and connectivity patterns evolve over time. Recomputing a coarsened graph from scratch after every update is often infeasible, which limits scalability and real-time applicability. To address this, we propose a unified framework for coarsening discrete-time dynamic graphs by incrementally updating the coarsening mapping matrix. The framework initializes from any static coarsening technique and then efficiently incorporates real-world graph events, including node additions, node deletions, and edge modifications. We instantiate this framework with two optimization based incremental update algorithms tailored to different dynamic regimes, one focusing on efficiently integrating growth related changes and another handling broader topology evolution with adaptive reassignment. We derive fast and scalable solvers with convergence guarantees, and provide theoretical guarantee via $\epsilon$-similarity bounds that quantify and control quality degradation in the coarsened graph. Extensive experiments under realistic dynamic scenarios show substantial improvements in runtime and memory, delivering significant speedups while maintaining or improving downstream task performance, including graph neural network accuracy.

[1]Yardi ScAI, IIT Delhi, India [2]MFSDSAI, IIT Roorkee, India [3]Department of EE, IIT Delhi, India [4]Department of CS, Stony Brook University, United States [5]BSTTM, IIT Delhi, India. Correspondence to: Abhishek Gupta <aiz248316@iitd.ac.in>, Manoj Kumar <manoj.kumar@mfs.iitr.ac.in>, Sarthak <ee1210673@iitd.ac.in>, Ujjwal <ee1210669@iitd.ac.in>, Yifan Sun <yifan.sun@stonybrook.edu>, Sandeep Kumar <ksandeep@iitd.ac.in>.

*Proceedings of the 43$^{rd}$ International Conference on Machine Learning*, Seoul, South Korea. PMLR 306, 2026. Copyright 2026 by the author(s).

## 1. Introduction

Graphs are fundamental to advancements in big data and machine learning, enabling breakthroughs in diverse domains such as social networks and opinion dynamics (Sharma et al., 2024; Alfas et al.), neuroscience (Wang et al., 2021; Gupta et al.), circuits (Shahane et al., 2023), and drug discovery (Gaudelet et al., 2021; Jiang et al.). These structures model relationships and interactions effectively, but the increasing size and complexity of real-world graphs pose significant computational and storage challenges. Dimensionality reduction techniques, such as graph coarsening (Hashemi et al., 2024a; Cai et al., 2021; Huang et al., 2021; Jin et al., 2020; Dickens et al., 2024; Kumar et al., 2023a), have emerged as an essential tool, compressing graphs while preserving their structural and feature-related properties.

Numerous methods, including Local variation (Loukas, 2019), Feature-aware Graph Coarsening (FGC) (Kumar et al., 2023b; 2024; Joly & Keriven, 2024), deep learning based (Dickens et al., 2024) have been developed for coarsening static graphs. These approaches consider various aspects of coarsening, such as structural similarity, smoothness guarantee, message passing guarantee, feature preservation, and convergence guarantees. While effective in static scenarios, these methods struggle to adapt to the dynamic nature of evolving graphs, where relationships, distributions, and topologies change over time. Current solutions require complete re-computation of the coarsening matrix upon graph updates, leading to inefficiencies and making them unsuitable for real-time or large-scale applications. This limitation is particularly acute in domains such as social networks with rapidly evolving user interactions, financial systems responding to market shifts, and molecular simulations tracking temporal changes.

Several studies have applied graph coarsening in evolving settings, but with domain-specific goals. Examples include robotic manipulation (Marchetti et al., 2021), grain growth modeling (Semiatin et al., 2006), streaming visualization (Sun & Zeng, 2024), dynamic layouts (Veldhuizen, 2007), subgraph query acceleration (Kansal & Spezzano, 2017), and reduced-order dynamics (Yu et al., 2025). These approaches prioritize task-specific objectives over structural preservation. Related spectral sparsification methods (Durfee et al., 2019; Chen et al., 2020) focus on cut or flow

estimation, not learning tasks. However, these techniques typically work on continuous temporal graphs and have not been evaluated on real-world discrete dynamic datasets such as ACM and DBLP, where the graph evolves at discrete intervals. To the best of our knowledge, no existing work addresses efficient and structure-preserving coarsening under such discrete dynamic conditions. For a detailed definition of dynamic graphs, please refer to *Appendix B*.

To address these challenges, we propose a novel dynamic graph coarsening framework that learns a coarsened graph at discrete time instants. In graph coarsening, the key objective is to learn a mapping matrix that assigns nodes in the original graph to supernodes in the coarsened graph. In our framework, we first obtain an initial mapping matrix at time t=0 using any state-of-the-art static graph coarsening technique. As the graph evolves, we have developed optimization-based algorithms that incrementally update this mapping matrix at subsequent time instants, avoiding repeated full-scale recomputation. The proposed optimization based algorithms are fast and provably convergent, and it explicitly handles realistic graph evolution events such as node addition, node deletion, and edge modifications. When used for downstream tasks on the coarsened graph, such as scaling graph neural networks, the proposed approach delivers significant speedups, reduces space complexity, and outperforms or performs competitively with state-of-the-art methods in most scenarios, making it well-suited for dynamic graph settings.

Our approach addresses dynamic graph coarsening under two key scenarios. In the first scenario, Incremental Growth, the graph evolves through node addition, node deletion, and edge updates that primarily affect newly added or removed regions, while the previously existing structure remains largely unchanged. The objective in this setting is to efficiently incorporate these changes into the coarsening matrix without recomputing the full coarsening. In the second scenario, Evolving Topology, the graph undergoes broader structural changes, where both the existing part of the graph and newly introduced components may change simultaneously due to node events and edge modifications. This setting requires adaptive updates to the coarsening matrix that accurately track evolving connectivity, while maintaining computational efficiency and preserving the structural integrity of the coarsened representation. In addition, we provide an $\epsilon$-similarity guarantee that quantifies the quality of the coarsened graph relative to the evolving original graph, ensuring that the coarsened representation remains representative even under substantial structural changes.

This work introduces a dynamic maintenance view of graph coarsening for discrete-time evolving graphs. Starting from any static coarsening method, the proposed framework maintains and updates the node-to-supernode mapping across graph snapshots, avoiding repeated full static re-coarsening. Within this framework, DIGC uses the block structure of the evolving adjacency matrix to update the mapping of newly added nodes through $\mathbf{\Delta C_t}$, while keeping previous assignments fixed for efficiency. ACNR further introduces adaptive reassignment through $\mathbf{\Delta C_{S_t}}$ together with the new-node assignment $\mathbf{\Delta C_{N_t}}$, allowing previously coarsened nodes to adjust under topology changes. The framework also incorporates alignment and pruning for deletions and non-monotonic graph evolution, and provides an $\epsilon$-similarity analysis specialized to incremental dynamic coarsening.

In summary, this work bridges the gap between static graph coarsening methodologies and the evolving needs of dynamic graph-based systems. Our framework integrates seamlessly with existing methods, empowering them with incremental adaptability while ensuring scalability, efficiency, and maintaining structural properties.

## 2. Background and Problem Formulation

In this section, we will review the concepts of graph theory, graph coarsening, and related works corresponding to static and dynamic graphs. For clarity, Appendix A summarizes the main notation used throughout the paper.

### 2.1. Graph

An undirected graph with associated features can be represented as a quadruplet $\mathcal{G} = (\mathcal{V}, \mathcal{E}, \mathcal{W}, X)$, where $\mathcal{V}$ denotes the set of nodes with $|\mathcal{V}| = N$, and $\mathcal{E} \subset \mathcal{V} \times \mathcal{V}$ represents the set of edges, with $|\mathcal{E}| = M$. The matrix $X \in \mathbb{R}^{n \times d}$ captures the node features, where each row $\mathbf{x}_i \in \mathbb{R}^d$ corresponds to the feature vector of the $i^{\text{th}}$ node. Additionally, the weight matrix $\mathcal{W} \in \mathbb{R}^{N \times N}$ encodes the edge weights between nodes. Graphs are commonly represented using two primary connectivity matrices: the adjacency matrix ($\mathbf{A}$) and the Laplacian matrix ($\mathbf{L}$), each possessing distinct properties and applications. The adjacency matrix $\mathbf{A} \in \mathbb{R}^{N \times N}$ is defined as (Merris, 1994):

$$A_{ij} = \begin{cases} w_{ij}, & \text{if } (i,j) \in \mathcal{E}, \\ 0, & \text{otherwise.} \end{cases}$$

where $w_{ij}$ denotes the weight of the edge $(i, j)$. On the other hand, the combinatorial Laplacian matrix $\mathbf{L} \in \mathbb{R}^{N \times N}$ is a symmetric, positive semi-definite matrix ($\mathbf{L} \succeq 0$) and is defined as (Merris, 1994):

$$\mathbf{L} = \mathbf{D} - \mathbf{A},$$

where $\mathbf{D}$ is a diagonal degree matrix with diagonal elements $D_{ii} = d_i = \sum_{j=1}^{n} A_{ij}$, where $d_i$ denotes the degree of the $i^{\text{th}}$ node.

## 2.2. Graph Coarsening

Graph coarsening is a graph dimensionality reduction technique that aims to learn a coarsened graph $\mathcal{G}_c = (\mathcal{V}_c, \mathcal{E}_c, \mathcal{W}_c, X_c)$ from an original graph $\mathcal{G} = (\mathcal{V}, \mathcal{E}, \mathcal{W}, X)$ such that $|\mathcal{V}_c| = n \ll N$ and $|\mathcal{E}_c| = m \ll M$, while preserving the essential structural and feature-based properties of the original graph.

Graph coarsening aims to learn a coarsening or mapping matrix $C \in \mathbb{R}_+^{N \times n}$, where each nonzero entry $C_{ij}$ indicates that the $i^{th}$ node in the original graph $\mathcal{G}$ is mapped to the $j$th super-node in the coarsened graph $\mathcal{G}_c$. The relationships between the original graph Laplacian $\mathbf{L} \in \mathbb{R}^{N \times N}$, the coarsened graph Laplacian $\mathbf{L}_c \in \mathbb{R}^{n \times n}$, and the mapping matrix $C \in \mathbb{R}_+^{N \times n}$, along with the feature mapping between the original graph feature matrix $\mathbf{X}$ and the coarsened graph feature matrix $\mathbf{X}_c$, are given by (Loukas, 2019; Loukas & Vandergheynst; Kumar et al., 2024; 2023b):

$$\mathbf{L}_c = C^\top \mathbf{L} C, \qquad X_c = C^\dagger X = PX \qquad (1)$$

where $P = C^\dagger \in \mathbb{R}_+^{n \times N}$ is the projection matrix. For a balanced mapping, the matrix $C$ belongs to the set (Kumar et al., 2023b):

$$\mathcal{C} = \big\{ C \in \mathbb{R}_+^{N \times n} \mid C \geq 0, \langle C_i, C_i \rangle = d_i \ \forall \ i \in [n],$$

$$\langle C_i, C_j \rangle = 0 \ \forall \, i \neq j, \|C_i\|_0 \geq 1, \& \, \|[C^\top]_i\|_0 = 1 \big\} \quad (2)$$

where $C_i$ and $C_j$ are the $i^{th}$ and $j^{th}$ orthogonal columns and $[C^\top]_i$ denotes the $i^{th}$ row of $C$. Several static graph coarsening techniques have been proposed in the literature. For instance, the approach presented in (Loukas, 2019) is heuristic-based and relies solely on the structure of the original graph. In contrast, the methods described in (Kumar et al., 2023b), (Kumar et al., 2023a), and (Kumar et al., 2024) are optimization-based techniques that consider either the Laplacian and feature information, or the Laplacian, feature, and some node label information of the graph. Furthermore, (Cai et al., 2021) introduces a deep learning-based graph coarsening algorithm.

All these state-of-the-art techniques aim to learn a mapping matrix $C \in \mathcal{C}$ that maps the original graph nodes to supernodes in the coarsened graph. However, with the increasing prevalence of dynamic graphs that evolve over time, applying these static methods repeatedly becomes computationally expensive and memory-intensive. Each update would require running the entire algorithm from scratch, making it impractical for large-scale and frequently changing graphs.

Several studies have explored graph coarsening in evolving settings, but primarily for domain-specific applications. For example, (Marchetti et al., 2021) models deformable-object dynamics in robotic manipulation using dynamic pooling

on coarsened graphs, without preserving standard adjacency for general analytics. In (Semiatin et al., 2006), coarsening represents grain growth in titanium alloys, a diffusion-driven physical process rather than an algorithmic one. Similarly, (Sun & Zeng, 2024) proposes a two-stage streaming visualization framework using linear projection and GNN-based cluster pooling to ensure smooth visual transitions, unlike DGC, which emphasizes structural fidelity under arbitrary updates. (Veldhuizen, 2007) focuses on aesthetic consistency in dynamic layouts through a multilevel force-directed method, while (Kansal & Spezzano, 2017) accelerates subgraph queries via dynamic node merging, prioritizing performance over structure preservation. Lastly, (Yu et al., 2025) develops a GNN-based framework for non-Markovian modeling of coarse-grained dynamics, aimed at reduced-order modeling rather than general-purpose learning. Beyond coarsening, works such as (Durfee et al., 2019) and (Chen et al., 2020) address dynamic spectral sparsification using Schur complements to preserve spectral properties under updates. However, these are tailored to tasks like dynamic cut or flow estimation or query approximation, not to learning on evolving graphs.

Collectively, these dynamic coarsening techniques (Marchetti et al., 2021; Semiatin et al., 2006; Sun & Zeng, 2024; Veldhuizen, 2007; Kansal & Spezzano, 2017; Yu et al., 2025) are designed for specific domains such as robotics, materials science, visualization, layout design, and query optimization, and typically assume continuous-time updates. More details in Appendix C. However, in real-world dynamic graphs such as citation networks, nodes and edges often arrive at different discrete time instants, introducing additional challenges in maintaining structural consistency and fidelity. To date, no existing work addresses this discrete and asynchronous update setting. Importantly, existing baselines have not been designed or evaluated on such evolving datasets, for example, ACM and DBLP, in discrete dynamic scenarios.

To address this challenge, a dynamic graph coarsening algorithm is required that can efficiently update the coarsened graph by considering only the newly introduced nodes at each time step, thereby significantly reducing computational overhead and memory consumption.

**Problem Statement:** Given an evolving graph $G$ at different time instants $t_1, t_2, \ldots, t_n$ (i.e., $G_{t_1}, G_{t_2}, \ldots, G_{t_n}$), our aim is to learn the corresponding coarsened graphs $G_{c_{t_1}}, G_{c_{t_2}}, \ldots, G_{c_{t_n}}$ for these time instants.

Dynamic Graph Coarsening (DGC) aims to efficiently maintain an updated coarsened representation of evolving graphs while preserving their structural properties. To achieve this, we introduce an optimization framework that encompasses two key approaches: **Dynamic Incremental Graph Coars-**

ening (DIGC) and **Adaptive Coarsening with Node Reassignment (ACNR)**. These approaches progressively refine the mapping matrix to accommodate evolving graph structures.

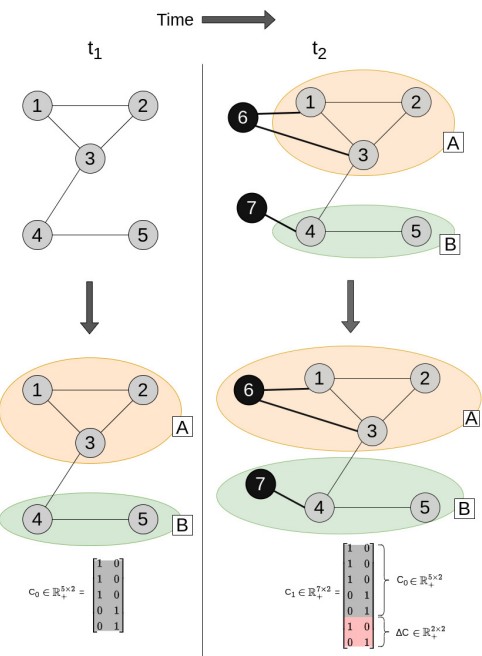

*Figure 1.* Toy example: Illustration of dynamic graph coarsening with incremental updates to the coarsening matrix. At time $t_1$, suppose the input large graph is coarsened into two super-nodes, $A$ and $B$, with an initial coarsening matrix $C_1 \in \mathbb{R}_+^{5\times2}$. At time $t_2$, the graph evolves with the addition of two new nodes (6 and 7) and new edges. Instead of recomputing the coarsening from scratch, the updated coarsening matrix $C_2 \in \mathbb{R}_+^{7\times2}$ is constructed by incorporating the previous coarsening matrix $C_1$ and an incrementally computed matrix $\Delta C \in \mathbb{R}_+^{2\times2}$.

## 2.3. Proposed DIGC Framework

Given an evolving graph $G_{t_1}, G_{t_2}, \ldots, G_{t_n}$, the proposed formulation is to learn the coarsened graph at time instants $t_1, t_2, \ldots, t_n$:

$$\begin{aligned} \underset{\mathbf{\Delta C_t}}{\text{minimize}} \quad & \frac{\omega}{2}f(A_{c_t}, A_{c_{t-1}}) + \frac{\zeta}{2}g(\mathbf{\Delta C_t}, X_{sn}, X_n) \\ \text{subject to} \quad & \mathbf{\Delta C_t} \geq 0, \quad \mathbf{\Delta C_t} \cdot \mathbf{1}_{n\times1} = \mathbf{1}_{k\times1} \end{aligned} \quad (3)$$

where $A_{c_t} \in \mathbb{R}^{n\times n}$ and $A_{c_{t-1}} \in \mathbb{R}^{n\times n}$ are the coarsened adjacency matrices at the $t$-th and $t-1$-th time instants, respectively. The initial coarsened adjacency matrix $A_{c_0} \in \mathbb{R}^{n\times n}$ is obtained using the state-of-the-art static graph coarsening technique. The function $f(.)$ ensures that the coarsened graph at the $t$-th time instant remains similar to the coarsened graph at the $t-1$-th time instant, implying that the deviation is minimal. This reflects the assumption that only a small number of nodes are added to the original graph over time, and the deviation in the

coarsened graph structure should not be significant. The newly added nodes are mapped to the super-nodes of the coarsened graph by learning $\Delta C_t \in \mathbb{R}^{k\times n}$ where $k$ is the number of nodes added in the original graph at each time instant $t$. The function $g(.)$ maps the features of the newly added nodes to the corresponding super-nodes in the coarsened graph. Also, $X_{sn}$ denotes the features of the existing super-nodes, and $X_n$ represents the features of the newly added nodes. Once the mapping matrix for the new nodes, $\mathbf{\Delta C_t}$, is determined at time $t$, the overall mapping matrix $C_t \in \mathbb{R}^{(N+k(t+1))\times n}$ can be obtained by vertically concatenating $\mathbf{\Delta C_t} \in \mathbb{R}^{k\times n}$ with the previously computed mapping matrix $C_{t-1} \in \mathbb{R}^{(N+kt)\times n}$ from time $t-1$. Here, $N$ is the number of nodes in the original graph at time instant $t_0$. This concatenation is given as:

$$C_t = [C_{t-1} \quad \mathbf{\Delta C_t}]^\top \quad (4)$$

This process is applied iteratively at each time interval, ensuring that we only compute the incremental mapping for newly introduced nodes. As a result, the coarsening mapping matrix grows with time, while preserving computation from previous time steps and avoiding an expensive re-computation of the entire mapping.

## 2.4. Proposed ACNR Framework

Unlike DIGC, which constrains previously coarsened nodes to their initial mappings, ACNR allows these mappings to evolve dynamically in response to changes in connectivity and the addition of new nodes. This dynamic adaptation enables a more flexible coarsening process, where the mapping matrix $C_{t-1}$, representing node assignments at time $t-1$, can be refined to better reflect the evolving graph structure while integrating newly introduced nodes.

In this framework, the term $\mathbf{\Delta C_{N_t}}$ represents the mapping of newly added nodes, analogous to $\mathbf{\Delta C_t}$ in the previous method. Additionally, $\mathbf{\Delta C_{S_t}}$ accounts for adjustments in the mappings of previously existing nodes, ensuring optimal assignments as the graph evolves.

$$\begin{aligned} \underset{\mathbf{\Delta C_{N_t}}, \mathbf{\Delta C_{S_t}}}{\text{minimize}} \quad & \frac{\omega}{2}f(A_{c_t}, A_{c_{t-1}}) + \frac{\zeta}{2}g(\mathbf{\Delta C_{N_t}}, X_{sn}, X_n) \\ & + \frac{\beta}{2}h(C_{t-1}, \mathbf{\Delta C_{S_t}}, \mathbf{\Delta C_{N_t}}) + \frac{\alpha}{2}\mathcal{T}(A_{c_{t-1}}^{\text{new}}, A_{c_{t-1}}) \\ \text{s.t.} \quad & \mathbf{\Delta C_{N_t}} \geq 0, \ (C_{t-1} + \mathbf{\Delta C_{S_t}}) \geq 0, \ \mathbf{\Delta C_{N_t}} \cdot \mathbf{1} = \mathbf{1} \end{aligned}$$
$$(5)$$

The first two terms retain the same interpretation as described earlier for (3). The function $h(C_{t-1}, \mathbf{\Delta C_{S_t}}, \mathbf{\Delta C_{N_t}})$ represents the node remapping consistency, which adjusts the mappings of previously existing nodes to reflect changes in graph connectivity and the influence of new nodes. This function refines the existing mapping matrix $C_{t-1}$ with the adjustment matrix $\mathbf{\Delta C_{S_t}}$, ensuring that the evolving structure is captured.

The term $\mathcal{T}(A_{c_{t-1}}^{\text{new}}, A_{c_{t-1}})$ focuses on Graph Topology Refinement, ensuring structural consistency in the coarsened graph by incorporating newly formed edges between existing nodes. This prevents drastic changes while allowing for fine-grained updates to the mapping matrix. Here, $A_{c_{t-1}}$ represents the coarsened adjacency matrix before the new edges are introduced, while $A_{c_{t-1}}^{\text{new}}$ reflects their updated connectivity at time $t$. By maintaining this refinement, the framework ensures that structural updates are smoothly integrated without significant deviation from the prior representation, thereby preserving the integrity of the coarsened graph over successive iterations. After computing the refinement matrices $\Delta\mathbf{C_{S_t}}$ and $\Delta\mathbf{C_{N_t}}$, the overall mapping matrix $C_t$ at time $t$ is updated as:

$$C_t = \begin{bmatrix} C_{t-1} + \Delta\mathbf{C_{S_t}} \\ \Delta\mathbf{C_{N_t}} \end{bmatrix} \quad (6)$$

This process is iteratively applied over time, enabling efficient incremental updates and continuous refinement of the mapping matrix.

## 3. Algorithm and Convergence Analysis

In this section, we develop the proposed DIGC and ACNR algorithms and analyze their convergence rates and the $\epsilon$-similarity bound.

### 3.1. DIGC Algorithm Development

Consider $f(.) = \|A_{c_t} - A_{c_{t-1}}\|_F^2$ and $g(.) = \|\Delta\mathbf{C_t} X_{sn} - X_n\|_F^2$. The proposed formulation for learning $\Delta\mathbf{C_t}$ at the $t_n$-th time instant is:

$$\underset{\Delta\mathbf{C_t}}{\text{minimize}} \quad \frac{\omega}{2}\|A_{c_t} - A_{c_{t-1}}\|_F^2 + \frac{\zeta}{2}\|\Delta\mathbf{C_t} X_{sn} - X_n\|_F^2$$
$$\text{subject to} \quad \Delta\mathbf{C_t} \geq 0, \quad \Delta\mathbf{C_t} \cdot 1 = 1. \quad (7)$$

where $A_{c_t}$ and $A_{c_{t-1}}$ are the adjacency matrices of the coarsened graphs at time intervals $t$ and $(t-1)$, respectively. The term $X_{\text{sn}}$ denotes the features of the existing super-nodes, while $X_n$ represents the features of newly added nodes. The parameters $\omega$ and $\zeta$ are hyperparameters that control structural consistency and feature similarity, respectively.

Next, the adjacency matrix at $t$-th time instant is given by:

$$A_{c_t} = C_t^\top A_t C_t$$

where $A_t$ is the adjacency matrix of the original graph at time $t$. The adjacency matrix at the $t$-th time instant is defined as:

$$A_t = \begin{bmatrix} M_1 & M_2 \\ M_3 & M_4 \end{bmatrix}, \quad (8)$$

where $M_1 = A_{t-1}$ is the adjacency matrix at time instant $(t-1)$, representing the connectivity of nodes at $(t-1)$.

Similarly, $M_4$ represents the adjacency matrix of newly added nodes at time instant $t$. The off-diagonal blocks $M_2$ and $M_3 = M_2^\top$ capture the connectivity between nodes at time instants $t$ and $(t-1)$.

Substituting the value of $C_t$ from equation 4, the adjacency matrix of the coarsened graph at the $t$-th time instant is:

$$A_{c_t} = \begin{bmatrix} C_{t-1} \\ \Delta\mathbf{C_t} \end{bmatrix}^\top \begin{bmatrix} M_1 & M_2 \\ M_3 & M_4 \end{bmatrix} \begin{bmatrix} C_{t-1} \\ \Delta\mathbf{C_t} \end{bmatrix}. \quad (9)$$

Finally, the difference between $A_{c_t}$ and $A_{c_{t-1}} = C_{t-1}^\top A_{t-1} C_{t-1}$ is:

$$A_{c_t} - A_{c_{t-1}} = \Delta\mathbf{C_t}^\top M_2^\top C_{t-1} + C_{t-1}^\top M_2 \Delta\mathbf{C_t}$$
$$+ \Delta\mathbf{C_t}^\top M_4 \Delta\mathbf{C_t}. \quad (10)$$

The optimization problem in Equation (7) is a non-convex optimization problem. Still, we can obtain the solution using the gradient descent technique by relaxing the constraint $\Delta\mathbf{C_t} \cdot 1 = 1$ to a $\|\Delta\mathbf{C_t} \cdot 1 - 1\|_F^2$ penalty. The update rule for $\Delta\mathbf{C_t}$ at the $(n+1)$-th iteration is given by:

$$[\Delta\mathbf{C_t}]^{n+1} \leftarrow \left[[\Delta\mathbf{C_t}]^n - \eta\nabla f(\Delta\mathbf{C_t}^n)\right]^+. \quad (11)$$

where $[x]^+ = \max\{x, 0\}$, and the gradient of $f(\Delta\mathbf{C_t})$ is: $\nabla f(\Delta\mathbf{C_t}) = 2\omega\left((M_2^\top C_{t-1} + M_4\Delta\mathbf{C_t})(A_{c_t} - A_{c_{t-1}})\right) + \zeta\left((\Delta\mathbf{C_t} X_{\text{sn}} - X_n)X_{\text{sn}}^\top\right) + 2(\Delta\mathbf{C_t} \cdot 1_{n \times 1} - 1_{k \times 1})1_{n \times 1}^\top$.

**Lemma 3.1.** *The optimization problem defined in* (7), *assuming each feature is normalized such that* $\|X_n\|_F \leq k$ *and* $\|X_{sn}\|_F \leq n$. *For any feasible* $\Delta\mathbf{C_t}$, *the function* $f(\Delta\mathbf{C_t})$ *has bounded function constants* $L$, $F$, *and* $\sigma$; *these constants correspond to the function's smoothness, maximum value, and maximum gradient norm, respectively. Specifically,*

$$L = O(\|M_2\|n^{4.5}(N + kt)k(\|M_2\|_2(N + kt) + \|M_4\|k)),$$
$$F = O(n^6 k^2(\|M_2\|_2(N + kt) + \|M_4\|_2 k)^2),$$
$$\sigma = O(n^{4.5}k(\|M_2\|_F n^3(N + kt) + \|M_4\|_F k)^2).$$

**Proof Sketch:** Because the variable matrices $\Delta\mathbf{C_t}$ is row stochastic, they are bounded in norm. A similar conclusion is true of all constants. Therefore, since the objective function is continuous and differentiable everywhere, it is also $L$-smooth, with bounded norm, and bounded objective value, with all finite constants. Precise details and derivations on the constants $L, F,$ and $\sigma$, along with the proof, are provided in *Appendix R*. The consequence is that SGD indeed converges stably.

The per iteration worst-case computational complexity of the proposed DIGC algorithm at $t^{th}$ time instant is $\mathcal{O}(Nkn + kn^2 + nkd)$.

**Algorithm 1 DIGC Algorithm**

**Step 1: Initialize** $\omega, \zeta, \eta, A_{c0}$
**for** $t = 1, 2, \ldots$ **do**
    **while** convergence criteria not met **do**
        Update $[\mathbf{\Delta C_t}]^\mathbf{n}$ using (11) .
    **end while**
    Compute $C_t$ using Equation (4).
    Compute $A_{c_t} = C_t^\top A_t C_t$.
**end for**

---

**Remark.** To maximize efficiency, DIGC treats prior mappings ($C_{t-1}$) as immutable. Consequently, structural errors from initialization ($C_0$) or previous updates are "locked in" rather than corrected. This motivates the adaptive mechanism (ACNR) in Section 3.2, which is required for error recovery in uncertain or volatile environments.

### 3.2. ACNR Algorithm Development

Consider $f(.) = \|A_{c_t} - A_{c_{t-1}}\|_F^2$, $g(.) = \|\mathbf{\Delta C_{N_t}} X_{sn} - X_n\|_F^2$, $h(.) = \|(C_{t-1} + \mathbf{\Delta C_{S_t}}) - D\mathbf{\Delta C_{N_t}}\|_F^2$, and $\mathcal{T}(.) = \|(C_{t-1} + \mathbf{\Delta C_{S_t}})^\top A_{t-1}^{\text{new}}(C_{t-1} + \mathbf{\Delta C_{S_t}}) - C_{t-1}^\top A_{t-1} C_{t-1}\|_F^2$, the proposed formulation for learning $\mathbf{\Delta C_{N_t}}$ and $\mathbf{\Delta C_{S_t}}$ is:

$$\min_{\mathbf{\Delta C_{N_t}}, \mathbf{\Delta C_{S_t}}} \frac{\omega}{2} \|A_{c_t} - A_{c_{t-1}}\|_F^2 + \frac{\zeta}{2} \|\mathbf{\Delta C_{N_t}} X_{sn} - X_n\|_F^2$$

$$+ \frac{\beta}{2} \|(C_{t-1} + \mathbf{\Delta C_{S_t}}) - D\mathbf{\Delta C_{N_t}}\|_F^2 + \frac{\alpha}{2} \|(C_{t-1} + \mathbf{\Delta C_{S_t}})^\top$$

$$A_{t-1}^{\text{new}}(C_{t-1} + \mathbf{\Delta C_{S_t}}) - C_{t-1}^\top A_{t-1} C_{t-1}\|_F^2 \quad (12)$$

$$\text{s.t.} \quad \mathbf{\Delta C_{N_t}} \geq 0, (C_{t-1} + \mathbf{\Delta C_{S_t}}) \geq 0, \mathbf{\Delta C_{N_t}} \cdot 1 = 1.$$

where $A_{c_t}$, $A_{c_{t-1}}$, $X_{sn}$, and $X_n$ represent the same matrices as defined in Equation 7. The matrix $D \in \mathbb{R}^{N \times k}$ represents the connectivity between the existing nodes and the newly added nodes. Furthermore, at the $t$-th time instant, $A_{t-1}^{\text{new}}$ represents the updated adjacency matrix of the graph that existed at the $(t-1)$-th time instant, in which some edges have changed compared to the previous adjacency matrix $A_{t-1}$. Similar to (9), the coarsened graph adjacency matrix can be represented as:

$$A_{c_t} = \begin{bmatrix} C_{t-1} + \mathbf{\Delta C_{S_t}} \\ \mathbf{\Delta C_{N_t}} \end{bmatrix}^\top \begin{bmatrix} M_1 & M_2 \\ M_3 & M_4 \end{bmatrix} \begin{bmatrix} C_{t-1} + \mathbf{\Delta C_{S_t}} \\ \mathbf{\Delta C_{N_t}} \end{bmatrix}. \quad (13)$$

The difference between the adjacency matrices of the coarsened graphs at time instant t and $t-1$ defined as $\Delta A_c$ is:

$$\Delta A_c = C_{t-1}^\top A_{t-1} \mathbf{\Delta C_{S_t}} + \mathbf{\Delta C_{S_t}}^\top A_{t-1} C_{t-1}$$

$$+ \mathbf{\Delta C_{S_t}}^\top A_{t-1} \mathbf{\Delta C_{S_t}} + \mathbf{\Delta C_{N_t}}^\top M_2^\top C_{t-1}$$

$$+ \mathbf{\Delta C_{N_t}}^\top M_2^\top \mathbf{\Delta C_{S_t}} + C_{t-1}^\top M_2 \mathbf{\Delta C_{N_t}} \quad (14)$$

$$+ \mathbf{\Delta C_{S_t}}^\top M_2 \mathbf{\Delta C_{N_t}} + \mathbf{\Delta C_{N_t}}^\top M_4 \mathbf{\Delta C_{N_t}}.$$

The optimization problem in Equation 12 is a multiblock, non-convex optimization problem. We solved it using the *block coordinate descent* method by relaxing the constraint $\mathbf{\Delta C_{N_t}} \cdot 1 = 1$ to $\|\mathbf{\Delta C_{N_t}} \cdot 1 - 1\|_F^2$. In this approach, we update one variable at a time while keeping the others fixed. We developed an iterative algorithm that follows this strategy and solved it using gradient descent. The update rules for $\mathbf{\Delta C_{S_t}}$ and $\mathbf{\Delta C_{N_t}}$ are:

$$[\mathbf{\Delta C_{S_t}}]^{n+1} \leftarrow \left[[\mathbf{\Delta C_{S_t}}]^n - \eta \nabla f(\mathbf{\Delta C_{S_t}}^n)\right]^+, \quad (15)$$

$$[\mathbf{\Delta C_{N_t}}]^{n+1} \leftarrow \left[[\mathbf{\Delta C_{N_t}}]^n - \eta \nabla f(\mathbf{\Delta C_{N_t}}^n)\right]^+ \quad (16)$$

where $[x]^+ = \max\{x, 0\}$, $\eta$ is the learning rate, and $\nabla_{\mathbf{\Delta C_{S_t}}} \mathcal{L}$ and $\nabla_{\mathbf{\Delta C_{N_t}}} \mathcal{L}$ are the gradients of (15) and (16) with respect to $\mathbf{\Delta C_{S_t}}$ and $\mathbf{\Delta C_{N_t}}$, respectively. More details are provided in the *Appendix* (T).

**Lemma 3.2.** *Consider the optimization problem in* (12), *assume each feature is normalized such that* $\|X_n\|_F \leq k$ *and* $\|X_{sn}\|_F \leq n$. *For any feasible pair* $(\mathbf{\Delta C_{N_t}}, \mathbf{\Delta C_{S_t}})$, *the function* $f(\mathbf{\Delta C})$ *has bounded function constants* $L, F$, *and* $\sigma$, *where* $f$ *is L-smooth, with maximum deviation* $F$, *and maximum gradient norm* $\sigma$.

Further details on the constants $L, F$, and $\sigma$, along with the proof, are provided in Appendix R. Again, the consequence is that SGD converges stably, as is shown in the next theorem.

**Theorem 3.3.** *Stochastic gradient descent(SGD) with a constant step size* $\eta$ *applied to DIGC and ACNR converges at the rate:*

$$\frac{1}{T} \sum_{t=0}^{T} \mathbb{E}[\|\nabla f(\theta^{(t)})\|_2^2] \leq \frac{F}{\eta T} + \frac{L\sigma^2 \eta}{2} \quad (17)$$

*where* $F \geq f(\theta^{(0)}) - \mathbb{E}[f(\theta^{(T)})]$, *the objective function is L-smooth, and* $\sigma$ *upper bounds the stochastic gradient norm.*

---

**Algorithm 2 ACNR Algorithm**

**Step 1: Initialize** $\omega, \zeta, \beta, \alpha, \eta, A_{c0}$
**for** $t = 1, 2, \ldots$ **do**
    **while** convergence criteria not met **do**
        Update $\mathbf{\Delta C_{N_t}}$ and $\mathbf{\Delta C_{S_t}}$ using 15 and 16.
    **end while**
    Compute $C_t$ using Equation (6).
**end for**

---

**Implementation of relaxed assignments.** In both DIGC and ACNR, the discrete assignment constraints are optimized through a relaxed continuous formulation. During optimization, $\mathbf{\Delta C_t}$ and $\mathbf{\Delta C_{N_t}}$ are treated as soft nonnegative

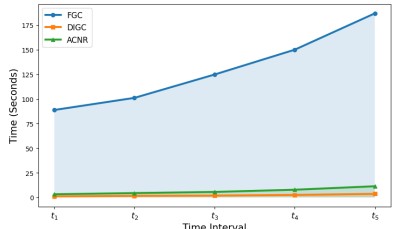
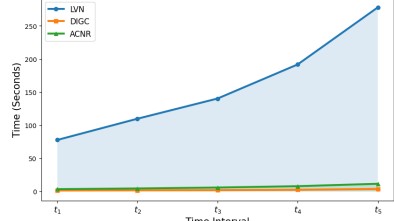
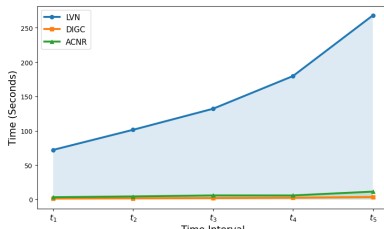

(a) Time to compute the mapping matrix on ACM using FGC for $C_0$, with updates via DIGC and ACNR.

(b) Time to compute the mapping matrix on ACM using LVN for $C_0$, with updates via DIGC and ACNR.

(c) Time to compute the mapping matrix on ACM using LHE for $C_0$, with updates via DIGC and ACNR.

*Figure 2.* Comparison of time taken to compute the mapping matrix $C_t$ over five time intervals for the ACM dataset for r = 0.25. Each subplot shows one initialization method: (a) FGC, (b) LVN, and (c) LHE, followed by updates using DIGC and ACNR.

assignment variables. In ACNR, $\mathbf{\Delta C_{S_t}}$ is not itself a standalone assignment matrix, rather, $C_{t-1} + \mathbf{\Delta C_{S_t}}$ represents the updated assignment block for previously existing nodes. Therefore, after each gradient update in Equations (11), (15), and (16), we project $\mathbf{\Delta C_t}$ and $\mathbf{\Delta C_{N_t}}$ to the nonnegative orthant, and in ACNR project the updated old-node block $C_{t-1} + \mathbf{\Delta C_{S_t}}$ to the nonnegative orthant. The row-stochastic constraint on the new-node assignment block is enforced through the corresponding quadratic penalty, which encourages each newly added node to assign its mass to the supernodes. Before constructing the final coarsened graph at each time step, the relaxed updated assignments are converted to hard assignments by retaining the largest entry in each row and setting the remaining entries to zero. This gives a valid discrete node-to-supernode mapping while avoiding the cost of exact projection onto the combinatorial feasible set at every gradient iteration.

**Lemma 3.4** ($\epsilon$-similarity bound). *Let $\mathcal{G}_t$ be a connected graph at time $t$ with Laplacian $L_t$ satisfying $\lambda_2(L_t) > 0$. Suppose $L_t$ admits the block decomposition*

$$L_t = \begin{bmatrix} L_1 & L_2 \\ L_3 & L_4 \end{bmatrix},$$

*where $L_1$ corresponds to the Laplacian of the graph at time $(t-1)$, $L_4$ is the Laplacian induced by the newly added $k$ nodes at time $t$, and $L_2 = L_3^\top$ encodes the interconnections between old and new nodes. Also $C_t$, $C_{t-1}$ and $\Delta C_t$ are defined as in (4). The corresponding coarsened and lifted Laplacians are defined as*

$$L_{c,t} := C_t^\top L_t C_t \in \mathbb{R}^{n \times n}, L_{\ell,t} := P^\top L_{c,t},$$

*where $P = C_t^\dagger$ as defined in Section 2.2. Then, on the subspace $\{x \in \mathbb{R}^{N+k(t+1)} : x \perp \mathbf{1}\}$, the $\epsilon$-similarity constant in the sense of energy-norm distortion satisfies*

$$\epsilon \leq \frac{\|L_t - L_{\ell,t}\|_2}{\lambda_2(L_t)} \leq \frac{\|L_t - L_{\ell,t}\|_F}{\lambda_2(L_t)}. \tag{18}$$

*Moreover, $\|L_t - L_{\ell,t}\|_F$ admits the explicit bound*

$$\|L_t - L_{\ell,t}\|_F \leq \|L_t\|_F + \frac{1}{\sigma_{\min}(C_{t-1})^2} \Big( \|C_{t-1}\|_2^2 \|L_1\|_F$$
$$+ 2 \|C_{t-1}\|_F \|L_2\|_2 \|\Delta C_t\|_F + \|L_4\|_2 \|\Delta C_t\|_F^2 \Big). \tag{19}$$

For detailed proof, please refer *Appendix S*.

# 4. Experiment

In this section, we demonstrate the efficacy of our proposed DIGC and ACNR algorithms by conducting experiments on real-world datasets with a realistic streaming graph setup. We evaluate our methods on ACM and DBLP, both of which are citation datasets that primarily exhibit monotonic graph growth, and on the AS-733 dataset, a dynamic network topology dataset characterized by non-monotonic evolution. We additionally consider synthetically generated dynamic graph evolutions. See *Appendix G and J.4* for more details on datasets and preprocessing.

## 4.1. Experimental Setup

At the initial time instant $t = 0$, we learn a coarsened graph $A_{c_0}$ by estimating a mapping matrix $C_0$ using a static graph coarsening technique. We employ one of three methods for this initial step: (1) FGC (Kumar et al., 2023b;a), (2) LVN (Variational Neighborhood) (Loukas, 2019; Loukas & Vandergheynst), or (3) LHE (Heavy Edges) (Loukas, 2019). After computing the initial coarsening matrix $C_0$, we evaluate three update strategies at each time step $t$. (1) Recompute $C$ from scratch using the same base method. (2) DIGC incrementally updates $C$ by incorporating the changes. (3) ACNR updates $C$ while also adjusting for structural changes in the graph. Each strategy is applied independently across all three base methods for comparison. Additional results on a real-world Kindle e-book co-purchasing network are provided in Appendix F.

In addition to node additions, our framework handles node deletions and topological changes through an alignment and pruning mechanism. Experiments evaluating non-monotonic evolution, including node deletion and topological realignment, are conducted on the AS-733 dataset and are discussed in *Appendix J*. To assess effectiveness, we evaluate how well the coarsened graphs preserve the structural and spectral characteristics of the original graph over temporal changes while maintaining node classification performance comparable to the original graphs and significantly reducing computational time and memory requirements. We also report an offline full-sequence baseline in Appendix L, which is included as an additional reference point.

## 4.2. REE, RE, and HE

To assess the effectiveness of our proposed methods, DIGC and ACNR, in preserving the properties of the original graph in discrete dynamic settings, we evaluate their performance using three widely adopted metrics: *Relative Eigen Error (REE)* (Loukas & Vandergheynst), *Reconstruction Error (RE)* (Liu et al., 2018), and *Hyperbolic Error (HE)* (Bravo Hermsdorff & Gunderson, 2019).

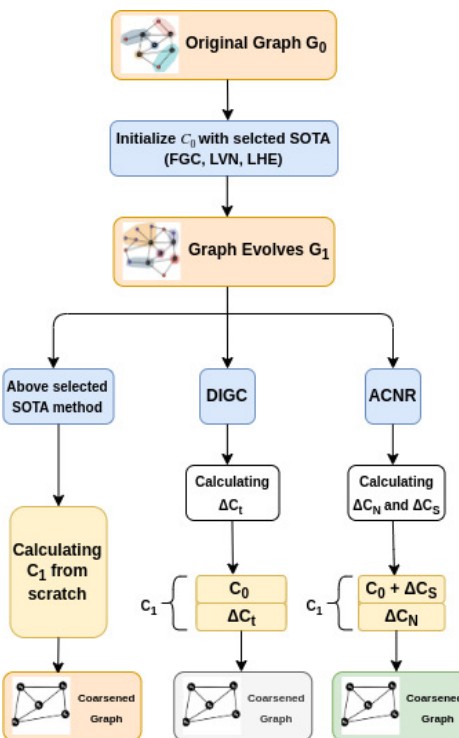

*Figure 3.* Experimental setup for dynamic graph coarsening. Starting with an initial graph $G_0$, a coarsening matrix $C_0$ is obtained using one of three methods: FGC, LVN, or LHE. As the graph evolves to $G_1$, three strategies are compared: (1) re-computing $C$ from scratch using the same method, (2) incrementally updating $C$ via DIGC, and (3) employing ACNR. This process is iterated for each coarsening method.

These metrics were computed at discrete time intervals across the three experimental setups described in Section 4.1. The definitions of REE, RE, and HE are presented in *Appendix H*. As shown in Table 1 and Table 10, our proposed approaches consistently outperform or perform competitively with state-of-the-art methods in the majority of scenarios. This effect is due to temporal regularization, which prevents snapshot-level overfitting and stabilizes spectral structure.

*Table 1.* **Structural Properties and Error Reduction.** We report the average Relative Eigenvalue Error (REE), Reconstruction Error (RE), and Hyperbolic Error (HE) across five time intervals. **Bold** values indicate dynamic updates that either **match or outperform** the static baseline. In nearly all cases, DIGC and ACNR maintain competitive structural integrity, often reducing distortion relative to the full static re-computation through adaptive temporal updates.

| INIT. | METHOD | ACM ↓ | | | DBLP ↓ | | |
|---|---|---|---|---|---|---|---|
| | | REE | RE | HE | REE | RE | HE |
| FGC | DIGC | **0.890** | **4.796** | **2.907** | 0.991 | 2.222 | **2.932** |
| | ACNR | **0.527** | **4.837** | 3.064 | **0.733** | 2.215 | **2.875** |
| | FGC (BASE) | 1.041 | 4.938 | 4.525 | 0.847 | **2.156** | 3.031 |
| LVN | DIGC | 0.627 | **4.849** | 3.118 | **0.494** | **2.206** | 2.819 |
| | ACNR | **0.292** | 4.861 | 3.136 | **0.331** | 2.216 | 2.817 |
| | LVN (BASE) | 0.622 | 4.852 | 2.896 | 0.617 | 2.265 | 3.041 |
| LHE | DIGC | **0.600** | 4.838 | 3.118 | **0.455** | **2.195** | 2.815 |
| | ACNR | **0.248** | 4.851 | 3.110 | **0.361** | 2.204 | 2.819 |
| | LHE (BASE) | 0.763 | 4.908 | 3.327 | 0.679 | 2.284 | 3.033 |

## 4.3. Node Classification

In this section, we evaluate the performance of our proposed methods, DIGC and ACNR, on the task of node classification in dynamically evolving graphs. Implementation details are provided in *Appendix I*.

*Table 2.* **Average Node Classification Accuracy and Performance Preservation.** We report the mean accuracy alongside the standard deviation over all time steps. The Preservation column ($\mathcal{P}$) quantifies the retention of the static baseline performance by our dynamic methods. High preservation values (approaching 100%) indicate that DIGC and ACNR serve as high-fidelity proxies for the full static re-computation, effectively maintaining structural integrity during graph evolution.

| INIT. | METHOD | ACM | DBLP | AVG. $\mathcal{P}$ |
|---|---|---|---|---|
| FGC | DIGC | $85.27^{\pm1.37}$ | $78.36^{\pm2.28}$ | **98.1%** |
| | ACNR | $86.49^{\pm1.28}$ | $79.93^{\pm1.13}$ | **99.8%** |
| | FGC (BASE) | $86.74^{\pm1.25}$ | $80.07^{\pm1.43}$ | 100% |
| LVN | DIGC | $87.66^{\pm1.70}$ | $79.47^{\pm2.57}$ | **96.2%** |
| | ACNR | $89.57^{\pm1.91}$ | $81.78^{\pm1.36}$ | **98.6%** |
| | LVN (BASE) | $91.05^{\pm1.74}$ | $82.75^{\pm1.68}$ | 100% |
| LHE | DIGC | $86.75^{\pm1.57}$ | $80.79^{\pm1.31}$ | **97.3%** |
| | ACNR | $89.37^{\pm1.22}$ | $81.65^{\pm1.53}$ | **99.3%** |
| | LHE (BASE) | $90.21^{\pm1.92}$ | $82.12^{\pm1.06}$ | 100% |

As shown in Table 2, both DIGC and ACNR achieve node classification performance that is competitive with the original graphs and state-of-the-art baselines when averaged over

five discrete time intervals, while preserving the structural and spectral properties of the evolving graphs and significantly reducing computational time and memory requirements. Notably, ACNR more closely matches the performance of the original graphs compared to DIGC, as it adaptively updates the mappings of previously coarsened nodes in response to evolving graph topology, whereas DIGC restricts updates to the mappings of newly added nodes.

### 4.4. Time and Memory Efficiency

As this is the first method addressing dynamic graphs evolving at discrete time instants, we evaluate temporal efficiency over five discrete time steps, $t_1$ to $t_5$. The runtime analysis is presented in Figure 2, and the memory usage comparison is shown in Figure 4. Across all three initialization methods, recomputing the coarsening matrix from scratch becomes increasingly expensive as the graph evolves. In contrast, DIGC and ACNR reuse the previously computed mapping matrix and optimize only the update variables required at the current time step. This leads to substantially lower computation time and peak GPU memory.

The runtime and memory trends also reflect the different design choices of DIGC and ACNR. DIGC is typically the fastest and most memory-efficient because it keeps previous assignments fixed and only learns the incremental update $\Delta \mathbf{C_t}$ for newly added nodes. ACNR is slightly more expensive because it optimizes both $\Delta \mathbf{C_{N_t}}$ and $\Delta \mathbf{C_{S_t}}$, allowing new-node assignment as well as reassignment of previously existing nodes. Nevertheless, ACNR remains significantly more efficient than full static recomputation while providing greater robustness under topology changes.

For completeness, we also report in Appendix O the end-to-end node-classification runtime and memory, including the dynamic coarsening update cost. These results show that the efficiency gains are not restricted to computing the coarsening matrix alone, but also carry over to the downstream learning pipeline.

### 4.5. Ablation, Hyperparameter, and Additional Analysis

We further evaluate the role of individual objective terms, sensitivity to hyperparameters, and behavior under controlled synthetic graph evolution. Appendix M presents experiments on synthetic dynamic datasets, where node and edge updates can be controlled to study different evolution patterns. These experiments complement the real-dataset results and provide additional evidence that DIGC is effective under incremental growth, while ACNR is more suitable when the topology changes more broadly.

We perform an ablation study on Equation 12 by removing terms individually or in pairs to assess their contributions. The results are discussed in Appendix P. The ablation study shows that the temporal smoothness, feature-alignment, reassignment, and topology-refinement terms play complementary roles. In particular, reassignment and topology refinement are important for non-monotonic graph evolution, while feature alignment helps preserve downstream node-classification performance.

Hyperparameter sensitivity is reported in Appendix N. The results show that the proposed methods remain stable across a reasonable range of parameter choices. A theoretical space-complexity comparison of DIGC and ACNR is provided in Appendix Q, which further explains the trade-off between the lower memory cost of DIGC and the additional reassignment capacity of ACNR.

### 4.6. Robustness and Error Recovery

Table 10 shows that DIGC accumulates errors, while ACNR maintains structural properties. This behavior is driven by ACNR's adaptive remapping term $h(\cdot)$, which corrects suboptimal assignments under topological changes. Sensitivity analysis (*Appendix K*) further reveals that DIGC suffers from error lock-in due to poor initialization, whereas ACNR achieves *asymptotic recovery*. Consequently, DIGC is suitable for strictly growing graphs, while ACNR is essential for non-stationary settings.

## 5. Conclusion

We introduced an efficient dynamic graph coarsening framework that incrementally updates the coarsening matrix without recomputation. By addressing both incremental graph growth and evolving topologies through DIGC and ACNR, our approach balances scalability, structural preservation, and adaptability. Extensive experiments demonstrate improved efficiency over static baselines, highlighting the suitability of our method for large scale and real-time graph learning applications.

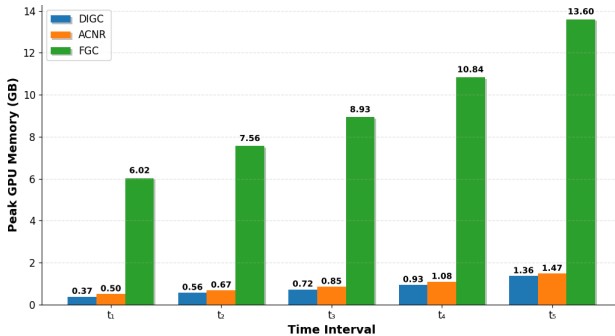

*Figure 4.* Peak GPU memory required (in GB) to compute the mapping matrix $C_t$ across five time intervals on ACM. $C_0$ is initially computed using the state-of-the-art FGC method, and dynamically updated using our proposed DIGC and ACNR methods.

## Impact Statement

This paper introduces a scalable framework for dynamic graph coarsening, enabling efficient adaptation to evolving graph structures. Our work advances graph-based ML and large-scale data processing, with potential applications in social networks, finance, and scientific simulations. While our approach improves scalability and efficiency, we do not foresee any immediate societal concerns requiring attention.

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

# Appendix Overview

# A. Notation Summary

*Table 3.* Summary of the main notation used in the paper.

| Notation | Description |
| --- | --- |
| $G_t = (V_t, E_t, W_t, X_t)$ | Graph snapshot at time $t$, with nodes, edges, edge weights, and node features. |
| $G_{c_t}$ | Coarsened graph corresponding to $G_t$. |
| $N_t = |V_t|$ | Number of active nodes in the original graph at time $t$. |
| $n = |V_{c_t}|$ | Number of supernodes in the coarsened graph. |
| $k_t$ | Number of newly added nodes between times $t-1$ and $t$. |
| $d$ | Node-feature dimension. |
| $A_t, L_t$ | Adjacency and Laplacian matrices of the original graph at time $t$. |
| $A_{c_t}, L_{c_t}$ | Adjacency and Laplacian matrices of the coarsened graph at time $t$. |
| $C_t \in \mathbb{R}_+^{N_t \times n}$ | Coarsening matrix at time $t$ mapping original nodes to supernodes. |
| $C_0$ | Initial coarsening matrix obtained using a static coarsening method. |
| $C_t^\dagger$ | Moore–Penrose pseudoinverse of $C_t$. |
| $\mathbf{\Delta C_t} \in \mathbb{R}_+^{k_t \times n}$ | DIGC update matrix assigning newly added nodes to existing supernodes. |
| $\mathbf{\Delta C_{N_t}} \in \mathbb{R}_+^{k_t \times n}$ | ACNR assignment matrix for newly added nodes. |
| $\mathbf{\Delta C_{S_t}} \in \mathbb{R}_+^{N_{t-1} \times n}$ | ACNR refinement matrix for previously existing nodes. |
| $X_n$ | Feature matrix of newly added nodes. |
| $X_{sn}$ | Feature matrix of existing supernodes. |
| $M_1, M_2, M_3, M_4$ | Block components of $A_t$, where $M_1 = A_{t-1}$, $M_4$ captures edges among new nodes, and $M_2, M_3$ capture edges between old and new nodes. |
| $A_{t-1}^{new}$ | Updated adjacency matrix of the previously existing subgraph after topology change at time $t$. |
| $D$ | Connectivity matrix between previously existing nodes and newly added nodes in ACNR. |
| $f(\cdot, \cdot)$ | Temporal structural consistency term between consecutive coarsened graphs. |
| $g(\cdot)$ | Feature-alignment term for assigning newly added nodes to supernodes. |
| $h(\cdot)$ | Assignment-consistency term used by ACNR to refine mappings of previously existing nodes. |
| $T(\cdot, \cdot)$ | Topology-refinement term for edge changes among previously existing nodes. |
| $\omega, \zeta, \beta, \alpha$ | Hyperparameters controlling structural consistency, feature alignment, reassignment consistency, and topology refinement, respectively. |
| $\eta$ | Learning rate used in the projected gradient updates. |
| $r$ | Coarsening ratio used in experiments. |

# B. Introduction to Dynamic Graphs

Graph data can be broadly categorized into static and dynamic graphs based on whether their structure changes over time (Liang et al., 2024; Mo et al., 2023).Static graphs have fixed nodes and edges, represented by an adjacency matrix that remains constant. Graph coarsening methods for static graphs aim to reduce graph size by aggregating nodes into super-nodes while preserving structural and spectral properties (Kumar et al., 2023b; Chen et al., 2022).

In contrast, dynamic graphs evolve as new nodes and edges are added, removed, or modified. Dynamic graphs are typically divided into two categories: Discrete-Time Dynamic Graphs (DTDGs) and Continuous-Time Dynamic Graphs (CTDGs).

## B.1. Discrete-Time Dynamic Graphs (DTDGs)

*Discrete-Time Dynamic Graphs (DTDGs)* represent evolving networks as a sequence of time-indexed snapshots $\{G_{t_1}, G_{t_2}, \ldots, G_{t_n}\}$, where each snapshot $G_{t_i}$ captures the state of the graph at a specific interval(Gao & Ribeiro, 2021). Between consecutive snapshots, the network may undergo several types of changes, including the addition or removal of nodes (e.g., users joining or leaving a social platform), the insertion or deletion of edges (e.g., formation or dissolution of interactions), and updates to node or edge weights that reflect evolving properties or interaction strengths. These snapshots are naturally ordered in time, providing a structured way to model network evolution without the need to track individual events continuously.

This snapshot-based representation is particularly effective in scenarios where changes occur in batches or can be aggregated over regular intervals, such as social networks with daily or weekly updates, financial transaction networks with periodic reporting, or citation and co-authorship graphs that evolve over months or years. A key advantage of DTDGs is that they allow algorithms designed for static graphs to be applied to each snapshot, while temporal dependencies can be captured across snapshots using sequence models such as recurrent or attention-based architectures (Pareja et al., 2020; Sankar

et al., 2020). However, static coarsening approaches become inefficient in this setting because they require recomputing the coarsened graph from scratch for every update, which is computationally expensive and unsuitable for large-scale or real-time applications (Hashemi et al., 2024b).

By discretizing time, DTDGs strike a balance between temporal resolution and computational efficiency, making them a widely adopted paradigm for modeling large-scale evolving networks.

### B.2. Continuous-Time Dynamic Graphs (CTDGs)

*Continuous-Time Dynamic Graphs (CTDGs)* represent network evolution as a stream of timestamped events rather than discrete snapshots. Each interaction, such as an edge creation, deletion, or weight update, is recorded with precise timing, providing fine-grained temporal resolution. This event-driven representation is well-suited for applications requiring real-time dynamics, such as communication networks or online transaction streams. However, the high temporal resolution of CTDGs comes with increased computational complexity, as streaming updates must be processed efficiently without relying on static graph snapshots.

In this work, we focus primarily on discrete-time dynamic graphs, as their snapshot-based structure naturally supports incremental and adaptive coarsening strategies. This enables scalable and accurate dynamic graph coarsening without the overhead of full recomputation at each time step.

## C. Extended Related Work and Qualitative Comparisons

While the management of evolving graph data has attracted significant attention, existing literature largely bifurcates into model-centric continual learning or static reduction techniques, leaving a specific gap for dynamic structural preservation.

**Continual Graph Learning (CGL) Frameworks.** A substantial body of work addresses continual or adaptive learning on evolving graphs; however, the primary objective of these methods is model adaptation specifically, the mitigation of catastrophic forgetting in neural weights ($\theta$), rather than the incremental maintenance of a coarsening operator ($C$). For instance, E-CGL (Guo et al., 2025) enhances training efficiency by replacing the message-passing encoder with an MLP that shares the GCN's weight space, employing replay-based sampling to accelerate adaptation while preserving accuracy. Similarly, GCAL (Qiao et al., 2025) targets domain adaptation under evolving out-of-distribution (OOD) sequences via a bilevel "adapt-and-generate" strategy, condensing input graphs into latent memory banks to sustain a single model across domain shifts. In distributed settings, POWER (Zhu et al., 2025) addresses federated continual learning via client-side experience replay and server-side pseudo-prototype reconstruction. Recent class-incremental work such as GOTHAM (Shahane et al., 2025) addresses weakly supervised few-shot and zero-shot class increments using prototype-based adaptation and teacher-student distillation, but remains focused on continual node classification rather than maintaining an operator-valid coarsening matrix. Critically, these frameworks focus on optimizing classifier performance across sequential tasks, whereas our framework focuses on sustaining a geometrically valid coarsened representation across structural evolution events.

**Graph Condensation and Replay Mechanisms.** Closely related to coarsening are graph condensation and memory replay strategies, though their objectives differ fundamentally from dynamic spectral approximation. Open-World Graph Condensation (OpenGC) (Gao et al.) frames the challenge as handling distribution shifts caused by continuously arriving nodes. It proposes temporal-invariance condensation to distill datasets into compact, synthetic graphs for efficient training. However, these condensed graphs are synthetic artifacts optimized for gradient matching, lacking the geometric interpretability required of a coarsened Laplacian. TACO (Han et al., 2024) aligns closer to our domain by integrating coarsening into continual learning; it performs timestamp-wise reduction to store representative subgraphs in a memory buffer. Nevertheless, TACO treats coarsening merely as a memory construction mechanism to facilitate experience replay for node classification, rather than as a primary objective to track graph evolution with operator-validity constraints.

**Static Coarsening and Efficiency.** Finally, several distinct efforts improve the efficiency of static coarsening or structure learning but do not constitute dynamic baselines. Faster Graph Embeddings (Fahrbach et al., 2020) accelerates embedding computations by exploiting Schur-complement graphs via approximate sparse Cholesky factorization; this serves as a solver speedup technique rather than a dynamic update method. In the realm of granularity, GBGC (Xia et al., 2025) proposes an adaptive static pipeline that splits graphs from coarse to fine using granular balls as supernodes. While effective for preprocessing, it does not account for temporal updates. Similarly, AH-UGC (Kataria et al., 2025a) employs consistent

hashing to enable resolution adaptivity, removing fixed bin-widths but remains a static snapshot technique that requires re-hashing under topological drift. GraphFLEx (Kataria et al., 2025b), while addressing scalable structure inference in expanding graphs, focuses on learning missing edges (Graph Structure Learning) rather than maintaining a reduced operator. In contrast to these approaches, our DGC framework explicitly learns a coarsened graph at discrete time instants, initializing from any static coarsener at $t = 0$, and subsequently maintaining the coarsening map through dedicated incremental updates under node and edge evolution events.

The works discussed above highlight a distinct gap in the current landscape: existing methods are either algorithmically static (requiring prohibitive re-computation for every update), structurally synthetic (producing non-geometric graph condensations), or objective-orthogonal (optimizing for model memory rather than spectral properties). Consequently, no existing state-of-the-art framework provides a direct, functionally equivalent baseline for the problem of incrementally updating a coarsening operator under discrete-time topological evolution.

## D. Dynamic Baselines

To rigorously validate the efficacy of our proposed DIGC and ACNR frameworks, we compare against dynamic heuristic baselines. These baselines represent the most logical extensions of static coarsening to the dynamic setting: utilizing a stable initial coarsening and greedily assigning new nodes based on local information without re-optimization. We formally define these heuristics, Feature Similarity and Neighborhood Overlap, below.

**Nearest-Super-node Assignment (NSA)**: In the absence of direct competitors for dynamic graph coarsening, we introduce NSA baselines. These heuristics operate on a "lazy update" principle: given a stable coarsening matrix initialized at $t = 0$, they assign newly arriving nodes at time to existing super-nodes via greedy local maximization.

### D.1. Feature Similarity Heuristic (NSA-Feat)

This baseline leverages the homophily hypothesis in feature space, assuming that nodes with similar attributes belong to the same spectral cluster. It operates purely on node content, ignoring the evolving edge structure during the assignment phase. For a newly added node with feature vector , we compute its similarity against the representative centroids of all existing super-nodes, denoted by the matrix . The assignment is determined by maximizing the Cosine Similarity between the new node and the -th super-node:

$$S_k = \frac{x_{new} \cdot (X_{coarse}[k])^T}{|x_{new}| * 2 \cdot |X * coarse[k]|_2} \tag{20}$$

The new node is assigned to the super-node $k^* = \operatorname{argmax}_k(S_k)$.

### D.2. Neighborhood Overlap Heuristic (NSA-Ovlp)

This baseline relies on structural cohesion to preserve graph cut properties. It assumes that a new node should merge into the super-node that contains the majority of its neighbors, thereby minimizing inter-cluster edges. For a newly added node , the algorithm identifies its set of neighbors within the existing graph structure at time . It then aggregate "votes" for each super-node based on the current membership of these neighbors. Let $C_{\text{old}}$ be the binary assignment matrix from the previous timestamp. The overlap score for super-node is calculated as the count of neighbors residing in that super-node:

$$O_k = \sum_{u \in \mathcal{N}(v_{new})} C_{\text{old}}[u, k] \tag{21}$$

This can be efficiently vectorized , where $O_k$ represents the adjacency connections between new and existing nodes. The node is assigned to $k^* = \operatorname{argmax}_k(O_k)$.

A critical theoretical limitation of NSA-Ovlp is the "cold start" scenario, where a new node has no edges connecting to the existing graph (e.g., isolated nodes or cliques of purely new nodes). In such instances where , the method strictly falls back to the NSA-Feat strategy to ensure a valid assignment.

# E. Experimental Results: Comparison with Dynamic Heuristics

To evaluate the effectiveness of our proposed optimization-based frameworks relative to dynamic heuristic baselines, we conduct a comparative study on the ACM and DBLP datasets. The AS-733 dataset is excluded from this analysis, as it involves node deletions along with node additions. The NSA heuristics, Feature Similarity, and Neighborhood Overlap are not designed to handle node deletions and the disappearance of super-node members, and tend to degenerate into unstable or near-random assignments under such non-monotonic evolution.

### E.1. Experimental Setup

We follow the same experimental protocol as described in Section 4.1. At the initial time step ($t = 0$), the input graph is coarsened using FGC(Kumar et al., 2023b) to obtain an initial mapping matrix $C_0$. For subsequent time steps ($t = 1 \ldots 5$), the coarsening matrix $C_t$ is updated using five different strategies. Specifically, we compare our proposed DIGC, which performs incremental optimization-based updates by assigning only newly added nodes while preserving existing mappings, and ACNR, which extends this approach by allowing adaptive reassignment of previously coarsened nodes to correct structural drift. We contrast these methods with two dynamic heuristic baselines: NSA-Feat, which assigns new nodes greedily based on feature similarity, and NSA-Ovlp, which assigns new nodes based on neighborhood overlap. Finally, we include FGC (Base), which recomputes the coarsening matrix from scratch at each time step, as a static upper-bound baseline.

*Table 4.* **Comparison with Dynamic Heuristics.** We report the average Relative Eigenvalue Error (REE), Reconstruction Error (RE), and Hyperbolic Error (HE) across five time intervals for ACM and DBLP. The graph is initialized using FGC at $t = 0$. **Bold** values indicate methods that outperform the static re-computation baseline (FGC).

| INIT. | METHOD | ACM ↓ | | | DBLP ↓ | | |
|---|---|---|---|---|---|---|---|
| | | REE | RE | HE | REE | RE | HE |
| FGC | DIGC (OURS) | **0.890** | **4.796** | **2.907** | 0.991 | 2.222 | **2.932** |
| | ACNR (OURS) | **0.527** | **4.837** | **3.064** | **0.733** | 2.215 | **2.875** |
| | NSA-FEAT | 1.294 | 5.571 | 3.436 | 1.552 | 2.519 | 3.459 |
| | NSA-OVLP | 1.527 | 5.380 | 3.341 | 1.456 | 2.714 | 3.725 |
| | FGC (BASE) | 1.041 | 4.938 | 4.525 | 0.847 | **2.156** | 3.031 |

### E.2. Results Analysis

Table 4 summarizes the quantitative results. Across both datasets and nearly all metrics, the proposed methods, DIGC and ACNR, consistently outperform the heuristic baselines NSA-Feat and NSA-Ovlp. Notably, the heuristic approaches often incur substantially higher error than even the full static re-computation baseline, despite their low computational cost. We further observe that during the initial timestamps, the error discrepancy was relatively contained as the heuristics operated on the recently stabilized coarsening matrix $C_0$. However, as temporal evolution progressed, this gap diverged significantly, demonstrating the cumulative failure of greedy updates to track spectral drift.

Overall, while the NSA heuristics provide computationally inexpensive update rules, they lack a global optimization objective and therefore accumulate structural distortion over time. Our optimization-based frameworks bridge this gap by combining the efficiency of dynamic updates with the structural and spectral fidelity typically associated with static coarsening solvers.

# F. Additional New Dataset Evaluation

### F.1. Kindle E-book Co-purchasing Network

To further evaluate the generality of the proposed dynamic coarsening framework, we additionally consider a real-world Kindle e-book co-purchasing network. In this graph, nodes correspond to Kindle e-books and edges represent co-purchasing relationships between books. This dataset provides an additional real-world evolving graph with semantics different from the citation networks ACM and DBLP and the Internet topology dataset AS-733.

**Pre-processing.** We preprocess the Kindle dataset using the same protocol as the real dynamic datasets described in Appendix G. We first construct a sequence of cumulative graph snapshots at discrete time instants. At each time step, the snapshot contains all nodes and edges observed up to that time. We then extract the largest connected component from each snapshot to ensure that the spectral and reconstruction-based evaluation metrics are well-defined. Node identities are kept consistent across snapshots so that newly added nodes between consecutive time steps can be identified. These newly added nodes are then used to form the incremental update variables for DIGC and ACNR. We use the same evaluation protocol as in the main experiments, including the same coarsening ratio and the same three static initializers: FGC, LVN, and LHE.

## F.2. Results on Kindle

We report node classification accuracy and structural preservation metrics on the Kindle e-book co-purchasing network. All values are averaged over the five temporal transitions.

*Table 5.* Node classification accuracy on the Kindle e-book co-purchasing network. Values are averaged over five temporal transitions.

| Initialization | DIGC | ACNR | BASE |
|---|---|---|---|
| FGC | 55.1 | 55.7 | 56.1 |
| LVN | 54.2 | 54.8 | 55.3 |
| LHE | 54.7 | 56.1 | 56.3 |

*Table 6.* Structural error metrics on the Kindle e-book co-purchasing network. Values are averaged over five temporal transitions. Lower values are better.

| Initialization | Method | REE ↓ | RE ↓ | HE ↓ |
|---|---|---|---|---|
| FGC | DIGC | 0.94 | 9.66 | 4.77 |
| | ACNR | 0.79 | 9.61 | 4.81 |
| | FGC (BASE) | 0.93 | 9.64 | 4.78 |
| LVN | DIGC | 0.82 | 9.51 | 6.33 |
| | ACNR | 0.72 | 9.48 | 6.18 |
| | LVN (BASE) | 0.83 | 9.60 | 6.30 |
| LHE | DIGC | 0.87 | 9.53 | 6.19 |
| | ACNR | 0.70 | 9.50 | 6.12 |
| | LHE (BASE) | 0.82 | 9.58 | 6.12 |

The Kindle results are consistent with the trends observed on the other datasets. ACNR generally gives stronger structural preservation because it can refine previous assignments under graph evolution, while DIGC remains competitive and efficient when the evolution is dominated by incremental growth. The node classification results also show that the proposed dynamic updates maintain downstream performance close to the corresponding static recomputation baseline.

# G. Dataset and Pre-processing

In this section, we describe the datasets utilized for our experiments. To evaluate the proposed approach, we employ both real-world streaming graphs that exhibit natural temporal evolution, as well as synthetic datasets derived from static PyTorch Geometric (PyG) benchmarks, which are transformed into discrete dynamic graphs. Below, we detail each dataset used in our study along with the pre-processing steps applied.

## G.1. Real Dynamic Datasets

To evaluate the effectiveness of our proposed algorithms DIGC and ACNR, we conducted experiments on two real-world dynamic citation networks, ACM and DBLP. We applied the preprocessing steps detailed below to each dataset similar to (Han et al., 2024), and Table 7 summarizes their key statistics.

The **ACM** dataset is a citation network where each node represents a research paper and edges represent citation relationships. Since citations evolve over time, this data set naturally lends itself to modeling as a discrete dynamic graph. Specifically,

we construct cumulative snapshots of the citation network at six discrete time instants: 1995, 1997, 1999, 2001, 2003, and 2005, which we denote as $t_0$ through $t_5$, respectively. At each time instant $t_i$, the graph includes all papers published up to that year, with edges pointing to earlier or contemporaneous papers to represent citations. This temporal construction allows us to track the evolution of the network structure as new nodes (papers) and edges (citations) are added over time. To ensure connectivity and meaningful analysis, we extract the largest connected component at each snapshot. Additionally, we identify and flag the newly added papers at each time step to support incremental evaluation and dynamic coarsening.

The **DBLP** dataset is also a citation network. In this dataset, we select papers published between 1995 and 2000 across four major research fields: Databases, Data Mining, Artificial Intelligence, and Computer Vision. Each paper is assigned a class label based on its publication venue. To construct node features, we extract keywords and fields of study that appear at least 500 times in the dataset and represent each paper as a binary feature vector based on these attributes. At each discrete time instant $t_i$, the graph includes all papers published up to that year, with directed edges representing citation links to earlier or contemporaneous papers. Similar to the ACM dataset, we extract the largest connected component from each snapshot and mark newly introduced papers at each time step to support incremental evaluation and dynamic analysis. During our experiments, we apply a coarsening ratio of **r = 0.25** for both the datasets.

*Table 7.* **Dynamic graph statistics across time.** The table reports the number of nodes and edges for the ACM and DBLP datasets across six discrete time steps, illustrating the temporal expansion of the underlying graphs.

| TIMESTEP | ACM | | DBLP | |
|---|---|---|---|---|
| | NODES | EDGES | NODES | EDGES |
| $t_0$ | 3237 | 19790 | 6548 | 13057 |
| $t_1$ | 4192 | 26212 | 7208 | 14402 |
| $t_2$ | 5229 | 34066 | 7761 | 15680 |
| $t_3$ | 6147 | 41786 | 8644 | 17705 |
| $t_4$ | 7414 | 52838 | 9728 | 20258 |
| $t_5$ | 9219 | 67786 | 10997 | 23382 |

## G.2. Synthetic Dynamic Graph Datasets

To simulate temporal evolution in static benchmark datasets, we transform PyTorch Geometric datasets, specifically **CORA**, **Coauthor-CS**, and **PUBMED**, into sequences of discrete graph snapshots. These transformed datasets are denoted with the prefix "D-" (e.g., **D-CORA**, **D-PUBMED**, **D-Co-CS**) to distinguish them from real temporal datasets. Specifically, we begin by extracting a subgraph containing approximately 50% of the nodes for CORA, and approximately 30% of the nodes for the remaining two datasets. At each subsequent time step, we unhide additional nodes to emulate temporal growth: 100 nodes per iteration for CORA, and 500 nodes per iteration for Coauthor-CS and PubMed.

In each snapshot, only the edges among the currently visible nodes are retained. During our experiments, we apply a coarsening ratio of **r = 0.3** for the D-CORA dataset and **r = 0.05** for the D-Coauthor-CS and D-PUBMED datasets, relative to the size of their original graphs. Additionally, we introduce perturbation step by randomly flipping a percentage of ones to zeros in the adjacency matrix, simulating edge removals or structural noise. All computations are executed with a fixed random seed ensuring reproducibility. The statistics of the synthetic datasets are summarized in Table 8.

*Table 8.* **Node statistics of synthetic dynamic graph datasets.** The table illustrates the number of visible nodes at each discrete time step ($t_0$ to $t_5$) for three synthetic dynamic datasets—D-CORA, D-Co-CS, and D-PUBMED. The initial snapshot ($t_0$) contains approximately 50% of the total nodes for D-CORA and 30% for D-Co-CS and D-PUBMED, with additional nodes progressively revealed at each subsequent step.

| | DATASET | | |
|---|---|---|---|
| | **D-CORA** | **D-Co-CS** | **D-PUBMED** |
| $t_0$ | 1354 | 5500 | 5915 |
| $t_1$ | 1454 | 6000 | 6415 |
| $t_2$ | 1554 | 6500 | 6915 |
| $t_3$ | 1654 | 7000 | 7415 |
| $t_4$ | 1754 | 7500 | 7915 |
| $t_5$ | 1854 | 8000 | 8415 |

## H. Error Metrics

To evaluate the quality of graph coarsening and the ability of our methods to preserve structural information, we employ three widely-used error metrics: *Relative Eigen Error (REE)*, *Reconstruction Error (RE)*, and *Hyperbolic Error (HE)*. These metrics provide complementary insights into the spectral and structural properties between the original and coarsened graphs.

**1. Relative Eigen Error (REE)** (Loukas & Vandergheynst): REE measures the relative discrepancy between the top $k$ eigenvalues of the original graph Laplacian $L$ and the coarsened graph Laplacian $L_c$. It is defined as:

$$\text{REE} = \frac{1}{k} \sum_{i=1}^{k} \frac{|\lambda_{c,i} - \lambda_i|}{\lambda_i},$$

where $\lambda_i$ and $\lambda_{c,i}$ are the $i$-th largest eigenvalues of $L$ and $L_c$, respectively. In our experiments, we set $k = 100$.

**2. Reconstruction Error (RE)** (Liu et al., 2018): RE quantifies the Frobenius norm difference between the original Laplacian $L$ and the reconstructed (or lifted) Laplacian $L_l$, which is obtained by projecting the coarsened Laplacian back to the original space using a projection matrix $P$. The error is given by:

$$\text{RE} = \|L - L_l\|_F^2, \quad \text{where } L_l = P^\top L_c P.$$

This metric captures the information loss incurred during the coarsening and reconstruction process.

**3. Hyperbolic Error (HE)** (Bravo Hermsdorff & Gunderson, 2019): HE evaluates the structural distortion in hyperbolic space between the original and lifted Laplacians. It is defined as:

$$\text{HE} = \text{arccosh}\left(1 + \frac{\|(L - L_l)X\|_F^2 \cdot \|X\|_F^2}{2 \cdot \text{tr}(X^\top L X) \cdot \text{tr}(X^\top L_l X)}\right),$$

where $X$ is the node feature matrix of the original graph. This metric captures spectral distortions in a hyperbolic geometry, thereby assessing how well the coarsened graph retains the original structural characteristics.

## I. Node Classification

We evaluate node classification performance on dynamically evolving graphs using a two-layer Graph Convolutional Network (GCN). At each time step $t$, the graph is processed through a learned coarsening matrix $C_t$. When coarsening is employed, the assignment matrix is discretized by projecting each row of $C_t$ to its maximum entry, yielding a hard clustering that defines the coarsened graph Laplacian $L_c$.

The GCN architecture consists of two graph convolutional layers, with 64 hidden units in the first layer and an output layer whose dimensionality equals the number of target classes. Model parameters are optimized using the Adam optimizer with a learning rate of $0.01$ and weight decay of $10^{-4}$. Training is conducted for up to $100$ epochs per time step, with early stopping applied based on validation accuracy using a patience of $20$ epochs.

For node classification, we adopt a random 80/10/10 split of nodes into training, validation, and test sets at each time step. Labels corresponding to validation and test nodes are masked during coarse-label construction to prevent information leakage. The GCN is trained on the coarsened graph using coarse labels derived via the pseudoinverse of the assignment matrix, while validation and final evaluation are always performed on the original fine-grained graph. Validation accuracy on the fine graph is monitored during training and used for early stopping. After convergence, test accuracy is computed on the held-out test nodes of the original graph.

To ensure a fair comparison and robust generalization, the hyperparameters are tuned using Tree-Structured Parzen Estimator (TPE) Bayesian optimization, implemented via Hyperopt (Bergstra et al., 2011). To account for stochasticity in data splits and optimization, all reported results are averaged over five independent runs with different random seeds, where each seed controls data splitting and model initialization. Additional details on the hyperparameter search space are provided in *Appendix N*.

# J. Handling Node Deletions and Topological Alignment

While the preceding discussions primarily addressed Incremental Growth and Evolving Topology, real-world dynamic graphs frequently exhibit node deletions. Ignoring deleted nodes results in ghost rows within the coarsening matrix $C$ corresponding to graph elements that no longer exist and which would lead to dimensional mismatches during matrix multiplication and erroneous gradient computations. To address this, we introduce a *Graph Alignment and Pruning* mechanism. This procedure operates as a pre-processing step for each timestamp prior to the core DIGC or ACNR updates, ensuring that the coarsening matrix $C_{t-1}$ is strictly aligned with the active vertex set at time $t$.

## J.1. Set-Theoretic Partitioning and Alignment

To mathematically formalize the alignment process, let $\mathcal{V}_{t-1}$ denote the set of vertices existing at time $t-1$, and $\mathcal{V}_t$ denote the set of vertices at time $t$. We partition the evolving vertex space into three disjoint sets. First, we identify the *Preserved Nodes*, defined as the intersection $\mathcal{V}_{keep}$, representing vertices present in both timestamps. Second, we define the *Deleted Nodes* as $\mathcal{V}_{drop}$, which corresponds to vertices removed at time $t-1$. Finally, the *New Nodes* are defined as $\mathcal{V}_{new}$, representing additions to the graph.

$$\mathcal{V}_{\text{keep}} = \mathcal{V}_{t-1} \cap \mathcal{V}_t \qquad \mathcal{V}_{\text{drop}} = \mathcal{V}_{t-1} \setminus \mathcal{V}_t \qquad \mathcal{V}_{\text{new}} = \mathcal{V}_t \setminus \mathcal{V}_{t-1}$$

The coarsening matrix from the previous time step, $C_{t-1} \in \mathbb{R}^{|\mathcal{V}_{t-1}| \times n}$, maps the nodes in $\mathcal{V}_{t-1}$ to $n$ super-nodes. To account for deletions, the rows corresponding to $\mathcal{V}_{drop}$ in $C_{t-1}$ must be removed. In order to do that, we define a Selection Matrix $S \in \{0,1\}^{|\mathcal{V}_{keep}| \times |\mathcal{V}_{t-1}|}$ to perform this row-slicing operation, where the entry $S_{ij} = 1$ if and only if the $i$-th node in the sorted set $\mathcal{V}_{keep}$ corresponds to the $j$-th node in $\mathcal{V}_{t-1}$. The *Aligned Coarsening Matrix* $\tilde{C}_{t-1}$ is subsequently computed as:

$$\tilde{C}_{t-1} = S \cdot C_{t-1} \tag{22}$$

This operation gives a matrix $\tilde{C}_{t-1} \in \mathbb{R}^{|\mathcal{V}_{keep}| \times k}$ that accurately reflects the current nodes while retaining the learned super-node assignments for surviving vertices.

## J.2. Super-node Space Pruning

A secondary consequence of node deletion is the potential emergence of *Empty Super-nodes*. If the set of nodes mapping to a specific super-node $j$ is a subset of $\mathcal{V}_{drop}$, the $j$-th column of $\tilde{C}_{t-1}$ will become a zero vector. To maintain structural integrity and prevent rank deficiency in the subsequent optimization objectives, we apply column pruning. Let $c_j$ denote the $j$-th column of $\tilde{C}_{t-1}$. We define the set of active super-nodes $\mathcal{J}_{active}$ as

$$\mathcal{J}_{active} = \{j \mid \|c_j\|_1 > 0\},$$

By constructing a column-selector matrix $L \in \{0,1\}^{k \times |\mathcal{J}_{active}|}$, we update the coarsening matrix $C_t$ to:

$$C'_{t-1} = \tilde{C}_{t-1} \cdot P \tag{23}$$

Consequently, the effective dimensionality of the super-node space is reduced from $n$ to $n' = |\mathcal{J}_{active}|$. This pruned matrix $C'_{t-1}$ serves as the consistent initialization for the subsequent optimization steps.

## J.3. Optimization Consistency and Integration

The alignment and pruning process guarantees that the dimensions of the matrices in our optimization functions (Eq. 7 and Eq. 12) remain consistent with the evolving topology. In the context of **DIGC**, the matrix $C'_{t-1}$ serves as the immutable block for existing nodes, and the algorithm solves strictly for $\mathbf{\Delta C_t} \in \mathbb{R}^{|\mathcal{V}_{new}| \times n'}$ to map newly added nodes to the surviving super-nodes. Conversely, in the **ACNR** framework, $C'_{t-1}$ acts as a reference point. The algorithm learns an adjustment matrix $\mathbf{\Delta C_{S_t}} \in \mathbb{R}^{|\mathcal{V}_{keep}| \times k'}$ to reassign surviving nodes which is a critical capability when deletions significantly alter local connectivity, alongside the mapping $\mathbf{\Delta C_{N_t}}$ for new nodes.

---

**Algorithm 3 Dynamic Alignment and Pruning Strategy**

---

**Require:** Previous coarsening matrix $C_{t-1}$, Node sets $\mathcal{V}_{t-1}, \mathcal{V}_t$

  **Set Identification:**
  $\mathcal{V}_{keep} \leftarrow \mathcal{V}_{t-1} \cap \mathcal{V}_t$
  $\mathcal{V}_{new} \leftarrow \mathcal{V}_t \setminus \mathcal{V}_{t-1}$
  **Row Alignment:**
  Construct mapping indices for $\mathcal{V}_{keep}$.
  $\tilde{C}_{t-1} \leftarrow C_{t-1}[\text{indices}(\mathcal{V}_{keep}), :]$
  **Column Pruning:**
  Compute column sums: $s \leftarrow \sum_{rows} \tilde{C}_{t-1}$
  Identify non-zero columns: $\text{mask} \leftarrow (s > \epsilon)$
  $C'_{t-1} \leftarrow \tilde{C}_{t-1}[:, \text{mask}]$ **return** Aligned matrix $C'_{t-1}$ and new node set $\mathcal{V}_{new}$

---

## J.4. Dataset and Pre-processing

To rigorously evaluate the efficacy of our framework on a discrete dynamic graph exhibiting simultaneous additions and deletions, we utilized the Autonomous Systems (AS-733) dataset. This dataset records the daily evolution of peering relationships between autonomous systems on the Internet. To construct a suitable discrete temporal sequence, we selected ten snapshots from the raw daily instances.

### J.4.1. TEMPORAL FORMATTING AND GLOBAL INDEXING

A critical challenge in processing dynamic graphs with node deletions is ensuring consistent identity tracking across non-consecutive snapshots. We implemented a global indexing scheme that maps every unique node identifier appearing across the entire timeline to a fixed global index. This mechanism allows for the precise tracking of node lifecycles, specifically distinguishing between *preserved*, *added*, and *deleted* nodes between any two consecutive timestamps $t$ and $t+1$.

### J.4.2. STRUCTURAL FEATURE GENERATION

Given that the AS-733 dataset consists of purely topological data without any explicit node attributes, we generated synthetic structural features to initialize the learning framework. We adopted a random walk-based embedding strategy designed to capture latent structural roles and community contexts. Specifically, we first computed Pointwise Mutual Information (PMI) matrices derived from node co-occurrence frequencies observed within random walks sampled from the graph. We then applied Singular Value Decomposition (SVD) to these matrices to extract dense, 128-dimensional feature vectors for each node. This effectively encodes the structural equivalence and local connectivity patterns of the vertices, providing a rich feature representation derived solely from the graph topology. Finally, the evolution of the graph over the ten snapshots is summarized in Table 9. The data highlights the dynamic nature of the network, with significant variations observed at each interval. During our experiments, we apply a coarsening ratio of **r = 0.25** for AS-733.

*Table 9.* **Temporal evolution of graph size and node dynamics in AS-733.** The table summarizes ten discrete time snapshots, reporting the number of active nodes and edges, along with nodes added to and deleted from the graph relative to the preceding time step.

| TIME STEP | ACTIVE NODES | EDGES | NODES ADDED | NODES DELETED |
|---|---|---|---|---|
| $t_0$ | 3,015 | 10,695 | 3,015 | 0 |
| $t_1$ | 3,264 | 11,922 | 337 | 88 |
| $t_2$ | 3,537 | 13,284 | 404 | 131 |
| $t_3$ | 3,757 | 14,314 | 347 | 127 |
| $t_4$ | 4,055 | 15,554 | 440 | 142 |
| $t_5$ | 4,346 | 16,674 | 399 | 108 |
| $t_6$ | 4,587 | 17,978 | 368 | 127 |
| $t_7$ | 4,943 | 19,892 | 468 | 112 |
| $t_8$ | 5,354 | 21,742 | 522 | 111 |
| $t_9$ | 5,738 | 23,335 | 530 | 146 |

## J.5. Experimental Validation on Node Deletions

To empirically validate the effectiveness of our *Alignment and Pruning* mechanism, we evaluated the spectral and structural properties of the coarsened graphs on the AS-733 dataset. We compare our proposed incremental strategies, DIGC and ACNR against the static baseline where the coarsening matrix is re-computed from scratch using FGC at each timestamp.

### J.5.1. STRUCTURAL PRESERVATION METRICS

We utilized the three standard error metrics defined previously in *Appendix H*: Relative Eigen Error (REE), Reconstruction Error (RE), and Hyperbolic Error (HE). The results for time steps $t_1$ through $t_9$ are detailed in Table 10. Based on these results, we derive the following key observations regarding the handling of node deletions along with node additions and edge modifications.

**Observation 1: Adaptive refinement preserves structural and geometric properties under node deletions.** The results in Table 10 show that ACNR consistently achieves a lower error than both the static baseline (FGC) and the rigid incremental method (DIGC) for all three structural metrics. This improvement is most pronounced in terms of Hyperbolic Error (HE), where ACNR maintains a consistently low and stable distortion throughout the evolution process, while both FGC and DIGC exhibit substantially higher geometric distortion under node deletions. These trends indicate that ACNR's alignment and pruning mechanism effectively mitigates the structural disruptions introduced by node additions and deletions, whereas static re-computation and rigid incremental updates lack the ability to recover from such non-monotonic changes.

**Observation 2: Adaptive reassignment enables long-term robustness under topological volatility.** A clear distinction in temporal stability emerges when examining error trajectories across time. The static FGC baseline exhibits pronounced variability across snapshots, reflecting the difficulty of independently re-optimizing the coarsening objective as the graph undergoes non-monotonic evolution. DIGC demonstrates improved stability relative to the static baseline; however, its reconstruction and geometric errors show a gradual degradation over time, revealing the cumulative effect of locking prior node assignments. In contrast, ACNR maintains the most stable error trajectory across all timestamps, confirming that adaptive reassignment of previously coarsened nodes is essential for sustaining long-term structural properties in the presence of repeated node deletions and evolving connectivity.

These results highlight the necessity of adaptive reassignment and pruning mechanisms for robust dynamic coarsening in non-monotonic graph evolution settings.

*Table 10.* **Structural preservation under dynamic node additions and deletions.** The table reports Relative Eigen Error (REE), Reconstruction Error (RE), and Hyperbolic Error (HE) across time steps $t_1$–$t_9$ on the AS-733 dataset. **Bold** values indicate dynamic updates that either match or outperform the static baseline. Results compare a static baseline (FGC), a rigid incremental method (DIGC), and the proposed adaptive approach (ACNR). All experiments are conducted with a coarsening ratio $r = 0.25$, where the initial coarsening matrix $C_0$ is initialized using FGC. Lower values indicate better structural accuracy.

| METRIC | METHOD | $t_1$ | $t_2$ | $t_3$ | $t_4$ | $t_5$ | $t_6$ | $t_7$ | $t_8$ | $t_9$ |
|---|---|---|---|---|---|---|---|---|---|---|
| REE ($\downarrow$) | DIGC | **0.953** | **0.946** | **0.941** | 0.940 | **0.934** | **0.933** | **0.931** | **0.927** | **0.925** |
| | ACNR | **0.795** | **0.685** | **0.585** | **0.513** | **0.461** | **0.445** | **0.470** | **0.515** | **0.591** |
| | FGC (BASE) | 0.996 | 0.998 | 1.001 | 0.797 | 0.998 | 0.984 | 0.979 | 0.984 | 0.979 |
| RE ($\downarrow$) | DIGC | **5.747** | **5.858** | **5.963** | **6.010** | **6.102** | 6.190 | 6.252 | 6.286 | 6.345 |
| | ACNR | **4.705** | **4.640** | **4.696** | **4.754** | **4.849** | **4.952** | **5.070** | **5.148** | **5.236** |
| | FGC (BASE) | 5.847 | 5.957 | 6.035 | 6.021 | 6.127 | 5.892 | 6.205 | 6.133 | 6.341 |
| HE ($\downarrow$) | DIGC | **5.074** | **5.338** | **5.508** | **5.660** | **5.848** | 5.963 | 6.106 | 6.261 | 6.448 |
| | ACNR | **3.643** | **3.558** | **3.604** | **3.596** | **3.728** | **3.791** | **3.792** | **3.882** | **3.915** |
| | FGC (BASE) | 6.831 | 8.304 | 7.448 | 5.753 | 6.799 | 5.460 | 5.557 | 5.421 | 5.331 |

# K. Sensitivity Analysis: Initialization Quality and Error Recovery

A key challenge in dynamic graph coarsening is sensitivity to the quality of the initial coarsened graph. Since our proposed framework initializes from a static coarsening method at $t_0$, the resulting mapping matrix $C_0$ may be structurally suboptimal in practice due to noisy early snapshots, limited historical context, or the use of fast but approximate static coarsening algorithms. In such cases, imperfections in $C_0$ can introduce spectral and structural distortions that may persist as the graph

evolves. This raises an important question specific to dynamic coarsening:

*Can incremental coarsening strategies recover from a poor initial mapping, or do structural errors introduced at initialization propagate indefinitely over time?*

To evaluate robustness to initialization quality, we conduct experiments on the AS-733 dataset under an intentionally degraded initialization, where the initial coarsening matrix $C_0$ is sub-optimally initialized. We then track the evolution of structural properties over time using Relative Eigen Error (REE), Reconstruction Error (RE), and Hyperbolic Error (HE). Table 11 compares the behavior of the rigid incremental strategy (DIGC) and the adaptive reassignment approach (ACNR) under both standard and poor initialization settings.

**Observation 1: Rigid incremental updates lead to error lock-in (DIGC).**   Under degraded initialization, DIGC exhibits a pronounced error lock-in effect. While DIGC performs reasonably when initialized with a high-quality coarsening map, its structural errors remain largely unchanged when starting from a deficient $C_0$. This behavior is a direct consequence of DIGC's design, which freezes the mappings of previously coarsened nodes to maximize computational efficiency. As a result, structural deficiencies present at initialization cannot be corrected, causing spectral and geometric distortions to persist throughout the evolution process.

**Observation 2: Adaptive reassignment enables asymptotic error recovery (ACNR).** In contrast, ACNR demonstrates strong resilience to poor initialization. Despite starting from a structurally deficient mapping, ACNR progressively improves structural properties over time across all evaluated metrics. This recovery behavior stems from the adaptive reassignment term in the ACNR objective, which allows previously coarsened nodes to be remapped in response to evolving topology. Consequently, ACNR is able to correct early structural errors and escape suboptimal initial configurations, exhibiting a clear asymptotic recovery trend.

These results show that adaptive reassignment is essential for robust dynamic graph coarsening, enabling recovery from poor initialization that rigid incremental strategies cannot achieve.

*Table 11.* **Sensitivity to initialization quality under dynamic graph evolution.** The table reports Relative Eigen Error (REE), Reconstruction Error (RE), and Hyperbolic Error (HE) across time steps $t_1$–$t_9$ on the AS-733 dataset under *standard* and *poor* initialization settings. Results compare the rigid incremental update strategy (DIGC) and the proposed adaptive reassignment method (ACNR). All experiments are conducted with a coarsening ratio $r = 0.25$ and $C_0$ is initialized using FGC. **Bold** values indicate the lowest error achieved at each time step under the same initialization setting.

| METRIC | INIT. | METHOD | TIME STEP | | | | | | | | |
|---|---|---|---|---|---|---|---|---|---|---|---|
| | | | $t_1$ | $t_2$ | $t_3$ | $t_4$ | $t_5$ | $t_6$ | $t_7$ | $t_8$ | $t_9$ |
| REE ($\downarrow$) | STANDARD | DIGC | 0.953 | 0.946 | 0.941 | 0.940 | 0.934 | 0.933 | 0.931 | 0.927 | 0.925 |
| | | ACNR | **0.795** | **0.685** | **0.585** | **0.513** | **0.461** | **0.445** | **0.470** | **0.515** | **0.591** |
| | POOR | DIGC | 0.982 | 0.983 | 0.979 | 0.981 | 0.976 | 0.975 | 0.972 | 0.967 | 0.963 |
| | | ACNR | **0.907** | **0.839** | **0.776** | **0.719** | **0.666** | **0.653** | **0.658** | **0.684** | **0.718** |
| RE ($\downarrow$) | STANDARD | DIGC | 5.747 | 5.858 | 5.963 | 6.010 | 6.102 | 6.190 | 6.252 | 6.286 | 6.345 |
| | | ACNR | **4.705** | **4.640** | **4.696** | **4.754** | **4.849** | **4.952** | **5.070** | **5.148** | **5.236** |
| | POOR | DIGC | 6.072 | 6.153 | 6.234 | 6.239 | 6.326 | 6.503 | 6.562 | 6.696 | 6.353 |
| | | ACNR | **5.310** | **5.225** | **5.113** | **5.134** | **5.173** | **5.228** | **5.279** | **5.342** | **5.405** |
| HE ($\downarrow$) | STANDARD | DIGC | 5.074 | 5.338 | 5.508 | 5.660 | 5.848 | 5.963 | 6.106 | 6.261 | 6.448 |
| | | ACNR | **3.643** | **3.558** | **3.604** | **3.596** | **3.728** | **3.791** | **3.792** | **3.882** | **3.915** |
| | POOR | DIGC | 5.452 | 5.504 | 5.706 | 5.784 | 6.062 | 6.186 | 6.434 | 6.621 | 6.803 |
| | | ACNR | **4.451** | **4.315** | **4.167** | **4.115** | **4.190** | **4.179** | **4.137** | **4.187** | **4.215** |

## L. Offline Full-Sequence Baseline

In the main experiments, the static baseline, denoted as BASE, recomputes the coarsening matrix independently at each graph snapshot using the corresponding static coarsening method. In response to the offline full-sequence comparison, we additionally evaluate a stronger baseline, denoted as Base-Full, that has access to the full temporal evolution. Specifically,

*Table 12.* Comparison with the offline full-sequence baseline on ACM and DBLP. DIGC and ACNR operate in the online dynamic-update setting. BASE recomputes the static coarsening independently at each snapshot. Base-Full has access to the full temporal evolution and is included as an offline reference. Lower values are better.

| Initialization | Method | ACM ↓ | | | DBLP ↓ | | |
|---|---|---|---|---|---|---|---|
| | | REE | RE | HE | REE | RE | HE |
| FGC | DIGC | 0.890 | 4.796 | 2.907 | 0.991 | 2.222 | 2.932 |
| | ACNR | 0.527 | 4.837 | 3.064 | 0.733 | 2.215 | 2.875 |
| | BASE | 1.041 | 4.938 | 4.525 | 0.847 | 2.156 | 3.031 |
| | Base-Full | 0.452 | 4.615 | 2.786 | 0.722 | 2.142 | 2.804 |
| LVN | DIGC | 0.627 | 4.849 | 3.118 | 0.494 | 2.206 | 2.819 |
| | ACNR | 0.292 | 4.861 | 3.136 | 0.331 | 2.216 | 2.817 |
| | BASE | 0.622 | 4.852 | 2.896 | 0.617 | 2.265 | 3.041 |
| | Base-Full | 0.285 | 4.557 | 2.816 | 0.285 | 2.028 | 2.785 |
| LHE | DIGC | 0.600 | 4.838 | 3.118 | 0.455 | 2.195 | 2.815 |
| | ACNR | 0.248 | 4.851 | 3.110 | 0.361 | 2.204 | 2.819 |
| | BASE | 0.763 | 4.908 | 3.327 | 0.679 | 2.284 | 3.033 |
| | Base-Full | 0.237 | 4.752 | 3.091 | 0.364 | 2.099 | 2.807 |

Base-Full applies the corresponding static coarsening method using the full temporal graph sequence, and is then evaluated using the same structural metrics and averaging protocol as Table 1.

This baseline is not an online dynamic method and is therefore not directly comparable to DIGC and ACNR in terms of computational setting. Instead, it provides an additional reference point for understanding how close the online dynamic updates are to a stronger offline coarsening baseline. Table 12 compares DIGC, ACNR, the original per-snapshot BASE, and Base-Full on ACM and DBLP.

The results show that Base-Full is generally stronger than the original per-snapshot BASE, as expected, since it has access to the full temporal evolution. However, this comparison also highlights the different computational settings. Base-Full is an offline reference, whereas DIGC and ACNR update the coarsening matrix incrementally from one snapshot to the next. Despite this online constraint, ACNR remains close to Base-Full in several cases and even slightly improves over it on some metrics, such as REE for LHE on DBLP. This suggests that adaptive reassignment can recover much of the structural quality of a stronger offline baseline while retaining the online dynamic-update setting.

## M. Experiments on Synthetic Dynamic Datasets

In this section, we evaluate the effectiveness of our proposed algorithms, DIGC and ACNR, in comparison to state-of-the-art graph coarsening methods. The evaluation is conducted on our synthetically generated discrete dynamic graph datasets: D-CORA, D-Coauthor-CS, and D-PUBMED.

To validate the performance of our methods, we carried out a series of experiments analogous to those described for the real-world datasets. The experimental procedures and evaluation metrics are consistent across both real and synthetic settings, ensuring a fair and comprehensive comparison. The specific experimental results on the synthetic datasets are presented below.

### M.1. Error Metrics Comparison

In this section, we evaluate the average REE(Loukas & Vandergheynst), RE (Liu et al., 2018) and HE(Bravo Hermsdorff & Gunderson, 2019) for each method. We compare three settings: one where the coarsening matrix $C_t$ is recomputed from scratch at each time interval, and another where $C_t$ is dynamically updated from $C_{t-1}$ using our proposed DIGC and ACNR approaches over five time intervals.

As shown in Table 13, the DIGC and ACNR methods achieve the lowest average error values across most cases, demonstrating superior performance compared to state-of-the-art methods that recompute the coarsening matrix independently at each step.

*Table 13.* **Structural Fidelity across Dynamic Citation Networks.** We report the mean Relative Eigenvalue Error (REE), Reconstruction Error (RE), and Hyperbolic Error (HE) over five time intervals. **Bold** values indicate dynamic methods that either **match or outperform** the static baseline. The consistent bolding across ACNR and DIGC rows demonstrates that our framework provides high-fidelity coarsening that is competitive with, and often superior to, full static re-computation.

| INIT. | METHOD | D-CORA ↓ | | | D-CO-CS ↓ | | | D-PUBMED ↓ | | |
|---|---|---|---|---|---|---|---|---|---|---|
| | | REE | RE | HE | REE | RE | HE | REE | RE | HE |
| FGC | DIGC | 0.8569 | 2.5663 | 4.2047 | **0.9000** | 3.198 | 3.148 | 1.353 | 2.118 | 5.608 |
| | ACNR | **0.8200** | **2.5658** | 4.2047 | **0.9010** | 3.184 | **3.041** | **1.060** | 1.923 | **2.891** |
| | FGC (BASE) | 0.8209 | 2.5662 | 4.1142 | 0.9155 | 3.174 | 3.054 | 1.075 | 1.132 | 3.105 |
| LVN | DIGC | 0.856 | 2.566 | 4.281 | **0.870** | **3.840** | **5.197** | 0.904 | 2.678 | 4.736 |
| | ACNR | **0.820** | **2.562** | 4.204 | **0.916** | **3.838** | **5.007** | **0.895** | **2.679** | **4.650** |
| | LVN (BASE) | 0.820 | 2.566 | 4.114 | 0.999 | 3.843 | 7.387 | 0.997 | 2.687 | 6.874 |
| LHE | DIGC | **0.721** | **2.538** | **3.440** | 1.020 | 4.263 | 4.514 | 0.875 | 3.132 | 4.336 |
| | ACNR | **0.647** | **2.543** | **3.422** | 0.844 | **4.254** | 4.426 | **0.824** | **3.115** | **3.841** |
| | LHE (BASE) | 0.829 | 2.562 | 4.179 | 0.826 | 4.261 | 4.293 | 0.826 | 3.132 | 4.261 |

## M.2. Timing and GPU Usage

In this section, we present the time required and peak GPU memory consumed to compute the coarsening matrix $C_t$ at each time interval $t$ for the three different settings, as described in the previous section, using the D-PUBMED dataset. As shown in Figure 5 and Figure 6, the DIGC and ACNR approaches are approximately 10–20× faster than recomputing the coarsening matrix $C_t$ from scratch at each time step. Moreover, there is a significant reduction in peak GPU memory usage compared to the state-of-the-art method (FGC). This highlights the computational efficiency of our proposed methods.

These results demonstrate that DIGC and ACNR not only achieve comparable or better performance in terms of accuracy and error metrics but also offer substantial improvements in runtime and GPU memory efficiency.

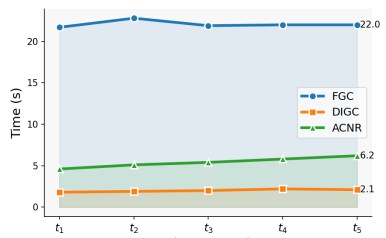
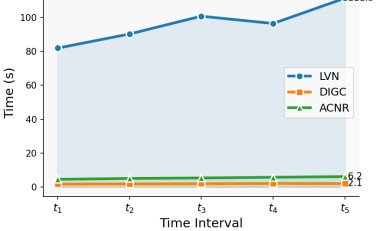
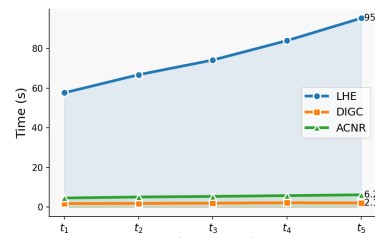

(a) Total time (in seconds) to compute the mapping matrix on D-PUBMED using FGC for $C_0$, with dynamic updates via DIGC and ACNR.

(b) Total time (in seconds) to compute the mapping matrix on D-PUBMED using LVN for $C_0$, with dynamic updates via DIGC and ACNR.

(c) Total time (in seconds) to compute the mapping matrix on D-PUBMED using LHE for $C_0$, with dynamic updates via DIGC and ACNR.

*Figure 5.* Comparison of total time taken (in seconds) to compute the mapping matrix $C_t$ over five time intervals for the D-PUBMED dataset. Each subplot shows one initialization method: (a) FGC, (b) LVN, and (c) LHE, followed by dynamic updates using DIGC and ACNR.

## N. Hyperparameter Sensitivity

To assess the sensitivity of hyperparameters in the DIGC and ACNR frameworks, we evaluate their impact directly on structural preservation quality. Specifically, we report the average Relative Eigen Error (REE), Reconstruction Error (RE), and Hyperbolic Error (HE) on the ACM dataset with a coarsening ratio of $r = 0.25$, where the initial coarsening matrix $C_0$ is obtained using FGC.

We consider two settings for both DIGC and ACNR: (i) *without hyperparameter tuning*, where all weighting and refinement coefficients are fixed to default values, and (ii) *with hyperparameter tuning*, where these coefficients are optimized using a Tree-Structured Parzen Estimator (TPE) based Bayesian optimization strategy.

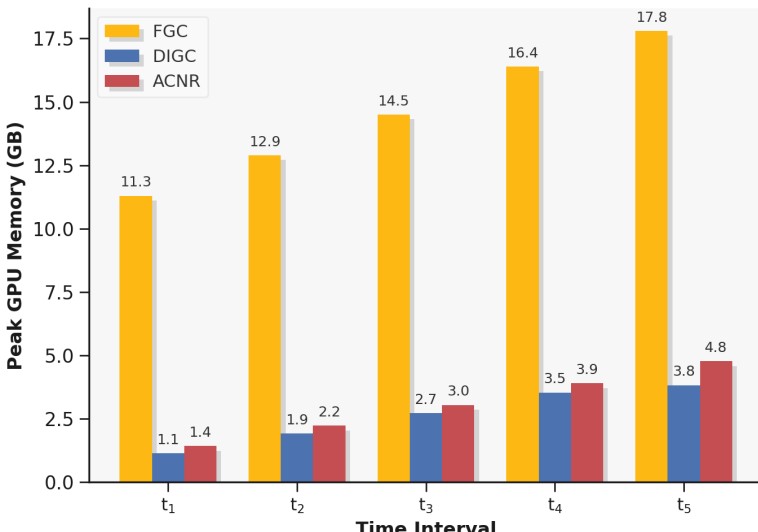

*Figure 6.* Peak GPU memory usage (in GB) for computing the mapping matrix $C_t$ on the D-PUBMED dataset across five time intervals. Initially, $C_0$ is generated using the state-of-the-art FGC method and subsequently updated at each step using our proposed DIGC and ACNR approaches, alongside a full recomputation from scratch.

**Hyperparameter tuning protocol.** We adopt a TPE-based Bayesian optimization framework (via Hyperopt) to tune all method-specific weighting and refinement coefficients in a unified workflow. In TPE, each hyperparameter configuration is represented by a vector $x$, and its performance is quantified by the corresponding structural error metrics. A threshold $y^*$, typically chosen as a high quantile of previously observed evaluations, partitions the objective values into "good" ($y < y^*$) and "bad" ($y \geq y^*$) regions. TPE models the conditional densities $p(x \mid y < y^*)$ and $p(x \mid y \geq y^*)$ and proposes new configurations by maximizing the expected improvement,

$$\mathrm{EI}(x) = \int_{-\infty}^{y^*} (y^* - y) \, p(y \mid x) \, dy,$$

thereby balancing exploration and exploitation of the hyperparameter space.

For each time snapshot, we perform 20 TPE evaluations, where each trial samples a new set of coefficients from a log-uniform search space, executes the corresponding coarsening update, and computes the resulting structural error metrics. The optimal hyperparameter configuration is selected by minimizing these errors, and the reported values correspond to the full objective.

Table 14 summarizes the average structural errors under both settings. Without tuning, DIGC exhibits higher REE, reflecting sensitivity to fixed weighting choices and its limited ability to correct incremental assignment errors. ACNR already demonstrates improved spectral fidelity due to its adaptive reassignment mechanism. With optimized hyperparameters, ACNR achieves the lowest REE while maintaining comparable reconstruction and hyperbolic errors, highlighting its robustness and stability under dynamic graph evolution. Overall, these results indicate that ACNR is less sensitive to hyperparameter selection than DIGC, particularly with respect to spectral preservation.

**Remark.** Table 14 shows that the structural error metrics for both DIGC and ACNR vary only marginally between tuned and untuned settings. Even without explicit hyperparameter optimization, REE, RE, and HE remain close to their optimized values, indicating that the proposed objectives are well-conditioned. In particular, ACNR maintains strong spectral properties under default parameter choices, demonstrating robustness to hyperparameter selection.

## O. Additional Efficiency Measurements

In addition to measuring the time required to update the coarsening matrix, we also report the end-to-end node-classification cost including the coarsening update in Table 15. This measurement compares the full-graph pipeline with the coarsened

*Table 14.* **Average structural error metrics under hyperparameter sensitivity analysis.** The table reports the average Relative Eigen Error (REE), Reconstruction Error (RE), and Hyperbolic Error (HE) for DIGC and ACNR on the ACM dataset with FGC initialization and coarsening ratio $r = 0.25$. Results are shown for each method with and without hyperparameter tuning using TPE-based Bayesian optimization. Lower is better for all metrics.

| Method | Setting | REE | RE | HE |
|--------|---------|------|------|------|
| DIGC | No tuning | 0.9252 | 4.8378 | 2.9244 |
| DIGC | With tuning | 0.8899 | 4.7961 | 2.9065 |
| ACNR | No tuning | 0.6214 | 4.8415 | 3.0673 |
| ACNR | With tuning | 0.5273 | 4.8374 | 3.0637 |

pipeline after including the dynamic update overhead. The purpose of this experiment is to verify that the efficiency trend observed for coarsening-map computation also carries over to the downstream learning pipeline.

*Table 15.* End-to-end node-classification runtime and peak GPU memory, including coarsening-update overhead. Runtime is in seconds and memory is in MiB. Values are averaged over temporal snapshots.

| Dataset | Method | Runtime ↓ | Peak Memory ↓ | Setting |
|---------|--------|-----------|---------------|---------|
| | Full graph | 1.04 | 1427 | No coarsening |
| ACM | DIGC | 0.19 | 735 | Coarsening + GNN |
| | ACNR | 0.24 | 881 | Coarsening + GNN |
| | Full graph | 0.84 | 1508 | No coarsening |
| DBLP | DIGC | 0.20 | 536 | Coarsening + GNN |
| | ACNR | 0.29 | 819 | Coarsening + GNN |

These results show that the additional optimization step introduced by DIGC and ACNR does not remove the downstream efficiency benefit of coarsening. The coarsened pipelines reduce both total runtime and peak memory relative to full-graph node classification.

## P. Ablation Study

We conduct an ablation study to systematically assess the individual contributions of each component in the ACNR objective (Equation 12). Specifically, we remove each term independently, as well as selected combinations of terms, and evaluate the resulting degradation in structural properties using Relative Eigen Error (REE), Reconstruction Error (RE), and Hyperbolic Error (HE). All experiments are performed on the ACM dataset with FGC initialization to ensure a consistent baseline.

**No terms removed.** The full ACNR objective consistently achieves the lowest structural distortion, most notably in terms of REE, demonstrating the importance of jointly modeling temporal smoothness, feature alignment, node remapping consistency, and topology refinement. Across all settings, ACNR substantially outperforms DIGC in spectral preservation, highlighting the benefit of adaptively updating the mappings of previously coarsened nodes rather than restricting updates solely to newly added nodes.

**Without Term-1 (Temporal smoothness).** Removing the temporal smoothness term results in a clear increase in REE for both DIGC and ACNR, indicating that enforcing consistency between consecutive coarsened graphs is critical for stabilizing spectral properties under dynamic evolution. In the absence of this constraint, the coarsened representation becomes more sensitive to snapshot-level fluctuations, leading to increased eigenvalue distortion.

**Without Term-2 (Feature alignment for newly added nodes).** Ablating the feature alignment term primarily degrades REE and HE, suggesting that feature-guided assignment is essential for maintaining both spectral and geometric consistency when integrating new nodes. This observation confirms that purely topology-driven updates are insufficient in attributed graphs, particularly during the initial embedding of newly introduced nodes.

**Without Term-3 (Node remapping consistency).** Removing the remapping consistency term leads to one of the most pronounced degradations in ACNR performance, especially in terms of REE. This highlights the central role of remapping

consistency in enabling adaptive correction of earlier coarsening decisions. Without this mechanism, ACNR loses its ability to realign existing nodes in response to evolving connectivity and effectively degenerates toward a rigid incremental behavior similar to DIGC.

**Without Term-2 and Term-3.**    The combined removal of feature alignment and remapping consistency further amplifies structural degradation across all metrics. This result demonstrates that these two components play complementary roles: feature alignment guides meaningful reassignment of nodes, while remapping consistency ensures stability and coherence during adaptive updates.

**Without Term-4 (Topology refinement).**    Ablating the topology refinement term increases REE and HE while leaving reconstruction error relatively stable. This indicates that explicitly modeling changes in connectivity among previously existing nodes is crucial for preserving global spectral and geometric structure, even when local adjacency reconstruction remains accurate.

**Without Term-2 and Term-4.**    Finally, removing both feature alignment and topology refinement reveals a distinct failure mode. Although RE remains comparatively stable, both REE and HE deteriorate noticeably, particularly for ACNR. This suggests that feature alignment and topology refinement jointly govern global structural coherence: without feature guidance, newly added nodes are poorly embedded, and without topology refinement, evolving edge patterns among existing nodes are inadequately captured. Consequently, the coarsened graph preserves local reconstruction fidelity but fails to maintain global spectral and geometric consistency.

Overall, this ablation study demonstrates that no individual component is sufficient on its own to achieve robust dynamic coarsening. Instead, the strong performance of ACNR arises from the synergistic interaction of all objective terms, with node remapping consistency and topology refinement playing particularly critical roles in long-term structural preservation under evolving graph dynamics.

*Table 16.* **Ablation study on ACNR: Structural error analysis.** We report the average Relative Eigen Error (REE), Reconstruction Error (RE), and Hyperbolic Error (HE) for DIGC and ACNR under different objective term removal settings. Results are averaged across all time steps on the ACM dataset with FGC initialization for $r = 0.25$. Lower values indicate better structural preservation. Best results within each column are highlighted in **bold**.

| ACNR Variant | DIGC | | | ACNR | | |
|---|---|---|---|---|---|---|
| | REE | RE | HE | REE | RE | HE |
| All terms (full ACNR) | **0.8899** | **4.7961** | 2.9065 | **0.5273** | **4.8374** | **3.0637** |
| w/o term-1 | 0.9730 | 4.8316 | 2.9412 | 0.5702 | 4.8471 | 3.1433 |
| w/o term-2 | 0.9557 | 4.8251 | 2.9819 | 0.5811 | 4.8530 | 3.1246 |
| w/o term-3 | 0.9507 | 4.8199 | 2.9625 | 0.5865 | 4.8450 | 3.1019 |
| w/o term-2 & term-3 | 0.9628 | 4.8045 | **2.8789** | 0.5589 | 4.8586 | 3.1288 |
| w/o term-2 & term-4 | 0.9165 | 4.8127 | 2.8999 | 0.6154 | 4.8462 | 3.0994 |

## Q. Space Complexity

We provide the per-step space complexity of DIGC and ACNR in Table 17. The sparse form accounts for the sparse graph-update structure used in implementation, while the dense form gives the corresponding worst-case bound.

*Table 17.* Per-step space complexity of DIGC and ACNR.

| Method | Sparse space complexity | Dense worst-case space complexity |
|---|---|---|
| DIGC | $O\big(N_t n + kn + \mathrm{nnz}(M_2) + \mathrm{nnz}(M_4) + nd + kd + n^2\big)$ | $O\big(N_t n + kn + N_t k + k^2 + nd + kd + n^2\big)$ |
| ACNR | $O\big(2N_t n + kn + \mathrm{nnz}(D) + \mathrm{nnz}(A_{t-1}^{new}) + \mathrm{nnz}(M_4) + nd + kd + n^2\big)$ | $O\big(2N_t n + kn + N_t k + N_t^2 + k^2 + nd + kd + n^2\big)$ |

The difference follows from the update variables maintained by the two methods. DIGC stores the previous mapping and the incremental assignment block $\Delta\mathbf{C_t}$. ACNR additionally stores the reassignment block $\Delta\mathbf{C_{S_t}}$ together with $\Delta\mathbf{C_{N_t}}$, which increases the memory requirement but allows the method to adapt previous assignments under topology changes.

In the dense worst case, the dominant terms are

$$O(N_t n + N_t k)$$

for DIGC and

$$O(2N_t n + N_t^2)$$

for ACNR. This reflects the intended trade-off between the two algorithms: DIGC is more memory efficient for incremental growth, while ACNR uses additional space to support reassignment and topology refinement.

## R. Convergence analysis

## Convergence rates

**Theorem R.1.** *SGD with constant step size $\eta$ applied to DIGC and ACNR converges at rate*

$$\frac{1}{T}\sum_{t=0}^{T}\mathbb{E}[\|\nabla f(\theta^{(t)})\|_2^2] \leq \frac{F}{\eta T} + \frac{L\sigma^2\eta}{2} \tag{24}$$

*where $F \geq f(\theta^{(0)}) - \mathbb{E}[f(\theta^{(T)})]$, the objective function is L-smooth, and $\sigma$ upper bounds the stochastic gradient norm. Specifically, for DIGC,*

$$
\begin{aligned}
L &= O(\|M_2\|n^{4.5}(N+kt)k(\|M_2\|_2(N+kt)+\|M_4\|k) \\
F &= O(n^6 k^2(\|M_2\|_2(N+kt)+\|M_4\|_2 k)^2) \\
\sigma &= O(n^{4.5}k(\|M_2\|_F n^3(N+kt)+\|M_4\|_F k)^2
\end{aligned}
$$

*and for ACNR*

$$
\begin{aligned}
L &= O(\|A_{t-1}\|_F(N+kt)^3 n^{4.5}(\|A_{t-1}\|_F+\|M_2\|_F)+\|D\|_2^2+\|A_{t-1}^{new}\|_F n^{4.5}(N+tk)^3(3\|A_{t-1}^{new}\|_F+\|(A_{t-1}^{new}-A_{t-1})\|_F) \\
F &= O((7\|A_{t-1}\|_F+\|M_2\|)n^3(N+kt)^2+n^5(N+kt)^2+n^6(N+tk)^4(4\|A_{t-1}^{new}\|_F+\|A_{t-1}\|_F)^2) \\
\sigma &= O((\|A_{t-1}\|_F+\|M_2\|_F)^2 n^{4.5}(N+kt)^3+n^3(N+kt)^2\|D\|_F+\|A_{t-1}^{new}\|_F n^3(N+tk)^3(\|A_{t-1}^{new}\|_F+\|A_{t-1}\|_F))
\end{aligned}
$$

The proof of (24) can be found in (Bottou, 2010; Garrigos & Gower, 2023), and the constant derivations are given in Lemmas R.3 and R.4.

**Lemma R.2.** *The function*

$$f(X) = \frac{1}{2}\|B^T X + X^T B + X^T M X + E\|_F^2$$

*is L-smooth , with*

$$L = \|B\|_F \max_X \|X\|_F(2\|B\|_F + \|M\|_F \max_X \|X\|_F) + \|B\|_F\|E\|_F.$$

*Additionally, the gradient norm is bounded as*

$$\|\nabla f(X)\|_F \leq (\|M\|_F\|X\|_F+\|B\|_F)(2\|B\|_F\|X\|_F+\|M\|_F\|X\|_F^2+\|E\|_F)$$

*Proof.*

$$
\begin{aligned}
f(X) &= \frac{1}{2}\|B^T X + X^T B + X^T M X + E\|_F^2 \\
&= \frac{1}{2}\|(M^{1/2}X + M^{-1/2}B)^T \underbrace{(M^{1/2}X + M^{-1/2}B)}_{C} - BMB^T + E\|_F^2
\end{aligned}
$$

so

$$\nabla f(X) = M^{T/2}C \underbrace{(C^TC - BMB^T + E)}_{D}$$

and writing $C_X = M^{1/2}X + M^{-1/2}B$, $C_Y = M^{1/2}Y + M^{-1/2}B$ and $D_X = C_X^T C_X - BMB^T + E$, $D_Y = C_Y^T C_Y - BMB^T + E$, and using Cauchy Schwartz inequality,

$$\|AB\|_F^2 = \mathbf{tr}(B^T A^T A B) \leq \|A^T A\|_F \|B^T B\|_F \leq \|A\|_F^2 \|B\|_F^2$$

where the last inequality follows from the observation that if $\sigma_i$ are the singular values of $A$, then

$$\|A^T A\|_F = \sqrt{\sum_i (\sigma_i^2)^2} = \sqrt{\sum_i \sigma_i^4} \leq \sum_i \sigma_i^2 = \|A\|_F^2.$$

So

$$
\begin{aligned}
\|\nabla f(X) - \nabla f(Y)\|_F &= \|B^{T/2}(C_X^T D_X - C_Y^T D_Y)\|_F \\
&\leq \|B\|_F \|X^T D_X - Y^T D_Y\|_F \\
&\leq \|B\|_F \max_X \|D_X\|_F \|X - Y\|_F
\end{aligned}
$$

Further simplifying,

$$
\begin{aligned}
\max_X \|D_X\|_F &= \max_X \|B^T X + X^T B + X^T M X + E\|_F \\
&\leq \max_X (2\|B^T X\|_F + \|X^T M X\|_F + \|E\|_F) \\
&\leq 2\|B\|_F \max_X \|X\|_F + \|M\| \max_X \|X\|_F^2 + \|E\|_F.
\end{aligned}
$$

Next,

$$
\begin{aligned}
\|\nabla f(X)\|_F &= \|M^{T/2}C(C^TC - BMB^T + E)\|_F \\
&\leq \|M^{T/2}C\|_F \|C^TC - BMB^T + E\|_F \\
&= \|MX + B\|_F \|B^T X + X^T B + X^T M X + E\|_F \\
&\leq (\|M\|_F \|X\|_F + \|B\|_F)(2\|B\|_F \|X\|_F + \|M\|_F \|X\|_F^2 + \|E\|_F)
\end{aligned}
$$

$\square$

**Lemma R.3** (DIGC (DGC 1.0) properties). *Consider the optimization problem*

$$
\begin{aligned}
\min_{\mathbf{\Delta C_t}} \quad & f(\mathbf{\Delta C_t}) := \frac{\omega}{2}\|A_{c_t} - A_{c_{t-1}}\|_F^2 + \frac{\zeta}{2}\|\mathbf{\Delta C_t} X_{sn} - X_n\|_F^2 \\
\text{subj. to} \quad & \mathbf{\Delta C_t} \geq 0, \quad \|\mathbf{\Delta C_t}.1 - 1\|_F^2 = 0
\end{aligned}
\tag{25}
$$

*where*

$$A_{c_t} - A_{c_{t-1}} = \mathbf{\Delta C_t}^\top M_2^\top C_{t-1} + C_{t-1}^\top M_2 \mathbf{\Delta C_t} + \mathbf{\Delta C_t}^\top M_4 \mathbf{\Delta C_t}. \tag{26}$$

*Assume each feature is normalized, e.g. $\|X_n\|_F \leq k$ and $\|X_{sn}\|_F \leq n$. Then, for all feasible $\Delta C_t$, the function $f(\Delta C_t)$ satisfies the following:*

- *$f$ is L-smooth with $L = \omega L_1 + \zeta L_2$*

$$L_1 = 2\|M_2\|_2^2(N + kt)^2 kn^{4.5} + \|M_4\|\|M_2\|k^2 n^{4.5}(N + kt), \qquad L_2 = n^2$$

- *For initial value $\Delta C^{(0)}$, the constant bound $F \geq f(\Delta C^{(0)}) - f^*$ is satisfied by $F = \omega F_1 + \zeta F_2$*

$$F_1 = n^6 k^2 (2\|M_2\|_2^2 (N + kt)^2 + \frac{1}{2}\|M_4\|_2^2 k^2 + 2\|M_2\|_2 (N + kt)\|M_4\|_2 k), \qquad F_2 = k^2 (\frac{n^5}{2} + n^{2.5} + \frac{1}{2})$$

- *The stochastic gradients $\nabla_i f(\Delta C_t)$ satisfy $\|\nabla_i f(\Delta C_t)\|_F \leq \|\nabla f(\Delta C_t)\|_F \leq \sigma = \omega \sigma_1 + \zeta \sigma_2$ for*

$$\sigma_1 = n^3 k \big( 2\|M_2\|_F \|M_4\|_F n^{4.5}(N + kt)k + n^{1.5}k^2 \|M_4\|_F^2 + 2\|M_2\|_F^2 n^{1.5}(N + kt)^2 + \|M_4\|_F \|M_2\|_F k(N + kt) \big),$$

$$\sigma_2 = n^{3.5} + n^{2.5}k$$

*Proof.* First, note that $C_t \in \mathbb{R}^{N+kt \times n}$ and is row stochastic, so $\|C_t\|_F \leq \sqrt{n}\|C_t\|_1 = n^{3/2}(N + kt)$. Additionally, $\Delta C_t \in \mathbb{R}^{k \times n}$ and is row stochastic, so by the same logic $\|\Delta C_t\|_F \leq n^{3/2}k$.

**L-smoothness.** Next, taking $B = M_2^\top C_{t-1}$ and invoking Lemma R.2, then

$$f_1(\Delta C_t) := \frac{1}{2}\|A_{c_t} - A_{c_{t-1}}\| \tag{27}$$

is $L_1$ smooth with

$$
\begin{aligned}
\frac{\|\nabla f_1(C_1) - \nabla f_1(C_2)\|_F}{\|C_1 - C_2\|_F} &\leq \|M_2\| \cdot \|C_{t-1}\|_F \cdot \|\mathbf{\Delta C_t}\|_F \cdot (2\|M_2\|_2 \|C_{t-1}\|_F + \|M_4\| \|\mathbf{\Delta C_t}\|_F) \\
&\leq \|M_2\| \cdot (n^{3/2}(N + kt)) \cdot (n^{3/2}k) \cdot (2\|M_2\|_2 (n^{3/2}(N + kt)) + \|M_4\|(n^{3/2}k)) \\
&= 2\|M_2\|_2^2 (N + kt)^2 k n^{4.5} + \|M_4\| \|M_2\| k^2 n^{4.5}(N + kt) =: L_1
\end{aligned}
$$

Simplifying presents the result for $L_1$. For $L_2$, take

$$f_2(\mathbf{\Delta C_t}) = \frac{1}{2}\|\mathbf{\Delta C_t} X_{sn} - X_n\|_F^2 \tag{28}$$

which is a quadratic, and is $\|X_{sn}\|_2^2$-smooth, where $\|X_{sn}\|_2 \leq \|X_{sn}\|_F = n$.

**Function value bound** This is derived from first principles and plugging in norms for each variable:

$$
\begin{aligned}
\max_{\mathbf{\Delta C_t}} f_1(\mathbf{\Delta C_t}) &= \frac{1}{2}(2\|M_2\|_2 \|C_{t-1}\|_F \|\mathbf{\Delta C_t}\|_F + \|M_4\| \|\mathbf{\Delta C_t}\|_F^2) \\
&= \frac{1}{2}(2\|M_2\|_2 n^3 (N + kt)k + \|M_4\| n^3 k^2)^2 \\
&= n^6 k^2 (2\|M_2\|_2^2 (N + kt)^2 + \frac{1}{2}\|M_4\|_2^2 k^2 + 2\|M_2\|_2 (N + kt)\|M_4\|_2 k) =: F_1
\end{aligned}
$$

$$
\begin{aligned}
\max_{\mathbf{\Delta C_t}} f_2(\mathbf{\Delta C_t}) &= \frac{1}{2}\|\Delta C_0 X_{sn} - X_n\|_F^2 \leq \frac{1}{2}(\|\Delta C_0\|_F \|X_{sn}\|_F + \|X_n\|_F)^2 \\
&\leq \frac{1}{2}(n^{3/2}k \cdot n + k)^2 = k^2 (\frac{n^5}{2} + n^{2.5} + \frac{1}{2})
\end{aligned}
$$

**Gradient bound** Invoking Lemma R.2,

$$
\begin{aligned}
\|\nabla f_1(\mathbf{\Delta C_t})\|_F \;\leq\;\; & 2\|M_2^T C_{t-1}\|_F\|\mathbf{\Delta C_t}\|_F^2\|M_4\|_F + \|\mathbf{\Delta C_t}\|_F^3\|M_4\|_F^2 \\
& + 2\|M_2^T C_{t-1}\|_F^2\|\mathbf{\Delta C_t}\|_F + \|\mathbf{\Delta C_t}\|_F^2\|M_4\|_F\|M_2^T C_{t-1}\|_F \\
\leq\;\; & n^3 k\Big( 2\|M_2\|_F\|M_4\|_F n^{4.5}(N+kt)k + n^{1.5}k^2\|M_4\|_F^2 \\
& + 2\|M_2\|_F^2 n^{1.5}(N+kt)^2 + \|M_4\|_F\|M_2\|_F k(N+kt)\Big) =: \sigma_1
\end{aligned}
$$

$$
\begin{aligned}
\|\nabla f_2(\mathbf{\Delta C_t})\|_F \;=\;\; & \|X_{sn}^T(\mathbf{\Delta C_t}X_{sn} - X_n)\|_F \\
\leq\;\; & \|X_{sn}\|_F(\|\mathbf{\Delta C_t}\|_F\|X_{sn}\|_F + \|X_n\|_F) \\
\leq\;\; & n((n^{3/2}k)n + k) = n^{3.5} + n^{2.5}k =: \sigma_2
\end{aligned}
$$

$\square$

**Lemma R.4** (ACNR (DGC 2.0) Properties)**.** *Consider the optimization problem over* $\mathbf{\Delta C} = (\mathbf{\Delta C_{N_t}}, \mathbf{\Delta C_{S_t}})$

$$
\begin{aligned}
\min_{\mathbf{\Delta C_N}, \mathbf{\Delta C_S}} \quad & f(\mathbf{\Delta C}) := -\frac{\omega}{2}\|A_{c_t} - A_{c_{t-1}}\|_F^2 + \frac{\zeta}{2}\|\mathbf{\Delta C_{N_t}}X_{sn} - X_n\|_F^2 \\
& + \frac{\beta}{2}\|(C_{t-1} + \mathbf{\Delta C_{S_t}}) - D\mathbf{\Delta C_{N_t}}\|_F^2 \\
& + \frac{\alpha}{2}\|(C_{t-1} + \mathbf{\Delta C_{S_t}})^\top A_{t-1}^{new}(C_{t-1} + \mathbf{\Delta C_{S_t}}) - C_{t-1}^\top A_{t-1}C_{t-1}\|_F^2 \\
\text{subj. to} \quad & \mathbf{\Delta C_{N_t}} \geq 0, \quad \mathbf{\Delta C_{S_t}} \geq 0, \quad \|\mathbf{\Delta C_{N_t}} \cdot 1 - 1\|_F^2 = 0, \quad \|\mathbf{\Delta C_{S_t}} \cdot 1 - 1\|_F^2 = 0
\end{aligned}
\tag{29}
$$

*where*

$$
\begin{aligned}
(A_{c_t} - A_{c_{t-1}}) = \;& C_{t-1}^\top A_{t-1}\mathbf{\Delta C_{S_t}} + \mathbf{\Delta C_{S_t}}^\top A_{t-1}C_{t-1} + \mathbf{\Delta C_{S_t}}^\top A_{t-1}\mathbf{\Delta C_{S_t}} + \mathbf{\Delta C_{N_t}}^\top M_2^\top C_{t-1} \\
& + \mathbf{\Delta C_{N_t}}^\top M_2^\top \mathbf{\Delta C_{S_t}} + C_{t-1}^\top M_2 \mathbf{\Delta C_{N_t}} + \mathbf{\Delta C_{S_t}}^\top M_2 \mathbf{\Delta C_{N_t}} + \mathbf{\Delta C_{N_t}}^\top M_4 \mathbf{\Delta C_{N_t}}.
\end{aligned}
\tag{30}
$$

*Assume each feature is normalized, e.g.* $\|X_n\|_F \leq k$ *and* $\|X_{sn}\|_F \leq n$. *Then, for all feasible* $(\mathbf{\Delta C_{N_t}}, \mathbf{\Delta C_{S_t}})$, *the function* $f(\mathbf{\Delta C})$ *satisfies the following:*

- *f is L-smooth with* $L = \omega L_1 + \zeta L_2 + \beta L_3 + \alpha L_4$

  $L_1 = \|A_{t-1}\|_F(N+kt)^2 n^3 \max\{n^{1.5}(N+kt)(5\|A_{t-1}\|_F + \|M_2\|_F), 10\|A_{t-1}\|_F n^{1.5}(N+kt) + 6\|A_{t-1}\|_F\},$

  $L_2 = \|X_{sn}\|_2^2 \leq n^2, \qquad L_3 = (1+\|D\|_2)^2, \qquad L_4 = \|A_{t-1}^{new}\|_F(n^{3/2}(N+tk))^3(3\|A_{t-1}^{new}\|_F + \|(A_{t-1}^{new} - A_{t-1})\|_F).$

- *For initial value* $\mathbf{\Delta C^{(0)}}$, *the constant bound* $F \geq f(\mathbf{\Delta C^{(0)}}) - f^*$ *is satisfied by* $F = \omega F_1 + \zeta F_2 + \beta F_3 + \alpha F_4$

  $$
  F_1 = (7\|A_{t-1}\|_F + \|M_2\|)n^3(N+kt)^2, \qquad F_2 = \frac{n^5(N+kt)^2}{2} + n^{2.5}k(N+kt) + \frac{k^2}{2},
  $$

  $$
  F_3 = \frac{n^5(N+tk)^2}{2} + n^{2.5}(N+tk)k + \frac{k^2}{2}, \qquad F_4 = \frac{1}{2}n^6(N+tk)^4(4\|A_{t-1}^{new}\|_F + \|A_{t-1}\|_F)^2.
  $$

- *The stochastic gradients* $\nabla_i f(\mathbf{\Delta C_t})$ *satisfy* $\|\nabla_i f(\mathbf{\Delta C_t})\|_F \leq \|\nabla f(\mathbf{\Delta C_t})\|_F \leq \sigma = \omega\sigma_1 + \zeta F\sigma_2 + \beta\sigma_3 + \alpha\sigma_4$

  $$
  \sigma_1 = (9\|A_{t-1}\|_F + \|M_2\|_F)^2 n^{4.5}(N+kt)^3, \qquad \sigma_2 = n^{3.5}(N+kt) + nk,
  $$

  $$
  \sigma_3 = 2(n^{3/2}(N+kt))^2(\|D\|_F + \sqrt{n} + 1), \qquad \sigma_4 = 4\|A_{t-1}^{new}\|_F n^3(N+tk)^3\left(4\|A_{t-1}^{new}\|_F + \|A_{t-1}\|_F\right).
  $$

*Proof.* We approach this term by term.

**First term**  First of all, using $f_1(\mathbf{\Delta C}) = \frac{1}{2}\|(A_{c_t} - A_{c_{t-1}})\|_F^2$, then

$$f_1(\mathbf{\Delta C}^{(0)}) - f_1^* \leq \max_{\mathbf{\Delta C}} -f_1(\mathbf{\Delta C}) \quad \leq \quad (7\|A_{t-1}\|_F + \|M_2\|)(n^{3/2}(N + kt))^2$$

For $\sigma_1$ and $L_1$, we take them with respect to the two matrix variables separately, and use the larger one.

**First term, first variable.**  We rewrite

$$
\begin{aligned}
(A_{c_t} - A_{c_{t-1}}) &= C_{t-1}^\top A_{t-1}(\mathbf{\Delta C_{S_t}} + \mathbf{\Delta C_{N_t}}) + (\mathbf{\Delta C_{S_t}}^\top + \mathbf{\Delta C_{N_t}}^\top)A_{t-1}^\top C_{t-1} \\
&\quad + (\mathbf{\Delta C_{S_t}}^\top + \mathbf{\Delta C_{N_t}}^\top)A_{t-1}^\top(\mathbf{\Delta C_{S_t}} + \mathbf{\Delta C_{N_t}}) + \mathbf{\Delta C_{N_t}}^\top(M_2 - A_{t-1})\mathbf{\Delta C_{N_t}}.
\end{aligned}
$$

For a function $g(X + Y)$, $\nabla_X g(X + Y) = \nabla_Y g(X + Y) = \nabla_{X+Y} g(X + Y)$, so if $g$ is $L$-smooth w.r.t. $X + Y$, then it is $L$-smooth w.r.t. $X$ and w.r.t. $Y$. Then, invoking Lemma R.2, we have

$$
\begin{aligned}
\frac{\|\nabla_{\mathbf{\Delta C_{S_t,1}}} f_1(\mathbf{\Delta C}) - \nabla_{\mathbf{\Delta C_{S_t,1}}} f_1(\mathbf{\Delta C})\|_F}{\|\mathbf{\Delta C_{S_t,1}} - \mathbf{\Delta C_{S_t,2}}\|_F} \quad &\leq \quad \|A_{t-1}\|_F^2 \cdot \|C_{t-1}\|_F \cdot \|\mathbf{\Delta C_{S_t}} + \mathbf{\Delta C_{N_t}}\|_F \cdot (2\|C_{t-1}\|_F + \|\mathbf{\Delta C_{S_t}} + \mathbf{\Delta C_{N_t}}\|_F) \\
&\qquad + \|A_{t-1}\|_F \|C_{t-1}\|_F \|\mathbf{\Delta C_{N_t}}^\top(M_2 - A_{t-1})\mathbf{\Delta C_{N_t}}\|_F \\
&\leq \quad \|A_{t-1}\|_F \cdot (n^{3/2}(N + kt))^3 (4\|A_{t-1}\|_F + \|(M_2 - A_{t-1})\|_F) \\
&\leq \quad \|A_{t-1}\|_F \cdot (n^{3/2}(N + kt))^3 (5\|A_{t-1}\|_F + \|M_2\|_F)
\end{aligned}
$$

Additionally,

$$
\begin{aligned}
\|\nabla_{\mathbf{\Delta C_{S_t}}} f_1(\mathbf{\Delta C})\|_F \quad &\leq \quad \|A_{c_t} - A_{c_{t-1}}\|_F (2\|C_{t-1}^\top A_{t-1}\|_F + 2\|A_{t-1}\|_F \|\mathbf{\Delta C_{S_t}} + \mathbf{\Delta C_{N_t}}\|_F) \\
&\leq \quad (8\|A_{t-1}\|_F(n^{3/2}(N + kt))^2 + \|M_2 - A_{t-1}\|_F(n^{3/2}(N + kt))^2) \cdot ((6\|A_{t-1}\|_F n^{3/2}(N + kt))) \\
&\leq \quad (9\|A_{t-1}\|_F + \|M_2\|_F) \cdot (6\|A_{t-1}\|_F n^{4.5}(N + kt)^3) \\
&\leq \quad (9\|A_{t-1}\|_F + \|M_2\|_F)^2 \cdot n^{4.5}(N + kt)^3
\end{aligned}
$$

**First term, second variable.**  Since

$$
\begin{aligned}
(A_{c_t} - A_{c_{t-1}}) &= \mathbf{\Delta C_{N_t}}^\top M_2 \mathbf{\Delta C_{N_t}} + (C_{t-1} + \mathbf{\Delta C_{S_t}})^\top A_{t-1}(\mathbf{\Delta C_{N_t}}) + (\mathbf{\Delta C_{N_t}}^\top)A_{t-1}^\top(C_{t-1} + \mathbf{\Delta C_{S_t}}) \\
&\quad + (\mathbf{\Delta C_{S_t}}^\top)A_{t-1}^\top(\mathbf{\Delta C_{S_t}}) + C_{t-1}^\top A_{t-1}(\mathbf{\Delta C_{S_t}}) + (\mathbf{\Delta C_{S_t}}^\top)A_{t-1}^\top C_{t-1}.
\end{aligned}
$$

then

$$
\begin{aligned}
\frac{\|\nabla_{\mathbf{\Delta C_{N_t,1}}} f_1(\mathbf{\Delta C}) - \nabla_{\mathbf{\Delta C_{N_t,1}}} f_1(\mathbf{\Delta C})\|_F}{\|\mathbf{\Delta C_{N_t,1}} - \mathbf{\Delta C_{N_t,2}}\|_F} \quad &\leq \quad \|A_{t-1}(C_{t-1} + \mathbf{\Delta C_{S_t}})\|_F \|\mathbf{\Delta C_{N_t,1}}\|_F \Big(2\|A_{t-1}(C_{t-1} + \mathbf{\Delta C_{S_t}})\|_F \\
&\qquad + \|A_{t-1}\|_F \|\mathbf{\Delta C_{N_t,1}}\|_F\Big) + \|A_{t-1}(C_{t-1} + \mathbf{\Delta C_{S_t}})\|_F \cdot \\
&\qquad \|(\mathbf{\Delta C_{S_t}}^\top)A_{t-1}^\top(\mathbf{\Delta C_{S_t}}) + C_{t-1}^\top A_{t-1}(\mathbf{\Delta C_{S_t}}) + (\mathbf{\Delta C_{S_t}}^\top)A_{t-1}^\top C_{t-1}\|_F \\
&\leq \quad 10\|A_{t-1}\|_F^2 n^{4.5}(N + kt)^3 + 6\|A_{t-1}\|_F^2 n^3(N + kt)^2.
\end{aligned}
$$

so $L_1$ is the maximum of the two computed constants. Additionally,

$$
\begin{aligned}
\|\nabla_{\mathbf{\Delta C_{N_t}}} f_1(\mathbf{\Delta C})\|_F \quad &\leq \quad \|A_{c_t} - A_{c_{t-1}}\|_F (2\|(C_{t-1} + \mathbf{\Delta C_{S_t}})^\top A_{t-1}\|_F + 2\|M_2\|_F \|\mathbf{\Delta C_{N_t}}\|_F) \\
&\leq \quad (8\|A_{t-1}\|_F(n^{3/2}(N + kt))^2 + \|M_2 - A_{t-1}\|_F(n^{3/2}(N + kt))^2) \cdot (4\|A_{t-1}\|_F + 2\|M_2\|_F)n^{3/2}(N + kt) \\
&\leq \quad (9\|A_{t-1}\|_F + \|M_2\|_F) \cdot (4\|A_{t-1}\|_F + 2\|M_2\|_F)n^{4.5}(N + kt)^3 \\
&\leq \quad (9\|A_{t-1}\|_F + 2\|M_2\|_F)^2 n^4.5(N + kt)^3
\end{aligned}
$$

**Second term**

$$f_2(\mathbf{\Delta C}) := \frac{1}{2}\|\mathbf{\Delta C_{N_t}} X_{sn} - X_n\|_F^2 \tag{31}$$

Following the same logic as before,

$$
\begin{aligned}
L_2 &: \quad \|X_{sn}\|_2^2 \leq n^2 =: L_2 \\
\sigma_2 &: \quad \|X_{sn}\|_F(\|\mathbf{\Delta C_{N_t}}\|_F\|X_{sn}\|_F + \|X_n\|_F) \leq n((n^{3/2}(N+kt))n+k) = n^{3.5}(N+kt)+nk =: \sigma_2 \\
F_2 &: \quad \frac{1}{2}\|\mathbf{\Delta C_{N_t}^{(0)}} X_{sn} - X_n\|_F^2 \leq \frac{1}{2}(n^{3/2}(N+kt)\cdot n + k)^2 = \frac{n^5(N+kt)^2}{2} + n^{2.5}k(N+kt) + \frac{k^2}{2} =: F_2
\end{aligned}
$$

**Third term**

$$f_3(\mathbf{\Delta C}) := \frac{1}{2}\|(C_{t-1} + \mathbf{\Delta C_{S_t}}) - D\mathbf{\Delta C_N}\|_F^2 = \frac{1}{2}\left\|C_{t-1} + \begin{bmatrix} I & -D \end{bmatrix} \begin{bmatrix} \mathbf{\Delta C_{S_t}} \\ \mathbf{\Delta C_N} \end{bmatrix}\right\|_F^2 \tag{32}$$

so we can apply the same logic as above

$$
\begin{aligned}
L_3 &: \quad \|\begin{bmatrix} I & -D \end{bmatrix}\|_2^2 = (1 + \|D\|_2)^2 =: L_3 \\
F_3 &: \quad \frac{1}{2}(n^{3/2}(N+tk)\cdot n + k)^2 = \frac{n^5(N+tk)^2}{2} + n^{2.5}(N+tk)k + \frac{k^2}{2} =: F_3 \\
\sigma_3 &: \quad (\|\mathbf{\Delta C_{S_t}}\|_F + \|\mathbf{\Delta C_N}\|_F)((\|\mathbf{\Delta C_{S_t}}\|_F + \|\mathbf{\Delta C_N}\|_F)\|\begin{bmatrix} I & -D \end{bmatrix}\|_F + \|C_{t-1}\|_F) \\
&= \quad 2(n^{3/2}(N+kt))^2(\|D\|_F + \sqrt{n} + 1) =: \sigma_3
\end{aligned}
$$

**Fourth term**

$$
\begin{aligned}
f_4(\mathbf{\Delta C}) &:= \frac{1}{2}\|(C_{t-1} + \mathbf{\Delta C_{S_t}})^\top A_{t-1}^{\text{new}}(C_{t-1} + \mathbf{\Delta C_{S_t}}) - C_{t-1}^\top A_{t-1} C_{t-1}\|_F^2 \\
&= \frac{1}{2}\|(\mathbf{\Delta C_{S_t}})^\top A_{t-1}^{\text{new}}(C_{t-1}) + (C_{t-1})^\top A_{t-1}^{\text{new}}(\mathbf{\Delta C_{S_t}}) + (\mathbf{\Delta C_{S_t}})^\top A_{t-1}^{\text{new}}(\mathbf{\Delta C_{S_t}}) + C_{t-1}^\top(A_{t-1}^{\text{new}} - A_{t-1})C_{t-1}\|_F^2
\end{aligned}
$$

Using Lemma R.2,

$$
\begin{aligned}
L_4 &: \quad \|A_{t-1}^{\text{new}} C_{t-1}\|_F\|\mathbf{\Delta C_{S_t}}\|_F(2\|A_{t-1}^{\text{new}} C_{t-1}\|_F + \|A_{t-1}^{\text{new}}\|_F\|\mathbf{\Delta C_{S_t}}\|_F) + \|A_{t-1}^{\text{new}} C_{t-1}\|_F\|C_{t-1}^\top(A_{t-1}^{\text{new}} - A_{t-1})C_{t-1}\|_F \\
&\leq \quad \|A_{t-1}^{\text{new}}\|_F(n^{3/2}(N+tk))^3(3\|A_{t-1}^{\text{new}}\|_F + \|(A_{t-1}^{\text{new}} - A_{t-1})\|_F) =: L_4 \\
F_4 &:= \quad \frac{1}{2}n^6(N+tk)^4(4\|A_{t-1}^{\text{new}}\|_F + \|A_{t-1}\|_F)^2 \\
\sigma_4 &: \quad 2\|(C_{t-1} + \mathbf{\Delta C_{S_t}})^\top A_{t-1}^{\text{new}}(C_{t-1} + \mathbf{\Delta C_{S_t}}) - C_{t-1}^\top A_{t-1} C_{t-1}\|_F\|A_{t-1}^{\text{new}}(C_{t-1} + \mathbf{\Delta C_{S_t}})\|_F \\
&\leq \quad 4\|A_{t-1}^{\text{new}}\|_F n^3(N+tk)^3\left(4\|A_{t-1}^{\text{new}}\|_F + \|A_{t-1}\|_F\right)
\end{aligned}
$$

$$\square$$

# S. Epsilon similarity

*Proof.* For any $x \perp \mathbf{1}$, the connectedness assumption implies $\lambda_2(L_t) > 0$ and hence

$$x^\top L_t x \geq \lambda_2(L_t)\|x\|_2^2.$$

Moreover, by the definition of the spectral norm,

$$\left|x^\top(L_t - L_{\ell,t})x\right| \leq \|L_t - L_{\ell,t}\|_2\|x\|_2^2.$$

Combining these inequalities gives

$$\frac{\left|x^\top (L_t - L_{\ell,t})x\right|}{x^\top L_t x} \leq \frac{\|L_t - L_{\ell,t}\|_2}{\lambda_2(L_t)},$$

which yields the first bound in (18). The second bound in (18) follows from $\|M\|_2 \leq \|M\|_F$ for any matrix $M$.

Next, since $C_{t-1}$ has full column rank, the stacked matrix $C_t = \begin{bmatrix} C_{t-1} \\ \Delta C_t \end{bmatrix}$ also has full column rank and satisfies

$$C_t^\top C_t = C_{t-1}^\top C_{t-1} + \Delta C_t^\top \Delta C_t \succeq C_{t-1}^\top C_{t-1}.$$

Therefore $\sigma_{\min}(C_t) \geq \sigma_{\min}(C_{t-1})$, and hence

$$\|P\|_2 = \|C_t^\dagger\|_2 = \frac{1}{\sigma_{\min}(C_t)} \leq \frac{1}{\sigma_{\min}(C_{t-1})}. \tag{33}$$

Using the block decomposition

$$L_t = \begin{bmatrix} L_1 & L_2 \\ L_3 & L_4 \end{bmatrix}, \qquad L_3 = L_2^\top,$$

and $C_t = \begin{bmatrix} C_{t-1} \\ \Delta C_t \end{bmatrix}$, we expand

$$L_{c,t} = C_t^\top L_t C_t = C_{t-1}^\top L_1 C_{t-1} + C_{t-1}^\top L_2 \Delta C_t + \Delta C_t^\top L_3 C_{t-1} + \Delta C_t^\top L_4 \Delta C_t.$$

Taking Frobenius norms and applying the triangle inequality,

$$\|L_{c,t}\|_F \leq \|C_{t-1}^\top L_1 C_{t-1}\|_F + \|C_{t-1}^\top L_2 \Delta C_t\|_F + \|\Delta C_t^\top L_3 C_{t-1}\|_F + \|\Delta C_t^\top L_4 \Delta C_t\|_F.$$

Each term is controlled by standard submultiplicativity:

$$\|C_{t-1}^\top L_1 C_{t-1}\|_F \leq \|C_{t-1}\|_2^2 \|L_1\|_F,$$

$$\|C_{t-1}^\top L_2 \Delta C_t\|_F \leq \|C_{t-1}\|_F \|L_2\|_2 \|\Delta C_t\|_F,$$

$$\|\Delta C_t^\top L_3 C_{t-1}\|_F \leq \|\Delta C_t\|_F \|L_3\|_2 \|C_{t-1}\|_F = \|C_{t-1}\|_F \|L_2\|_2 \|\Delta C_t\|_F,$$

where we used $\|L_3\|_2 = \|L_2\|_2$, and

$$\|\Delta C_t^\top L_4 \Delta C_t\|_F \leq \|L_4\|_2 \|\Delta C_t\|_F^2.$$

Consequently,

$$\|L_{c,t}\|_F \leq \|C_{t-1}\|_2^2 \|L_1\|_F + 2 \|C_{t-1}\|_F \|L_2\|_2 \|\Delta C_t\|_F + \|L_4\|_2 \|\Delta C_t\|_F^2. \tag{34}$$

Finally, since $L_{\ell,t} = P^\top L_{c,t} P$, we have

$$\|L_{\ell,t}\|_F \leq \|P^\top\|_2 \|L_{c,t}\|_F \|P\|_2 = \|P\|_2^2 \|L_{c,t}\|_F,$$

and therefore, by the triangle inequality together with (33)–(34),

$$\|L_t - L_{\ell,t}\|_F \leq \|L_t\|_F + \|L_{\ell,t}\|_F \leq \|L_t\|_F + \|P\|_2^2 \|L_{c,t}\|_F$$
$$\leq \|L_t\|_F + \frac{1}{\sigma_{\min}(C_{t-1})^2} \Big( \|C_{t-1}\|_2^2 \|L_1\|_F + 2 \|C_{t-1}\|_F \|L_2\|_2 \|\Delta C_t\|_F + \|L_4\|_2 \|\Delta C_t\|_F^2 \Big),$$

which is (19). Substituting (19) into (18) yields the claimed $\epsilon$-similarity bound. Moreover, since $RE = \|L_t - L_{\ell,t}\|_F^2$, we have $\|L_t - L_{\ell,t}\|_F = \sqrt{RE}$ and hence $\epsilon \leq \sqrt{RE}/\lambda_2(L_t)$, completing the proof. $\qquad\square$

$\epsilon$**-Similarity:** Following the analysis in (Kumar et al., 2023b; Loukas, 2019), we provide a guarantee on the quality of the approximation. To quantify how well the Laplacian matrix of the original graph, $\mathbf{L}$, aligns with its reduced counterpart, $\mathbf{L_c}$, we establish an isometry guarantee. Specifically, we state that $\mathbf{L}$ and $\mathbf{L_c}$ are $\epsilon$-similar, defined as:

$$(1-\epsilon)\|X\|_\mathbf{L} \leq \|X_c\|_{\mathbf{L_c}} \leq (1+\epsilon)\|X\|_\mathbf{L} \tag{35}$$

for all $X \in \mathbb{R}^N$, where the norms $\|X\|_\mathbf{L} = \text{tr}(X^T \mathbf{L} X)$, $\|X_c\|_{\mathbf{L_c}} = \text{tr}(X_c^T \mathbf{L_c} X_c)$ and $X_c = C^\dagger X$.

*Proof.* We have $\|X\|_\Theta = \sqrt{\text{tr}(X^T \Theta X)}$ and $\|\tilde{X}\|_\Theta = \sqrt{\text{tr}(\tilde{X}^T \Theta_c \tilde{X})}$. Taking the absolute difference between $\|X\|_\Theta$ and $\|\tilde{X}\|_{\Theta_c}$, we get:

$$\left| \|X\|_\Theta - \|\tilde{X}\|_{\Theta_c} \right| = \left| \sqrt{\text{tr}(X^T \Theta X)} - \sqrt{\text{tr}(\tilde{X}^T \Theta_c \tilde{X})} \right| \tag{36}$$

As $\Theta$ is a positive semi-definite matrix using Cholesky's decomposition $\Theta = S^T S$ in (36), we get the following inequality:

$$\left| \|X\|_\Theta - \|\tilde{X}\|_{\Theta_c} \right| = \left| \sqrt{\mathbf{tr}X^T \Theta X} - \sqrt{\mathbf{tr}\tilde{X}^T \Theta_c \tilde{X}} \right| \tag{37}$$

$$= \left| \sqrt{\mathbf{tr}X^T S^T S X} - \sqrt{\mathbf{tr}\tilde{X}^T C^T S^T S C \tilde{X}} \right| \tag{38}$$

$$= \left| \|SX\|_F - \|SP^\dagger PX\|_F \right| \tag{39}$$

$$\leq \|SX - SP^\dagger PX\|_F \tag{40}$$

$$\leq \epsilon \|X\|_\Theta \tag{41}$$

$$\square$$

## T. ACNR Gradients

The gradient of (12) w.r.t. $\mathbf{\Delta C_{N_t}}$ is:

$$
\begin{aligned}
\nabla_{\mathbf{\Delta C_{N_t}}} \mathcal{L}(\mathbf{\Delta C_{N_t}}) = {} & 2\omega \Big( (C_{t-1}^\top + \mathbf{\Delta C_{S_t}}^\top) M_2 + \mathbf{\Delta C_{N_t}}^\top M_4 \Big)^\top (A_{c_t} - A_{c_{t-1}}) \\
& + \zeta \Big( \mathbf{\Delta C_{N_t}} X_{sn} - X_n \Big) X_{sn}^\top \\
& + \beta(-D)^\top \Big( C_{t-1} + \mathbf{\Delta C_{S_t}} - D\mathbf{\Delta C_{N_t}} \Big) \\
& + 2 \Big( \mathbf{\Delta C_{N_t}} 1_{n \times 1} - 1_{k \times 1} \Big) 1_{n \times 1}^\top.
\end{aligned}
\tag{42}
$$

and w.r.t. $\mathbf{\Delta C_{S_t}}$ is:

$$
\begin{aligned}
\nabla_{\mathbf{\Delta C_{S_t}}} \mathcal{L}(\mathbf{\Delta C_{S_t}}) = {} & 2\omega \Big( A_{t-1} C_{t-1} + A_{t-1} \mathbf{\Delta C_{S_t}} + M_2 \mathbf{\Delta C_{N_t}} \Big) (A_{c_t} - A_{c_{t-1}}) \\
& + \beta \Big( C_{t-1} + \mathbf{\Delta C_{S_t}} - D\mathbf{\Delta C_{N_t}} \Big) \\
& + 2\alpha \Big( C_{t-1} A_{t-1}^{\text{new}} + \mathbf{\Delta C_{S_t}}^\top A_{t-1}^{\text{new}} \Big) \\
& \times \Big( (C_{t-1} + \mathbf{\Delta C_{S_t}})^\top A_{t-1}^{\text{new}} (C_{t-1} + \mathbf{\Delta C_{S_t}}) - C_{t-1}^\top A_{t-1} C_{t-1} \Big).
\end{aligned}
\tag{43}
$$

