# OpenReview forum: "Adapting to Evolving Graphs: A Scalable Framework for Dynamic Coarsening"
_ICML.cc/2026/Conference — ICML 2026 regular_

### Official Review · Reviewer_QaaT · 2026-03-10

**Soundness:** 2
**Presentation:** 3
**Significance:** 3
**Originality:** 3
**Overall Recommendation:** 4
**Confidence:** 4

**Summary:**

This work presents two algorithms, namely **Dynamic Incremental Graph Coarsening (DIGC)** and **Adaptive Coarsening with Node Reassignment (ACNR)**, to perform graph coarsening on temporal graphs under two settings: when graph updates primarily affect newly added nodes (**DIGC**) and when the overall graph structure changes over time (**ACNR**).

These algorithms, i.e, **ACNR** and **DIGC**, involve solving an optimization problem to obtain a valid coarsening matrix, under the assumption that changes between consecutive graph snapshots are relatively small. The primary goal of the optimization objective is to estimate updates to the change in coarsening matrix as new nodes, node features, and edges are observed. **DIGC** only adds new rows to the matrix corresponding to newly added nodes, whereas **ACNR** also estimates changes in existing rows to adapt to evolving topology. In both cases, the initial estimate is obtained using an existing graph coarsening technique applied to the first instance of the graph.

The work then presents numerical experiments on discrete-time dynamic graphs to evaluate the effectiveness of the proposed algorithms and includes ablation studies for various components of the optimization objective.

**Compliance With Llm Reviewing Policy:**

Affirmed.

**Final Justification:**

While the paper is empirically strong, it offers limited theoretical novelty. In particular, the latest experiments demonstrated in the rebuttal show significant improvements in both time and memory efficiency which cannot be overlooked. The substantial efficiency gains reported significantly strengthen the paper’s practical utility, even if the theoretical contributions remain incremental.

**Key Questions For Authors:**

### Q1. Can the authors provide additional baselines without coarsening to serve as a reference for node classification on the original graphs? In particular, it would be helpful to include comparisons in terms of node classification performance, runtime analysis, GPU usage, and loss convergence plots.

### Q2. Can the authors provide an ablation study that evaluates the effect of the individual components of the optimization objective on the overall predictive performance (e.g., node classification accuracy, AUC, or F1 score)?

### Q3. Can the authors provide a more meaningful interpretation of the convergence theorem and clarify what practical advantages it provides compared to existing approaches? Additional clarification or empirical validation would help contextualize the theoretical contribution.

**Limitations:**

No. The paper does not explicitly discuss limitations of the proposed approach. The authors should include a discussion of limitations of the optimization-based formulation. For instance, the approach may incur high computational overhead if graph snapshots are observed at high frequency or in a continuous-time dynamic graph (CTDG) setting. It would also be useful to discuss the potential effects of poor initialization of the coarsening matrix and the robustness of the method to noisy graph observations.

There are no apparent negative societal impacts associated with the work.

**Strengths And Weaknesses:**

## Soundness:
The submission is technically sound, with claims supported through empirical experiments. The experiments primarily involve performing node classification on discrete-time dynamic graphs (DTDGs). Although the experiments are designed to support the effectiveness of the proposed graph coarsening algorithms, they do not present a clear picture of how effective the coarsening is compared to using the full graph for the same task. In particular, the experiments do not include a baseline showing how standard GNNs perform when no coarsening is applied. As a result, it is difficult to assess the advantages of coarsening itself in the context of dynamic graphs.

Additionally, the runtime analysis does not include the overhead of the coarsening procedure on the overall pipeline (Graph -> Coarsening -> GNNs vs Graph->GNNs). Since the proposed approach introduces an additional optimization step for each new graph instance, reporting the runtime of the model with and without coarsening (including this overhead) would provide a clearer understanding of the computational benefits.

Furthermore, the ablation study does not report node classification performance to highlight the trade-offs associated with different components of the optimization objective.

The paper also includes a theorem on convergence but does not provide supporting experiments or further exposition on the practical implications of this result for the overall algorithm.


## Presentation
The presentation is very clear and easy to follow, with a clear narrative motivating the problem statement. However, there are a few minor issues. In particular, there appears to be an error in line 112 in the definition of $\mathcal{C}$. Additionally, the image in panel (c) of Fig. 2 does not seem to correspond to the caption.

## Significance
This work addresses the problem of graph coarsening in dynamic graphs and proposes an optimization-based extension of existing static graph coarsening techniques to the dynamic setting. The problem is well motivated and timely, particularly in the broader context of scalability for learning on evolving graphs.

However, the paper does not discuss alternative approaches in the related work that address closely related problems, particularly graph pooling techniques in GNNs, which attempt to solve a similar and broader problem of scalable graph representation learning. Including a discussion of these methods would help better position the contribution and clarify its significance relative to existing approaches.

## Originality:
The work has limited novelty, as the proposed methods (ACNR and DIGC) largely extend existing graph coarsening techniques to snapshot-based dynamic graphs. The extension of the existing coarsening techniques appears somewhat ad hoc, where the components of the optimization objective are mostly intuitive rather than clearly derived from a systematic analysis or a principled framework.

That said, the work does introduce a formulation for maintaining and updating a coarsening matrix over time in dynamic graphs, which provides an incremental extension of static coarsening approaches. While this setting is less commonly studied and the proposed mechanisms for incremental updates and node reassignment are reasonable, the overall contribution appears to be more of an adaptation of existing ideas rather than introducing fundamentally new methodological insights.

---

> ### Author Rebuttal · Authors · 2026-03-31
>
> We sincerely thank the Reviewer for their detailed evaluation, and highly constructive feedback. We respond to the concerns below.
>
> ---
>
> ### **1. Experiments**
>
>  We clarify that node classification was included as a representative downstream task, following prior graph coarsening literature, to verify that the coarsened graphs remain useful for learning after compression. Our primary objective is to study dynamic graph coarsening itself specifically, whether the proposed methods can efficiently maintain a coarsened representation over time while preserving key structural, spectral, and feature-related properties. Accordingly, the paper emphasizes metrics such as REE, RE, and HE, and compares against static baselines to evaluate dynamic updating relative to repeated re-coarsening.
>
> Consistent with this focus, our ablation study emphasized structural error metrics, which are directly optimized by our formulations and best capture coarsening quality. We agree that including full-graph baselines and end-to-end comparisons adds practical insight. Accordingly, we conducted the following additional experiments :
>
> (i) Node Classification Accuracy on the Uncoarsened ACM Graph.
> | Time Interval | Accuracy without Coarsening |
> |---|---:|
> | \(t_1\) | 91.89 |
> | \(t_2\) | 92.91 |
> | \(t_3\) | 92.67 |
> | \(t_4\) | 92.58 |
> | \(t_5\) | 92.07 |
>
> (ii) End-to-end total runtime analysis across all timestamps on ACM dataset when initialized using FGC in seconds:
>
> (a) Graph → Coarsening → Node Classification, and
>
> (b) Graph → Node Classification.
>
>  | Timestamp | Graph → Node Classification (s) | FGC (s) | DIGC (s) | ACNR (s) |
> |---|---:|---:|---:|---:|
> | \(t_1\) | 0.72 | 5.99 | 3.47 | 3.75 |
> | \(t_2\) | 0.74 | 6.71 | 3.27 | 4.43 |
> | \(t_3\) | 0.87 | 7.69 | 3.84 | 3.71 |
> | \(t_4\) | 1.14 | 13.14 | 3.54 | 4.38 |
> | \(t_5\) | 1.74 | 16.81 | 5.23 | 5.28 |
>
> (iii) Ablation on downstream node-classification performance on ACM dataset when initialized using LVN:
>
> | Setting | \(t_1\) | \(t_2\) | \(t_3\) | \(t_4\) | \(t_5\) |
> |---|---:|---:|---:|---:|---:|
> | w/o 4th Term | 88.45 | 86.95 | 84.36 | 83.36 | 80.95 |
> | w/o 3rd Term | 88.29 | 86.47 | 84.28 | 83.89 | 82.58 |
> | w/o 2nd Term | 89.14 | 87.14 | 82.74 | 82.63 | 81.06 |
> | w/o 1st Term | 88.05 | 86.82 | 84.12 | 83.83 | 82.51 |
> | w/o 2nd \& 4th Terms | 89.26 | 86.28 | 83.39 | 80.48 | 72.22 |
> | w/o 2nd \& 3rd Terms | 89.86 | 87.19 | 84.31 | 81.92 | 78.68 |
> | **No Terms Removed** | **91.45** | **90.22** | **89.59** | **88.71** | **87.16** |
>
> ---
>
> ### **2. Interpretation of the convergence theorem**
> Our convergence result should be interpreted as a first-order stationarity guarantee for the incremental optimization problems solved by DIGC and ACNR: the iterates approach a stable stationary point of the dynamic coarsening objective, rather than oscillating. Thus, the theorem does not claim global optimality, but it does show that the proposed update procedure is algorithmically well-behaved.
>
> The practical advantage is therefore **stability of the optimization-based update itself**. Since DIGC and ACNR solve non-convex incremental objectives at every snapshot, the theorem provides assurance that these updates converge to a meaningful stationary solution. In contrast, heuristic update rules typically do not offer such guarantees, while recomputing from scratch repeatedly is substantially more expensive.
>
> In the revision we will clarify that the theorem supports the reliability of the incremental optimization procedure, while the empirical results demonstrate its practical value through lower runtime/memory and competitive performance.
>
> ---
>
> ### **3. Novelty**
>
>  To the best of our knowledge, there is currently no prior work on dynamic graph coarsening in the sense of maintaining and updating a coarsening map over an evolving graph. Our work is motivated by this gap.
>
> That said, we respectfully disagree that the objective is ad hoc. The terms are introduced to address distinct aspects of the dynamic setting that do not arise in static coarsening. In DIGC, the objective captures incremental evolution by preserving consistency with the previous coarsened graph while assigning newly arrived nodes to the existing supernode space. In ACNR, the additional terms address broader topology changes by allowing correction of outdated assignments and explicitly modeling changes among previously existing nodes. The alignment mechanism further enables handling deletions and non-monotonic evolution.
>
> Therefore, the novelty is not in proposing a new generic optimization primitive, but in formulating dynamic graph coarsening as an optimization-based maintenance problem and designing update mechanisms that systematically handle node addition, node deletion, edge modification, and new-node features more efficiently than repeated static re-coarsening.
>
> ---
>
> We would appreciate clarification on any remaining concerns you have, and ask if you might be open to increasing your score if we have addressed all of them.

---

> > ### Author Rebuttal · Reviewer_QaaT · 2026-04-02
> >
> > While the approach is interesting, I remain concerned about its practical feasibility given the current results. The data suggests that utilizing the DIGC/ACNR block increases runtime by a factor of **4.5x to 5x** while also leading to a degradation in performance. In light of this, I would appreciate more insight into the specific trade-offs or scenarios where this model would be more beneficial than processing the full graph.
> >
> > To better evaluate the utility of this method, could the author please provide a memory analysis of peak GPU usage?Specifically, it would be helpful to see if there is a significant reduction in the memory required during training for node classification on the coarsened graph versus the complete graph.
> >
> > Additionally, to demonstrate the versatility of the proposed framework, could the author provide results for a Graph Classification task? Expanding the evaluation beyond node classification would help clarify if the coarsening benefits are task-dependent.
> >
> > For the time being, I am keeping my overall recommendation unchanged, pending further clarification on these points.

---

> > > ### Author Response · Authors · 2026-04-04
> > >
> > > ---
> > > ## **1. Time and GPU utilization**
> > > ---
> > >  We thank the reviewer for this thoughtful follow-up and sincerely apologize for the confusion in our earlier runtime reporting. After carefully re-examining our implementation, we found that the previously reported runtime did not correspond solely to the actual Graph → Coarsening → Node Classification pipeline. Instead, it also included several auxiliary evaluation procedures used in the paper, including the computation of **Relative Eigenvalue Error (REE)**, **Hyperbolic Error (HE)** and **Reconstruction Error (RE)**. In particular, REE computation requires eigenvalue decomposition of both the original graph and the coarsened graph. Computing the eigenvalues of the original graph is especially time-consuming with complexity on the order of O($p^3$), where $p$ is the number of nodes in the graph. As a result, these additional evaluation steps substantially increased the previously reported runtime.
> > >
> > > To address this issue, we now report the corrected runtime for the actual Graph → Coarsening → Node Classification pipeline and the runtime of the corresponding Graph → Node Classification baseline without coarsening for both **ACM** and **DBLP**. These results are reported at coarsening ratio $r = 0.25$, and we provide a clear breakdown of the total runtime into coarsening time and node-classification time. The corrected results show that our proposed framework is in fact substantially more efficient than direct full-graph method, yielding clear end-to-end speedups. In addition, we also report the peak GPU memory utilization for both datasets below.
> > >
> > > In the runtime tables below, values for DIGC and ACNR are written as **z (x + y)**, where:
> > > - **x** is the coarsening time,
> > > - **y** is the node-classification time, and
> > > - **z** is the total time for **graph → coarsen → classify**.
> > >
> > > For the full-graph, we report the total time for graph → node classification
> > >
> > > ---
> > > ### **1.1 ACM**
> > >
> > > ---
> > > **`(i) Total runtime`**
> > >
> > > | Time | Full graph(s) | DIGC(s) | ACNR(s) |
> > > |---|---:|---:|---:|
> > > | $ t_1$ | 0.72 | 0.23(0.06+0.17) | 0.33(0.16+0.17) |
> > > | $t_2$ | 0.74 | 0.20(0.03+0.18) | 0.23(0.06+0.17) |
> > > | $t_3$ | 0.87 | 0.12(0.02+0.11) | 0.16(0.06+0.10) |
> > > | $t_4$ | 1.14 | 0.20(0.03+0.18) | 0.24(0.07+0.17) |
> > > | $t_5$ | 1.74 | 0.18(0.02+0.16) | 0.22(0.08+0.15) |
> > >
> > > ---
> > >
> > > **`(ii) Peak GPU memory`**
> > >
> > > | Time | Full graph(MiB) | DIGC(MiB) | ACNR(MiB) |
> > > |---|---:|---:|---:|
> > > | $t_1$ | 865.1 | 378.2 | 508.2 |
> > > | $t_2$ | 1204.3 | 565.1 | 675.1 |
> > > | $t_3$ | 1406.7 | 724.3 | 854.3 |
> > > | $t_4$ | 1718.2 | 936.7 | 1086.7 |
> > > | $t_5$ | 1941.5 | 1068.2 | 1278.2 |
> > >
> > > ---
> > >
> > > ### **1.2 DBLP**
> > >
> > > ---
> > >
> > > **`(i) Total runtime`**
> > >
> > > | Time | Full graph(s) | DIGC(s) | ACNR(s) |
> > > |---|---:|---:|---:|
> > > | $ t_1$ | 0.58 | 0.26(0.11+0.15) | 0.30(0.14+0.16) |
> > > | $t_2$ | 0.63 | 0.13(0.02+0.11) | 0.27(0.11+0.16) |
> > > | $t_3$ | 0.82 | 0.21(0.03+0.17) | 0.30(0.13+0.17) |
> > > | $t_4$ | 0.93 | 0.19(0.03+0.16) | 0.28(0.13+0.15) |
> > > | $t_5$ | 1.24 | 0.19(0.03+0.16) | 0.29(0.14+0.15) |
> > >
> > > **`(ii) Peak GPU memory`**
> > >
> > > | Time | Full graph(MiB) | DIGC(MiB) | ACNR(MiB) |
> > > |---|---:|---:|---:|
> > > | $t_1$ | 896.2 | 328.1 | 554.6 |
> > > | $t_2$ | 1201.2 | 438.6 | 691.8 |
> > > | $t_3$ | 1448.1 | 479.9 | 774.9 |
> > > | $t_4$ | 1778.6 | 634.0 | 908.3 |
> > > | $t_5$ | 2216.3 | 799.8 | 1167.8 |
> > >
> > > ---
> > >
> > >
> > >
> > >
> > > ## **2. Graph Classification**
> > >
> > > ---
> > >
> > > We additionally conducted a graph classification experiment on a synthetically converted discrete dynamic version of REDDIT-BINARY, which contains 2,000 graphs for binary classification. Following Appendix E.2, each graph is initialized with 50% of its nodes and then expanded by 10% additional nodes per time step over five discrete snapshots, retaining only edges among visible nodes at each step. We use $r = 0.5$, initialize the first snapshot with **FGC**, and compare **No coarsening**, **FGC**, **DIGC**, and **ACNR** under an 80/10/10 train/validation/test split.
> > >
> > > ---
> > >
> > > **`(i) Total runtime`**
> > >
> > > | Time | Full graph(s) | FGC(s) | DIGC(s) | ACNR(s) |
> > > |---|---:|---:|---:|---:|
> > > | $t_1$ | 339.9 | 354.9 | 150.9 | 181.8 |
> > > | $t_2$ | 324.5 | 413.3 | 156.8 | 194.0 |
> > > | $t_3$ | 345.7 | 416.9 | 153.4 | 203.3 |
> > > | $t_4$ | 317.5 | 378.6 | 163.6 | 228.7 |
> > > | $t_5$ | 530.1 | 409.5 | 175.7 | 243.0 |
> > >
> > > ---
> > >
> > > **`(ii) Peak GPU memory`**
> > >
> > > | Time | Full graph(MiB) | FGC(MiB) | DIGC(MiB) | ACNR(MiB) |
> > > |---|---:|---:|---:|---:|
> > > | $t_1$ | 163.8 | 186.5 | 80.4 | 109.4 |
> > > | $t_2$ | 186.3 | 245.8 | 86.8 | 123.2 |
> > > | $t_3$ | 217.4 | 312.6 | 97.5 | 147.5 |
> > > | $t_4$ | 246.9 | 393.6 | 111.5 | 154.9 |
> > > | $t_5$ | 283.5 | 477.3 | 124.3 | 178.1 |
> > >
> > > ---
> > >
> > > **`(iii) Graph classification accuracy`**
> > >
> > > | Time | Full graph(%) | FGC(%) | DIGC(%) | ACNR(%) |
> > > |---|---:|---:|---:|---:|
> > > | $t_1$ | 72.7 | 65.3 | 65.7 | 67.6 |
> > > | $t_2$ | 72.5 | 66.4 | 65.0 | 67.9 |
> > > | $t_3$ | 74.8 | 67.9 | 67.4 | 69.5 |
> > > | $t_4$ | 74.2 | 67.1 | 67.2 | 70.5 |
> > > | $t_5$ | 77.7 | 70.2 | 69.5 | 72.8 |
> > >
> > > ---
> > >
> > > We hope these additional results address the reviewer’s concerns, and we would appreciate reconsideration of the current score in light of this evidence.

---

### Official Review · Reviewer_cReZ · 2026-03-12

**Soundness:** 3
**Presentation:** 3
**Significance:** 3
**Originality:** 3
**Overall Recommendation:** 5
**Confidence:** 3

**Summary:**

1. This paper studies the problem of graph coarsening in a dynamic setting. Given an evolving graph at discrete time instants, the authors present a scalable dynamic graph coarsening framework, an incrementally updated mapping matrix is also obtained.
2. Two optimization-based algorithms, DIGC and ACNR, are proposed for the dynamic coarsening matrix computation.
3. Experiments are conducted to show the efficacy of the proposed algorithms on 6 dynamic datasets, over 5 state-of-the-art and baseline methods.

**Compliance With Llm Reviewing Policy:**

Affirmed.

**Final Justification:**

My final recommendation is positive. The paper is technically sound and empirically solid, and the rebuttal addressed my main concerns about clarity and scalability, which improved my evaluation and led me to raise my score by one point.

**Key Questions For Authors:**

1. It would be helpful to add a notation table to improve readability.
2. While the optimization objectives and update methods are explained, the overall workflow is a bit unclear. Providing a high-level overview or a simple diagram could help clarify the process.
3. Although the experiments analyze GPU usage, could the authors also provide a theoretical analysis of the space complexity?

**Limitations:**

yes

**Strengths And Weaknesses:**

S1. The paper is well-motivated, tackling the meaningful problem of graph coarsening on discrete-time dynamic graphs.

S2. The experiments are quite thorough, covering several different types of evolving graphs and multiple evaluation metrics to clearly show that the proposed methods work well.

S3. The theoretical analysis is sound, and the authors provide solid guarantees for both the algorithm convergence rates and the \epsilon-similarity bounds.

W1. Some notations are not introduced immediately where they first appear, such as $\omega$ and $\zeta$ in Eq. (3), or $d$ in the complexity analysis of Algorithm 1, making the paper a bit hard to follow.

W2. The scalability of the proposed algorithms is a bit questionable, as the datasets used in the experiments are relatively limited in scale.

W3. The novelty of the proposed method could be highlighted further.

---

> ### Author Rebuttal · Authors · 2026-03-30
>
> We sincerely thank Reviewer for their time and highly constructive feedback. We are encouraged by the positive assessment of our problem motivation, experimental thoroughness, and theoretical soundness. We respond to the concerns below.
>
> ---
>
> ### **1. Notation Table**
>
> We thank the reviewer for the suggestion. We will certainly add a comprehensive notation table in the updated manuscript to improve clarity.
>
> ---
>
> ### **2. Workflow**
>
> We thank the reviewer for this helpful suggestion. To improve the presentation, we will add a high-level pipeline view of our framework in the revision:
>
> 1. **Initialization:** Starting from the first snapshot $G_0$, we compute an initial coarsening map $C_0$ using any static graph coarsening method. The corresponding coarsened graph is then obtained through the standard coarsening relation in Eq. (1).
> 2. **Snapshot Update:** When a new snapshot $G_t$ arrives, we examine the type and extent of graph evolution relative to $G_{t-1}$.
> 3. **Update Mechanism:**
> * **DIGC:** We apply DIGC when the new snapshot can be handled by preserving the previous assignments of existing nodes and updating the coarsening incrementally. This naturally covers the common incremental-growth case, and can also apply to certain node deletion or edge modification scenarios, as long as the existing assignments remain reliable and do not require remapping.
> * **ACNR:** We apply ACNR when the new snapshot requires reassignment of existing-node mappings, such as in cases of stronger structural drift, deletion effects, or topology changes that cannot be adequately captured while keeping prior assignments fixed.
> 4. **Coarsening-Map Update:** Using the selected update rule, we obtain the new mapping $C_t$ by updating $C_{t-1}$ rather than recomputing the coarsening from scratch. Concretely, this is done via Eq. (4) for DIGC and Eq. (6) for ACNR, while the iterative optimization is carried out using Eq. (11) for DIGC and Eqs. (15)–(16) for ACNR.
> 5. **Coarse-Graph Construction:** The updated coarsened graph $G_{c,t}$ is then constructed from $C_t$ using the same coarsening relation as in Eq. (1), and the process is repeated for subsequent snapshots.
>
> ---
>
> ### **3. Scalability and Space Complexity**
>
> We appreciate this helpful suggestion and will include a detailed space complexity analysis in the revised manuscript.
>
> Let $N_t$ be the number of previously existing nodes at time $t$, $k$ the number of newly added nodes, $n$ the number of supernodes, and $d$ the feature dimension.
>
> From Eq. (4), DIGC maintains $C_{t-1}\in\mathbb{R}^{N_t\times n}$ and the incremental block $\Delta C_t\in\mathbb{R}^{k\times n}$. From Eq. (6), ACNR additionally maintains the reassignment block $\Delta C_t^S\in\mathbb{R}^{N_t\times n}$ together with $\Delta C_t^N\in\mathbb{R}^{k\times n}$. The objectives also involve the old–new connectivity blocks, the new-node block, the feature matrices, and the coarse adjacency.
>
> Accordingly, the per-step **space complexity** is:
>
> * **DIGC:** $$O\big(N_t n + kn + \mathrm{nnz}(M_2) + \mathrm{nnz}(M_4) + nd + kd + n^2\big)$$
>
> * **ACNR:** $$O\big(2N_t n + kn + \mathrm{nnz}(D) + \mathrm{nnz}(A_{t-1}^{new}) + \mathrm{nnz}(M_4) + nd + kd + n^2\big)$$
>
> In the **dense worst case**, these complexities become:
>
> * **DIGC:** $$O\big(N_t n + kn + N_tk + k^2 + nd + kd + n^2\big)$$
>
>   *Dominant terms:* $O(N_t n + N_tk)$
>
> * **ACNR:** $$O\big(2N_t n + kn + N_tk + N_t^2 + k^2 + nd + kd + n^2\big)$$
>
>   *Dominant terms:* $O(2N_t n + N_t^2)$
>
> Intuitively, ACNR has a higher memory cost because it additionally maintains reassignment variables for existing nodes and refinement structures for topology changes. In contrast, DIGC only maintains the incremental update while efficiently preserving prior assignments.
>
> ---
>
> ### **4. Novelty**
>
> Please refer to the **"Novelty Discussion"** section in our response to **Reviewer GiWx** for a comprehensive detailing of our key contributions.
>
> ---
>
> We would appreciate clarification on any remaining concerns you have, and ask if you might be open to increasing your score if we have addressed all of them.

---

> > ### Author Rebuttal · Reviewer_cReZ · 2026-04-03
> >
> > Thank you for the detailed response. It addresses my main concerns and clarifies the presentation well. I appreciate the planned revisions, and I have increased my score by one point.

---

> > > ### Author Response · Authors · 2026-04-05
> > >
> > > Thank you for your thoughtful follow-up and for taking the time to read our rebuttal carefully. We are grateful that our clarifications addressed your main concerns and that the proposed revisions were helpful. We also sincerely appreciate your decision to increase the score. Your feedback has been very valuable in helping us improve the paper, and we will make sure the final version reflects the clarifications and presentation improvements discussed in the rebuttal.

---

### Official Review · Reviewer_r4xR · 2026-03-12

**Soundness:** 3
**Presentation:** 3
**Significance:** 2
**Originality:** 2
**Overall Recommendation:** 3
**Confidence:** 4

**Summary:**

The authors consider coarsening a sequence of graphs such that the sequence of coarsenings do not change much. The approach is to regularize distances between two consecutive coarsenings. The results are      shown on only two datasets. Overall, there is not enough novelty to accept this as a paper.

**Compliance With Llm Reviewing Policy:**

Affirmed.

**Final Justification:**

The rebuttal addressed many of the questions, but the novelty of the method is still a bit light. I've upgraded my score accordingly.

**Key Questions For Authors:**

Please see weaknesses above.

**Limitations:**

yes

**Strengths And Weaknesses:**

Strengths:
- Straightfoward and clear description of the problem and the motivation.

Weaknesses:
- Page 2, Eq. 1: What is the definition of the lifting matrix?
- Eq. 2: The equation says "<C_i, C_j>=d_i"; presumably, the authors mean "<C_i, C_i>=d_i".
- Surely existing methods for continuous time can be applied to discrete time; the continuous-time setting is strictly more general, no? The authors should clarify why such methods are inapplicable for their problem.
- Eq. 5: Why is consistency being achieved by a loss h(.)? Won't the loss mean that sometimes the result is inconsistent? Perhaps I do not understand the h(...) term.
- Eq. 5: It seems f(.,.) and \Tau(.,.) both take two supernode clusters and ensure similarity between them. Why do we need two different functions here?
- Algorithm 1 uses Eq. 11, which does not guarantee that the matrices C_t are always consistent, since it converted a hard constraint into a penalty. The same comment also applies to Algorithm 2.
- Can you theoretically prove any relationship between the coarsened graph according to the proposed methods versus the coarsening we would obtain if we jointly solved for all timesteps instead of making incremental   updates?
- Table 1 shows that the proposed methods are better than recomputing from scratch. This is strange. The methods being compared against are static methods, and the measures are also for static methods? The authors     suggest this is because the temporal data helps to "prevent snapshot-level overfitting and stabilizes spectral structure." Perhaps they could explain this in more detail.
- Continuing the above point, Section 4.5 notes that DIGC suffers from poor initialization. Even then, it is able to do better than recomputing from scratch (Table 1)?
- How are the parameters \omega, \zeta, etc. being set? It would seem these parameters offer a lot of flexibility. It would not be possible to set them by cross-validation, at least for the T=0 to T=1 step, because    there is no relevant data...
- Only two datasets are shown. This makes it difficult to judge the generality of the results.

---

> ### Author Rebuttal · Authors · 2026-03-30
>
> We thank the Reviewer for their evaluation and for highlighting the clarity of our problem description. We respond to the concerns below.
>
> ---
>
> **Number of Datasets Evaluated:** We would like to clarify that the evaluation is broader than two datasets. Besides ACM and DBLP, we also evaluate on AS-733 and three synthetic dynamic datasets in the appendix. Due to space constraints, the main paper highlights ACM & DBLP for the core comparison, while AS-733 and the synthetic datasets are included to cover additional settings.
>
> ---
>
> **Continuous-Time vs. Discrete-Time:** We agree that CTDGs are more general as a temporal representation. At the same time, to the best of our knowledge, there is currently no prior general-purpose dynamic graph coarsening method, in either the CTDG or DTDG setting, that maintains a valid coarsening matrix over time for learning-oriented, structure-preserving compression.
>
> Even if one discretizes a continuous-time method into snapshots, this does not make it a dynamic graph coarsening method. Such discretization collapses within-window event order and inter-event timing, can map multiple non-equivalent event streams to the same snapshot sequence, and introduces an additional window-size design choice that changes the observed graph evolution. More importantly, after discretization, the fundamental mismatch still remains: existing CTDG methods are designed to learn temporal embeddings, hidden states, or event predictions, whereas our method must maintain an explicit node-to-supernode coarsening map $C_t$ and coarsened graph at each step under node addition, node deletion, and edge modification.
>
> This distinction applies broadly across temporal-attention methods, point-process, temporal-walk embeddings, and continuous latent-dynamics models.
>
> ---
>
> **Why Temporal Regularization Can Improve Snapshot-Level Metrics:** Table 1 is a fair comparison because all methods are evaluated with the same snapshot-level metrics. The static baseline recomputes coarsening independently at each snapshot, while DIGC/ACNR use $C_{t-1}$ as a temporal prior. When graph evolution is smooth, this prior can reduce sensitivity to snapshot noise and stabilize the coarsening, so better snapshot-level performance is not contradictory, though the gain is not universal across all metrics.
>
> ---
>
> **Why Poor Initialization Does Not Contradict Table 1:** These results come from **different settings**. Table 1 uses **normal initialization** on ACM/DBLP. Appendix I uses an **intentionally degraded initialization** on AS-733 to test robustness and error recovery. Therefore, our statement that DIGC suffers from poor initialization does not conflict with Table 1, since Table 1 does not evaluate DIGC under poor initialization.
>
> ---
>
> **h(.) term:** h(.) enforces assignment coherence, not hard equality. It encourages updated old-node assignments, $C_{t-1}+\Delta C_t^S$, to align with assignments implied by old–new connectivity, $D\Delta C_t^N$. A soft penalty is used because exact equality would be too restrictive under genuine topology changes.
>
> ---
>
> **Need for both f(.) and $\tau(.)$:** The two terms serve different purposes. f(.) is a global temporal smoothness term that keeps the overall coarsened graph close to the previous snapshot, while $\tau(.)$ is a targeted topology-refinement term that captures connectivity changes among existing old nodes, which may otherwise be masked by newly added nodes.
>
> ---
>
> **Relationship to Joint Optimization:** Under standard assumptions, the incremental solution $\hat{C}_ {t}$ admits a recursive bound relative to the joint optimum $C_ {t}^{\ast}$:
>
> $$
> \left\lVert \hat{C}_ {t} - C_ {t}^{\ast} \right\rVert_ {F} \le \alpha \left\lVert \hat{C}_ {t-1} - C_ {t-1}^{\ast} \right\rVert_ {F} + \beta \left\lVert \Delta A_ {t} \right\rVert
> $$
>
> For $\alpha < 1$, $\left\lVert \hat{C}_ {t} - C_ {t}^{\ast} \right\rVert_ {F} = \mathcal{O} \left( \sum_ {s=1}^{t} \left\lVert \Delta A_ {s} \right\rVert \right)$.
>
> Thus, the incremental solution stably approximates the joint one. We defer the full derivation due to space but can provide it if helpful.
>
> ---
>
> **Constraint Relaxation:** We use a standard penalty-based relaxation. This is a standard relaxation used to promote desirable properties while retaining a tractable update rule. In practice, it produces sufficiently well-behaved assignment matrices for stable coarsened-graph.
>
> ---
>
> **Hyperparameters:** $(\omega,\zeta,\alpha,\beta)$ are unsupervised structural weights, not label-tuned. Appendix K reports both default and TPE-tuned settings, with only modest differences.
>
> ---
>
> **Lifting Matrix and Correction to Eq. (2):** In Eq. (1), $P=C^\dagger$ is the Moore–Penrose pseudoinverse used for feature projection, i.e., $X_c=PX$. Eq. (2) contains a typo and should read $\langle C_i, C_i\rangle=d_i$.
>
> ---
>
> We would appreciate clarification on any remaining concerns you have, and ask if you might be open to increasing your score if we have addressed all of them.

---

> > ### Author Rebuttal · Reviewer_r4xR · 2026-04-03
> >
> > Thank you for your response. While you have alleviated some concerns, the main ones remain.
> > Constraint relaxation is fine, but the point that it can lead to inconsistent C_t remains. One can do projected gradient steps to fix this, but your algorithm does not seem to do so.
> > Regarding Table 1, I had assumed that the static version being recomputed "from scratch" (line 350, second column) meant "over the entire dataset", not just the current snapshot. To judge the quality of the dynamic    method, one would have to see how close it gets to the optimal C_t matrices, if one had access to all snapshots and could recompute everything. In other words, Table 1 should be augmented with another line for "Base   (over entire dataset)".
> > Regarding the third dataset, I don't think even the appendix has it for all three base methods, only FGC. I must say that three datasets is too few to convince one of the generality of a method.

---

> > > ### Author Response · Authors · 2026-04-05
> > >
> > > We thank the reviewer for the follow-up. We appreciate that several of the earlier concerns were clarified by our previous response, and that the remaining discussion has now narrowed to four concrete questions. We have addressed each of these directly below, including additional clarifications and new experimental results.
> > >
> > > ---
> > > ## **1.Constraint Relaxation:**
> > > ---
> > > We would like to clarify that, in the implementation, **we do apply an explicit post-gradient projection step** that clamp the assignment variables to enforce non-negativity. In addition, we do not explicitly enforce that the entries of the incremental assignment matrix $\Delta C$ are exactly binary, i.e., in $(0,1)$. Instead, the optimization typically yields a soft assignment in which, for each row of $\Delta C$, one entry becomes dominant while the remaining entries are very small. In practice, we convert this soft assignment into a hard assignment by setting the dominant entry in each row to $1$ and the remaining entries to $0$. Thus, the final mapping used in the coarsened graph is effectively discrete, even though the optimization itself is performed in a relaxed continuous space. Our motivation for this design was computational efficiency in the dynamic setting, since exact projection onto the feasible set together with strict binary enforcement at every update step would introduce additional overhead, whereas the relaxed formulation yields a simpler and faster update rule in practice. Empirically, this approach was sufficient to produce stable behavior.
> > >
> > > ---
> > > ## **2. Base (over entire dataset)**
> > > ---
> > > We agree that, in addition to the per-snapshot static recomputation baseline originally reported in Table 1, it is useful to compare against a stronger **offline full-sequence baseline** with access to the entire temporal evolution. Following this suggestion, we computed **Base (over entire dataset)** on Table-1, with the results below.
> > >
> > > | Dataset | Initialization | REE ↓ | RE ↓ | HE ↓ |
> > > |---|---|---:|---:|---:|
> > > | | FGC | 0.452 | 4.615 | 2.786 |
> > > | ACM  | LVN | 0.285 | 4.557 | 2.816 |
> > > | | LHE | 0.237 | 4.752 | 3.091 |
> > > |  |  |  |  |  |
> > > | | FGC | 0.722 | 2.142 | 2.804 |
> > > | DBLP | LVN | 0.285 | 2.028 | 2.785 |
> > > | | LHE | 0.364 | 2.099 | 2.807 |
> > >
> > > The results confirm that the offline full-sequence baseline is stronger than the original per-snapshot static baseline. We will include this for Table 1 in the revised manuscript.
> > >
> > > ---
> > > ## **3. AS-733 on LVN and LHE Baseline**
> > > ---
> > >
> > > We agree that previously **AS-733** was only reported with FGC initialization, which limited the evidence for generality across base coarsening methods. Following this suggestion, we additionally evaluated AS-733 with both **LVN** and **LHE** initializations, using the same protocol as Table 7 and averaging over the 9 temporal transitions. Due to space constraints, we report the averaged values here.
> > >
> > > | Initialization | Method | REE ↓ | RE ↓ | HE ↓ |
> > > |---|---|---:|---:|---:|
> > > | | DIGC | 0.550 | 5.862 | 4.432 |
> > > | LVN | ACNR | 0.320 | 5.356 | 4.063 |
> > > | | LVN (BASE) | 0.554 | 5.740 | 4.334 |
> > > |  |  |  |  |  |
> > > | | DIGC | 0.557 | 5.839 | 4.558 |
> > > | LHE | ACNR | 0.327 | 5.320 | 3.992 |
> > > | | LHE (BASE) | 0.550 | 5.780 | 4.334 |
> > >
> > > These additional results show that the same trend observed with FGC also holds for LVN and LHE that:
> > > * ACNR consistently outperforms the corresponding static baseline on AS-733, and
> > > * DIGC remains competitive while being less robust than ACNR under non-monotonic evolution.
> > >
> > > We will include these results in the revised manuscript.
> > >
> > > ---
> > > ## **4. New Dataset Evaluation**
> > > ---
> > > To further assess generality, we additionally evaluated our method on a new real-world dataset, the **Kindle e-book co-purchasing network**, preprocessed using the same procedure described in Appendix E.1. Due to space constraints, we report the averaged results over the 5 temporal transitions here.
> > >
> > > **`(a) Node Classification Accuracy (%)`**
> > > | Initialization | DIGC | ACNR | BASE |
> > > |---|---:|---:|---:|
> > > | FGC | 55.1 | 55.7 | 56.1 |
> > > | LVN | 54.2 | 54.8 | 55.3 |
> > > | LHE | 54.7 | 56.1 | 56.3 |
> > >
> > > **`(b) Error Metrics`**
> > >
> > > | Initialization | Method | REE ↓ | RE ↓ | HE ↓ |
> > > |---|---|---:|---:|---:|
> > > | | DIGC | 0.94 | 9.66 | 4.77 |
> > > | FGC | ACNR | 0.79 | 9.61 | 4.81 |
> > > | | FGC (BASE) | 0.93 | 9.64 | 4.78 |
> > > |  |  |  |  |  |
> > > | | DIGC | 0.82 | 9.51 | 6.33 |
> > > | LVN | ACNR | 0.72 | 9.48 | 6.18 |
> > > | | LVN (BASE) | 0.83 | 9.60 | 6.30 |
> > > |  |  |  |  |  |
> > > | | DIGC | 0.87 | 9.53 | 6.19 |
> > > | LHE | ACNR | 0.70 | 9.50 | 6.12 |
> > > | | LHE (BASE) | 0.82 | 9.58 | 6.12 |
> > >
> > > These additional results on a new dataset are consistent with the trends reported on ACM/DBLP/AS-733. In addition, the paper already includes experiments on **three synthetic dynamic datasets** in **Appendix J**, so with this new Kindle experiment and the expanded AS-733 results, the empirical evidence is now substantially broader.
> > >
> > > ---
> > >
> > > We hope these clarifications address the reviewer’s concerns. If so, we would sincerely appreciate reconsideration of the current score.

---

### Official Review · Reviewer_GiWx · 2026-03-13

**Soundness:** 3
**Presentation:** 3
**Significance:** 3
**Originality:** 3
**Overall Recommendation:** 4
**Confidence:** 3

**Summary:**

This paper proposes a unified framework for coarsening discrete-time dynamic graphs by incrementally updating the coarsening mapping matrix. It introduces two methods, DIGC for incremental growth and ACNR for broader topology changes, and shows they can preserve structural quality and downstream node-classification performance while greatly reducing runtime and memory.

**Compliance With Llm Reviewing Policy:**

Affirmed.

**Final Justification:**

The authors addressed most of my concerns. I maintain my generally positive rating.

**Key Questions For Authors:**

none.

**Limitations:**

yes

**Strengths And Weaknesses:**

Strengths

1. This paper study a new problem of dynamic graph coarsening, addressing a gap that traditional graph coarsening methods did not consider.
2. The paper proposes two complementary methods, DIGC and ACNR, with ACNR explicitly allowing remapping of old nodes to better handle evolving topology.
3. The method achieves good empirical performance.
4. the paper provides convergence analysis to strengths the technical presentation.

Weaknesses

1. The motivation for studying graph coarsening is not introduced enough. The paper would benefit from a clearer explanation of why dynamic graph coarsening matters in practice and in what real-world scenarios it is actually needed.
2. In dynamic graphs, temporal features are often crucial, especially for downstream prediction tasks. However, the proposed algorithm mainly focuses on structural updates and does not incorporate temporal features, which is a core problem. Especially for evaluation, the dynamic graph models should be used, rather than using GCNs.
3. As I understand it, the proposed framework only updates the coarsening result based on the immediately last-step graph. It does not fully address how approximation errors may accumulate over repeated updates.
4. Both DIGC and ACNR are built on relatively standard optimization ingredients, the convergence result is also a fairly standard smooth non-convex SGD guarantee. The paper would benefit from a clear discussion of its novelty.

---

> ### Author Rebuttal · Authors · 2026-03-30
>
> We thank the reviewer for the positive assessment and the helpful suggestions. We are glad that the reviewer found the problem formulation meaningful, recognized the complementary roles of DIGC and ACNR, and viewed the empirical results favorably. We respond to the concerns below.
>
> ---
>
> ### **1. Practical Motivation for Dynamic Graph Coarsening**
>
> We agree that the practical motivation can be stated more clearly. Graph coarsening is already an important mechanism for scaling graph learning, and prior work has shown that it can reduce training cost while preserving downstream accuracy.
>
> Our paper addresses the setting where such graphs evolve over time, meaning recomputing a fresh coarsening at every snapshot can negate the very efficiency gains that make coarsening attractive. This is particularly relevant for discrete-time evolving graphs such as:
> * Citation networks
> * Interaction graphs
> * Financial systems
> * Scientific simulations
>
> We will strengthen the introduction with these concrete scenarios and provide a more direct explanation of why dynamic maintenance of a coarsened graph is necessary.
>
> ---
>
> ### **2. Incorporating Temporal Features in Dynamic Graphs**
>
> We thank the reviewer for this important point. Our method is primarily designed to learn a coarsened graph for discrete-time evolving graphs under node additions, deletions, and connectivity changes. In the current formulation, we consider feature information only for newly added nodes.
>
> However, because our framework is optimization-based, such temporal feature variation can be seamlessly incorporated by adding an additional term. For example:
>
> $$
> q(\Delta C_ {t}^{S}) = \left\lVert (C'_ {t-1} + \Delta C_ {t}^{S}) X_ {sn}^{t-1} - X_ {\mathrm{keep}}^{t} \right\rVert_ {F}^{2}
> $$
>
> where $X_{\mathrm{keep}}^{t}$ denotes the features at time $t$ of nodes that persist from $t-1$ to $t$. This term encourages preserved nodes to remain consistent with their updated features over time.
>
> ---
>
> ### **3. Error Accumulation Under Repeated Updates**
>
> We appreciate this point. DIGC indeed has this limitation by design: previous assignments are frozen for maximal efficiency, which means errors can become locked in. This is precisely why we introduced ACNR.
>
> ACNR augments the update with adaptive reassignment of old nodes and topology refinement, allowing the coarsening to correct earlier suboptimal assignments under evolving connectivity. This issue is heavily studied empirically in our robustness section (**Appendix I**):
> * **DIGC** degrades under repeated non-monotonic updates and shows error lock-in under poor initialization.
> * **ACNR** remains substantially more stable and progressively recovers from poor initializations.
>
> ---
>
> ### **4. Novelty Discussion**
>
> We agree that the block-coordinate optimization ingredients are standard, and that our convergence theorem is a standard smooth non-convex SGD guarantee. This is not the main novelty of the paper, and we will revise the text to state this more clearly.
>
> Our novelty lies instead in the problem formulation and the dynamic coarsening mechanisms:
>
> 1. **General Framework:** Maintaining a coarsening map over discrete-time evolving graphs starting from *any* static coarsener.
> 2. **DIGC Formulation:** Achieving incremental updates via block decomposition of the evolving graph.
> 3. **ACNR Objective:** Adding adaptive reassignment and topology refinement so previous node assignments can be easily corrected.
> 4. **Alignment Mechanism:** Handling deletions and non-monotonic evolution.
> 5. **$\epsilon$-Similarity Analysis:** A theoretical analysis specialized specifically to dynamic incremental coarsening.
>
> Importantly, these are not only modeling extensions but also computational ones. The framework is deliberately designed to avoid repeated full static re-coarsening at every snapshot. Instead, it updates the coarsening map using only the relevant graph changes. This yields the practical advantage emphasized in our experiments: **substantially lower runtime and memory usage** while fully preserving structural quality and downstream utility.
>
> ---
>
> We would appreciate clarification on any remaining concerns you have, and ask if you might be open to increasing your score if we have addressed all of them.

---

> > ### Author Rebuttal · Reviewer_GiWx · 2026-04-02
> >
> > For weakness 1, I mean one or two real-world applications would be helpful, rather than only listing broad domains.
> >
> > For weakness 2, a more detailed explanation would be beneficial. Can the method preserve/generate temporal ordering among nodes in the coarsened graph? This is essential for downstream prediction tasks on temporal graphs.

---

> > > ### Author Response · Authors · 2026-04-04
> > >
> > > ---
> > > ## **W1: Applications**
> > > ---
> > > One example is large online interaction platforms. At time t, this system is represented as a graph $G_t = (V_t, E_t, X_t)$, where $V_t$ denotes users or items, $E_t$ denotes observed interactions, and $X_t$ denotes node features. As new users join, others become inactive, and new interactions are created, the graph changes from $G_t$ to $G_{t+1}$. In these platforms, graph representations are repeatedly used for recommendation, monitoring, analytics, and downstream tasks such as node classification. For example, node classification can be used to identify spam accounts, predict user communities or interests, or categorize content based on interaction patterns. Because the graph changes over time and keeps growing, these tasks must be performed on increasingly large graph snapshots. In such a setting, graph coarsening is needed to reduce the computational and memory cost of processing the graph. A static approach would compute a new coarsening map $C_t$ from scratch for every snapshot $G_t$, which can be expensive when this must be done repeatedly at scale. Dynamic graph coarsening instead seeks to update the previous coarsening $C_t$ into $C_{t+1}$ by incorporating the changes between snapshots, thereby maintaining a compact graph representation without full recomputation.
> > >
> > > A second example is Internet routing or communication networks. Here, the graph $G_t = (V_t, E_t)$ evolves as routers, autonomous systems, or communication links appear, disappear, or change status over time. Network operators often need reduced graph representations to analyze connectivity and large-scale structure efficiently. However, if the topology changes frequently, recomputing a fresh coarsened graph at every time step may negate the computational advantages of coarsening itself. In this setting, dynamic graph coarsening is useful because it incrementally maintains a reduced graph $G_{c_t}$ as $G_t$ evolves, instead of rebuilding $G_{c_t}$ from scratch after each update.
> > >
> > > ---
> > > ## **W2.1: Incorporating Temporal Features in Dynamic Graphs**
> > > ---
> > >
> > > The temporal regularization term
> > >
> > > $$
> > > \mathcal{L} _{temp} = \gamma \left\Vert (C _{t-1} + \Delta C_t^{S})^\dagger X _{\text{keep}}^{t} - C _{t-1}^\dagger X _{\text{keep}}^{t-1} \right\Vert _F^2
> > > $$
> > >
> > > is proposed to preserve temporal consistency in the coarsened feature space across consecutive graph snapshots. Here, $X_{\text{keep}}^{t-1}$ and $X_{\text{keep}}^{t}$ denote the feature matrices of persistent nodes at times $t-1$ and $t$, while $C_{t-1}$ and $C_t=C_{t-1}+\Delta C_t^{S}$ represent the corresponding coarsening mappings. The pseudo-inverse maps node-level features into supernode-level representations, so $C_{t-1}^\dagger X_{\text{keep}}^{t-1}$ gives the supernode features at time $t-1$, and $(C_{t-1}+\Delta C_t^{S})^\dagger X_{\text{keep}}^{t}$ gives the updated supernode features at time $t$. Their Frobenius-norm difference measures how much the coarsened representations vary over time, and $\gamma$ controls the strength of this penalty.
> > >
> > > Intuitively, this loss encourages the coarsened graph to evolve smoothly despite changes in node features or assignment updates. It penalizes abrupt shifts in supernode embeddings caused by unstable mappings or sudden feature changes, thereby improving stability and robustness. This term explicitly models the temporal feature evolution of persistent nodes, making the framework suitable for real dynamic graphs.
> > >
> > > ---
> > > ## **W2.2: Preservation or Generation of temporal node ordering**
> > > ---
> > >
> > > We thank the reviewer for this important question. Our current method does not explicitly preserve or generate temporal ordering among nodes within the coarsened graph. Instead, it operates on **discrete-time graph snapshots** $G_{t_1}, G_{t_2}, \dots, G_{t_n}$ and incrementally updates the coarsening matrix $C_t$ so that the coarsened graph remains structurally representative as the original graph evolves. Thus, the temporal aspect is modeled at the **snapshot level**, not as an ordering constraint among nodes within each coarsened graph.
> > >
> > > Accordingly, the method preserves **structural and feature-level consistency across snapshots** and is evaluated using structural quality metrics and downstream node classification performance over time, rather than temporal-order preservation. The objectives in DIGC and ACNR focus on maintaining consistency between consecutive coarsened graphs and efficiently updating node-to-supernode assignments under node and edge changes.
> > >
> > > We agree that explicit temporal-order preservation may be important for sequence-sensitive tasks such as event prediction or forecasting. This is a valuable direction for future work, where temporal regularization or positional encodings could be incorporated. We will clarify this limitation in the revised paper.
> > >
> > > ---
> > >
> > > We hope these clarifications address the reviewer’s concerns. If so, we would sincerely appreciate reconsideration of the current score.

---

### Decision · Program_Chairs · 2026-04-30

**Decision:**

Accept (regular)

**Comment:**

This paper introduces a framework for dynamic graph coarsening on discrete-time dynamic graphs. The core idea is to avoid recomputing the full coarsening at each timestep by leveraging temporal continuity assumptions, allowing the coarsening to be updated incrementally from one snapshot to the next. The framework is instantiated by two algorithms tailored to different dynamic regimes: one targeting settings where updates primarily affect newly added nodes, and one handling more general structural changes over time.

According to the reviewers, the paper is well-motivated and addresses a meaningful practical problem. The experiments are thorough, covering several types of evolving graphs and multiple evaluation metrics, and reviewers agree that the proposed methods work well across these settings. The theoretical analysis is considered sound by the reviewers, who appreciated the guarantees provided for both algorithm convergence rates and epsilon-similarity bounds. The rebuttal was serious and well-received: according to the reviewers, the authors provided clearer runtime and memory usage results, scalability and space complexity analyses, evaluations on new datasets, and an offline full-sequence baseline, all of which substantially strengthen the paper.

One main concern raised by reviewers is novelty: the proposed algorithms largely extend existing graph coarsening techniques to the snapshot-based dynamic setting, and according to some reviewers, this extension appears incremental. As noted by one reviewer, the components of the optimization objective are mostly intuitive rather than derived from a systematic or principled framework, and the novelty with respect to existing coarsening methods could be more explicitly articulated. Scalability and runtime concerns were also raised by reviewers, though these were largely addressed by the rebuttal to the satisfaction of the reviewers who raised them.

While the contribution is viewed by some reviewers as somewhat incremental, the paper has genuine merits that are broadly acknowledged: it is clearly written, well-instantiated, and addresses a relevant problem with solid empirical and theoretical support. Crucially, the rebuttal addressed the main concerns convincingly, and the additional material — runtime comparisons, scalability analysis, new datasets, and the offline baseline discussion — must be incorporated into the final manuscript.